# Scaling Law for SGD in Quadratically Parameterized Linear Regression

## Abstract

In machine learning, the scaling law describes how the model performance improves with the model and data size scaling up. From a learning theory perspective, this class of results establishes upper and lower generalization bounds for a specific learning algorithm. Here, the exact algorithm running using a specific model parameterization often offers a crucial implicit regularization effect, leading to good generalization. To characterize the scaling law, previous theoretical studies mainly focus on linear models, whereas, feature learning, a notable process that contributes to the remarkable empirical success of neural networks, is regretfully vacant. This paper studies the scaling law over a linear regression with the model being quadratically parameterized. We consider infinitely dimensional data and slope ground truth, both signals exhibiting certain power-law decay rates. We study convergence rates for Stochastic Gradient Descent and demonstrate the learning rates for variables will automatically adapt to the ground truth. As a result, in the canonical linear regression, we provide explicit separations for generalization curves between SGD with and without feature learning, and the information-theoretical lower bound that is agnostic to parametrization method and the algorithm. Our analysis for decaying ground truth provides a new characterization for the learning dynamic of the model.

## 1 Introduction

The rapid advancement of large-scale models has precipitated a paradigm shift across AI field, with the empirical scaling law emerging as a foundational principle guiding practitioners to scale up the model. The *neural scaling law* (Kaplan et al., 2020; Bahri et al., 2024) characterized a polynomial-type decay of excess risk against both the model size and training data volume. Originated from empirical observations, this law predict the substantial improvements of the model performance given abundant training resources. Enough powerful validations have supported the law as critical tools for development of model architecture and allocation of computational resources.

From the statistical learning perspective, neural scaling law formalizes an algorithm-dependent generalization that explicitly quantify how excess risk diminishes with increasing model size and sample size. This paradigm diverges from the classical learning theory, which prioritizes algorithm-agnostic guarantees through a uniform convergence argument for the hypotheses. Empirically, the neural scaling law demonstrates a stable polynomial-type decay of excess risk. This phenomenon persists even as model size approaches infinity, challenging the traditional intuitions about variance explosion. Theoretically, this apparent contradiction implies the role of implicit regularization. Learning algorithms, when coupled with specific parameterized architectures, realize good generalization that suppresses variance explosion. The critical interplay between parameterization methods, optimization dynamics, and generalization, positions algorithmic preferences as an implicit regularization governing scalable learning.

Theoretical progress in characterization of the polynomial-type scaling law has largely centered on linear models, motivated by two synergistic insights. First, the Neural Tangent Kernel (NTK) theory (Jacot et al., 2018; Arora et al., 2019) reveals that wide neural networks, when specially scaled and randomly initialized, can be approximated by linearized models, bridging nonlinear architectures to analytically tractable regimes. Second, linear systems allow for precise characterization of learning dynamics. The excess risk of linear model is associated with two key factors, the covariance operator

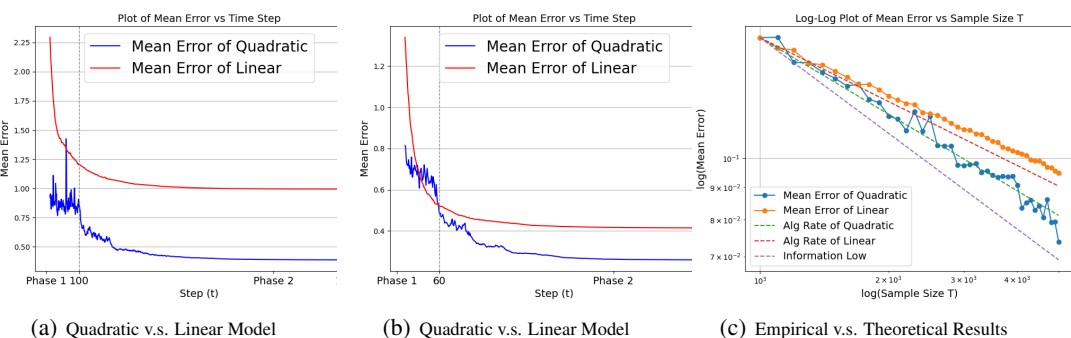

(a) Quadratic v.s. Linear Model    (b) Quadratic v.s. Linear Model    (c) Empirical v.s. Theoretical Results

Figure 1: Empirical results on the convergence rate of quadratically parameterized model with spectral decay v.s. traditional linear model. (a) and (b) show the curve of mean error against the number of iteration steps, with $\alpha = 2.5, \beta = 1.5$ in (a) and $\alpha = 3, \beta = 2$ in (b), respectively. (c) show the logarithmic curve of final mean loss against the sample size, where the solid lines represent the empirical results and the dashed lines represent the theoretical rates.

spectrum and the regularity of ground truth (Lin et al., 2024; Bahri et al., 2024). In the Reproducing Kernel Hilbert Space (RKHS) framework, these factors can be described by the capacity of the kernel and source conditions of the target function (Caponnetto & De Vito, 2007).

Compared with traditional studies in linear regression, recent analyses have shifted focus to high-dimensional problems with non-uniform and fine-grained covariance spectra and source conditions (Caponnetto & De Vito, 2007; Bartlett et al., 2021). The NTK spectrum is shown to exhibit power-law decay when the inputs are uniformly distributed on the unit sphere (Bietti & Mairal, 2019; Bietti & Bach, 2021). In the offline setting, Gradient Descent (GD) and kernel ridge regression (KRR) exhibit the implicit regularization and multiple descents phenomena, under various geometries of the covariance spectrum and source conditions (Gunasekar et al., 2017; Bartlett et al., 2020; Ghorbani et al., 2021; Zhang et al., 2024b). In the more widely studied online setting, Stochastic Gradient Descent (SGD) has been proven to achieve a polynomial excess risk under a power-law decay covariance spectrum and ground truth parameter (Dieuleveut & Bach, 2016; Lin & Rosasco, 2017; Wu et al., 2022).

However, significant gaps persist in explaining the scaling laws when relying on simplified linear models. A primary limitation of these models is their inability to capture the feature learning process, a mechanism that is widely regarded as crucial to the empirical success of deep neural networks (LeCun et al., 2015). This process enables neural networks to autonomously extract high-quality hierarchical representations from data, leading to effective generalization. This limitation arises because linear models inherently restrict the capacity to learn feature representations and tend to rapidly diverge from the initial conditions. In linear models, the parameter trajectory under SGD follows a predictable pattern: the estimation bias contracts at a constant rate proportional to the eigenvalue of each feature, while variance accumulates uniformly. However, neural networks are not constrained by an initial feature set; instead, they adaptively reconfigure their internal representations through coordinated parameter updates. The feature learning can often improve the performances. For example, even the enhanced convolutional neural tangent kernel based on the linearization of neural networks in the infinite-width limit has a performance gap compared to neural networks on the CIFAR10 dataset (Li et al., 2019).

In this paper, we study a quadratically parameterized model: $f(\mathbf{x}) = \langle \mathbf{Sx}, \mathbf{v}^{\odot 2} \rangle$, where $\mathbf{S} \in \mathbb{R}^M \times \mathbb{H}$ is the sketch matrix, and $\mathbf{x} \in \mathbb{H}$ is the input data, and $\mathbf{v} \in \mathbb{R}^M$ are the model parameters, as an alternative testbed to study the scaling law. This model can be regarded as a "diagonal" linear neural network and exhibits feature learning capabilities. As shown in Figure 1 (a) and (b), linear models exhibit a empirically suboptimal convergence rate on excess risk under SGD. This suboptimal performance is not solely attributed to the limitations of SGD itself. As demonstrated in Figure 1 (c), SGD achieves a significantly faster convergence rate on excess risk in quadratically parameterized models, aligning with our theoretical findings. Note that the previous studies for

quadratically parameterized models (HaoChen et al., 2021) often assume a sparse ground truth for the model where the variance will explode with the number of non-zero elements increasing and no polynomial rates are established. We instead consider an infinitely dimensional data input and ground truth, whose signal exhibits certain power-law decay rates. Specifically, for constants $\alpha, \beta > 1$, we assume that the eigenvalues of the covariance matrix decay as $\lambda_i \asymp i^{-\alpha}$ and that $\mathbf{v}_i^*$ the $i$-th alignment coordinate of the ground truth satisfies $\lambda_i \left(\mathbf{v}_i^*\right)^4 \asymp i^{-\beta}$. Suppose the model has access to the sketched covariates and their response, we study the excess risk of quadratically parameterized predictor with $M$ parameters and trained by SGD with tail geometric decay schedule of step size, given $T$ training samples.

We establish the upper bound for the excess risk, demonstrating that its follows a piecewise power law with respect to both the model size and the sample size throughout the training process. More concretely, the upper bounds of the excess risk $\mathcal{R}_M(\mathbf{v}^T) - \mathbb{E}[\xi^2]$ behaves as

$$
\underbrace{\frac{1}{M^{\beta-1}}}_{\text{approximation}} + \underbrace{\frac{\sigma^2 D}{T}}_{\text{variance}} + \underbrace{\frac{D}{T} + \frac{1}{D^{\beta-1}} \mathbb{1}_{D<M}}_{\text{bias}}
$$

where $D = \min\left\{T^{1/\max\{\beta,(\alpha+\beta)/2\}}, M\right\}$ serves as the effective dimension. The above result reveals that, for a fixed sample size, increasing the model size is initially beneficial, but the returns begin to diminish once a certain threshold is reached. Moreover, when the model size is large enough, SGD achieves the excess risk as $\tilde{\mathcal{O}}\left(T^{-1+\frac{1}{\beta}}\right)$ when $\alpha \leq \beta$, and the excess risk as $\tilde{\mathcal{O}}\left(T^{-\frac{2\beta-2}{\alpha+\beta}}\right)$. This indicates that when the true parameter aligns with the covariance spectrum ($\alpha \leq \beta$), the quadratically parameterized model, similar to the linear model, achieves the optimal rate (Zhang et al., 2024a). On the other hand, when the true parameter opposes the covariance spectrum ($\alpha > \beta$), SGD achieves a rate of $\tilde{\mathcal{O}}\left(T^{-\frac{2\beta-2}{\alpha+\beta}}\right)$ in the quadratically parameterized model, which outperforms the best rate SGD can achieve in the linear model $\tilde{\mathcal{O}}\left(T^{-\frac{\beta-1}{\alpha}}\right)$ (Zhang et al., 2024a).

In our analysis, we characterize the learning process of SGD into two typical stages. In the first "adaptation" stage, the algorithm implicitly truncates the first $D$ coordinates to form the effective dimension set $\mathcal{S}$, based on the initial conditions. The variables within $\mathcal{S}$ grow and oscillate around the ground truth, while the remaining variables are constrained by a constant multiple of the ground truth, leading to an acceptable excess risk. In the second "estimation" stage, the variables in the effective dimension set $\mathcal{S}$ converge to the ground truth, while the other variables remain within a region that produces a tolerable level of excess risk. The advantage beyond the linear model is easy to be observed in the "estimation" stage, where the step size is scaled by the certain magnitude of the ground truth due to the adaption, resulting in a faster convergence rate for the bias term.

Due to the non-convex nature of the quadratically parameterized model, our analysis is much more involved. The main challenge in our analysis is the diverse scaling of the ground truth signals and the anisotropic gradient noise caused by the diverse data eigenvalues. This requires us to provide individual bounds for the model parameters through the analysis and proposes a refined characterization for the learning process. This challenge does not exist in the traditional analysis in the quadratically parameterized model, since they consider near isotropic input data and $\Theta(1)$ ground truth (HaoChen et al., 2021). By constructing non-trivial couplings and employing truncated sequences, we provide a precise coordinate-wise analysis for the SGD dynamics, thereby overcoming this challenge.

We summarize the contribution of this paper as follows:

- The learning curves of SGD is proposed based on a quadratically parameterized model that emphasizes feature learning. We establish excess risk against sample and model sizes.

- A theoretical analysis for the dynamic of the quadratically parameterized model is offered, where we propose a new characterization to deal with the decaying ground truth and anisotropic gradient noise.

## 2 RELATED WORKS

**Linear Regression.** Linear regression, a cornerstone of statistical learning, achieves information-theoretic optimality $\widetilde{\mathcal{O}}\left(d\sigma^2/T\right)$ in finite dimensions for both offline and online settings (Bach & Moulines, 2013; Jain et al., 2018; Ge et al., 2019). Recent advances extend analyses to high-dimensional regimes under eigenvalue regularity conditions and parameter structure (Raskutti et al., 2014; Gunasekar et al., 2017; Bartlett et al., 2020; Hastie et al., 2022; Tsigler & Bartlett, 2023). Offline studies characterize implicit bias, benign overfitting, and multi-descent phenomena linked to spectral geometries (Liang et al., 2020; Ghorbani et al., 2021; Mei & Montanari, 2022; Lu et al., 2023; Zhang et al., 2024b), while online analyses reveal SGD's phased complexity release and covariance spectrum-dependent overfitting (Dieuleveut & Bach, 2016; Dieuleveut et al., 2017; Lin & Rosasco, 2017; Ali et al., 2020; Zou et al., 2021a;b; Wu et al., 2022; Varre et al., 2021). Recent work quantifies SGD's risk scaling under power-law spectral decays (Paquette et al., 2024; Lin et al., 2024; Bordelon et al., 2024; Bahri et al., 2024). We follow the geometric decay schedule of the step size (Ge et al., 2019; Wu et al., 2022; Zhang et al., 2024a) in Phase II due to its superiority in balancing rapid early-phase convergence and stable asymptotic refinement (Ge et al., 2019). However, in analysis of Phase II, we further require constructing auxiliary sequences to reach the desired convergence rate, which is much more technical.

**Feature Learning.** The feature learning ability of neural networks is the core mechanism behind their excellent generalization performance. In recent years, theoretical research has primarily focused on two directions: one is the analysis of infinitely wide networks within the mean-field framework, see e.g. Mei et al. (2018); Chizat & Bach (2018), and the other is the study of how networks align with low-dimensional objective functions including single-index models (Ba et al., 2022; Mousavi-Hosseini et al., 2022; Lee et al., 2024) and multi-index models (Damian et al., 2022; Vural & Erdogdu, 2024). Although significant progress has been made in these areas, the mean-field mode lacks a clear finite sample convergence rate. Assumptions such as sparse or low-dimensional isotropic objective functions weaken the generality and fail to recover the polynomial decay of generalization error with respect to sample size and model parameters. In this paper, we follow the previous quadratic parameterization (Vaskevicius et al., 2019; Woodworth et al., 2020; HaoChen et al., 2021) while develop a generalization error analysis under an anisotropic covariance structure, yielding generalization error results similar to those predicted by the neural scaling law.

## 3 SET UP

### 3.1 NOTATION

In this section, we introduce the following notations adopted throughout this work. Let $\mathcal{O}(\cdot)$ and $\Omega(\cdot)$ denote upper and lower bounds, respectively, with a universal constants, while $\widetilde{\mathcal{O}}(\cdot)$ and $\widetilde{\Omega}(\cdot)$ ignore polylogarithmic dependencies. For functions $f$ and $g$: $f \lesssim g$ denotes $f = \widetilde{\mathcal{O}}(g)$; $f \gtrsim g$ denotes $f = \widetilde{\Omega}(g)$; $f \asymp g$ indicates $g \lesssim f \lesssim g$. We denote $\mathbb{R}[\mathbf{z}]_{\leq k}$ as the vector space of polynomials with real coefficients in variables $\mathbf{z} = (\mathbf{z}_1, \cdots, \mathbf{z}_M)$, of degree at most $k$. For a positive integer $M$, let $[M]$ denote the set $\{1, \cdots, M\}$.

### 3.2 QUADRATICALLY PARAMETERIZED MODEL

We denote the covariate (feature) vector by $\mathbf{x} \in \mathbb{H}$, where $\mathbb{H}$ is a finite $d$-dimensional or countably infinite dimensional Hilbert space, and the corresponding response by $y \in \mathbb{R}$. Notice that the algorithm operates solely in finite-dimensional spaces. Following Lin et al. (2024), we assume access to $M$-dimensional sketched covariate vectors and their corresponding responses, denoted $(\mathbf{S}\mathbf{x}, y)$, where $\mathbf{S} \in \mathbb{R}^M \times \mathbb{H}$ is a fixed sketch matrix.

We focus on a quadratically parameterized model and measure the population risk of parameter $\mathbf{v}$ by the mean squared loss as:

$$\mathcal{R}_M(\mathbf{v}) = \mathbb{E}_{(\mathbf{x},y)\sim\mathcal{D}} \left(\langle \mathbf{S}\mathbf{x}, \mathbf{v}^{\odot 2} \rangle - y\right)^2, \tag{1}$$

where the expectation is taken over the joint distribution $\mathcal{D}$ of $(\mathbf{x}, y)$. In this paper, we study the quadratically parameterized model with the predictor $f_{\mathbf{v}}(\mathbf{x}) := \langle \mathbf{S}\mathbf{x}, \mathbf{v}^{\odot 2} \rangle$ for any $\mathbf{v} \in \mathbb{R}^M$. One

can generally use the parameterization as $\langle \mathbf{Sx}, \mathbf{v}_+^{\odot 2} - \mathbf{v}_-^{\odot 2} \rangle$ by the same technique as Woodworth et al. (2020). In contrast with linear model (Lin et al., 2024), quadratically parameterized model allows discovery of discriminative features through learning towards dominant directions of target. Thus, it models the feature learning mechanism while ensuring analytical tractability.

### 3.3 DATA DISTRIBUTION ASSUMPTIONS

We make the following assumptions of data distribution.

**Assumption 3.1** (Anisotropic Gaussian Data, Sub-Gaussian Noise, and Gaussian Remainder).

**[A$_1$]** (Independent Gaussian Data) For any $i \in [M]$, the sketched covariate $(\mathbf{Sx})_i \sim \mathcal{N}(0, \lambda_i)$. For any $i \neq j$, $(\mathbf{Sx})_i$ and $(\mathbf{Sx})_j$ are independent.

**[A$_2$]** (Sub-Gaussian Noise and Gaussian Remainder Term) There exist $\mathbf{v}^* \in \mathbb{R}^M$ and a sub-Gaussian random variable $\xi$ with parameter $\sigma_\xi > 0$ (see Definition E.1 for details) such that the remainder term $\zeta_M := y - \langle \mathbf{Sx}, \mathbf{v}^{*\odot 2} \rangle - \xi$ follows a normal distribution $\mathcal{N}(0, \sigma_{\zeta_M}^2)$. Moreover, $\mathbb{E}[\xi \zeta_M] = 0$. Additionally, for any polynomial $p(\mathbf{Sx}) \in \mathbb{R}[\mathbf{Sx}]_{\leq 3}$, we have $\mathbb{E}[p(\mathbf{Sx})\xi] = 0$ and $\mathbb{E}[p(\mathbf{Sx})\zeta_M] = 0$.

The assumption for independent Gaussian data is also used in other analyses for the quadratically parameterized model, such as HaoChen et al. (2021), whereas, we allow non-identical covariates. The independence assumption resembles (is slightly stronger than) the RIP condition, and is widely adopted in feature selection, e.g. Candes & Tao (2005), to ensure computational tractability, because in the worst case, finding sparse features is NP-hard (Natarajan, 1995). To mitigate the limitations associated with the independence assumption, we further introduce Assumption 3.2 and also establish a corresponding convergence guarantee (Theorem 4.2) for SGD on quadratically parameterized models under this assumption.

**Assumption 3.2** (General Gaussian Data, Sub-Gaussian Noise and Remainder).

**[A$_3$]** (General Gaussian Data) The sketched covariate vector $\mathbf{Sx} \sim \mathcal{N}(\mathbf{0}, \mathbf{A})$, where $\mathbf{A}$ is a positive semi-definite (PSD) matrix. The singular value decomposition (SVD) of $\mathbf{A}$ is given by $\mathbf{A} = \mathbf{Q_A} \cdot \text{diag}\{\lambda_i\}_{i \in [M]} \cdot \mathbf{Q_A}^\top$.

**[A$_4$]** (Sub-Gaussian Noise and Remainder Term) There exist $\mathbf{v}^* \in \mathbb{R}^M$ and a sub-Gaussian random variable $\xi$ with parameter $\sigma_\xi > 0$ such that the remainder term $\zeta_M := y - \langle \mathbf{Q_A}^\top \mathbf{Sx}, \mathbf{v}^{*\odot 2} \rangle - \xi$ is sub-Gaussian with parameter $\sigma_{\zeta_M} > 0$. Moreover, $\mathbb{E}[\xi \zeta_M] = 0$. Additionally, for any polynomial $p(\mathbf{Q_A}^\top \mathbf{Sx}) \in \mathbb{R}[\mathbf{Q_A}^\top \mathbf{Sx}]_{\leq 3}$, we have $\mathbb{E}\left[p(\mathbf{Q_A}^\top \mathbf{Sx})\xi\right] = 0$ and $\mathbb{E}\left[p(\mathbf{Q_A}^\top \mathbf{Sx})\zeta_M\right] = 0$.

Assumption 3.2 strictly generalizes Assumption 3.1 by allowing correlated Gaussian covariates with an arbitrary PSD covariance and by requiring only a sub-Gaussian remainder with low-degree orthogonality. Under this broader correlated-Gaussian assumption, the SGD convergence and feature-learning guarantees for diagonal-network predictors remain valid, and the diagonal independent case is recovered as a special instance. Our formulation aligns with the sketch method proposed by Lin et al. (2024). Furthermore, **[A$_3$]** in Assumption 3.2 holds for an arbitrary sketch matrix under the assumption that $\mathbf{x}$ follows a zero-mean Gaussian distribution.

We derive the scaling law for SGD under the following power-law decay assumptions of the covariance spectrum and prior conditions.

**Assumption 3.3** (Specific Spectral Assumptions).

**[A$_5$]** (Polynomial Decay Eigenvalues) There exists $\alpha > 1$ such that for any $i \in [M]$, the eigenvalue of data covariance $\lambda_i$ satisfy $\lambda_i \asymp i^{-\alpha}$.

**[A$_6$]** (Source Condition) There exists $\beta > 1$ such that the ground truth parameter $\mathbf{v}^*$ satisfies that for any $i \in [M]$, $\lambda_i \left(\mathbf{v}_i^*\right)^4 \asymp i^{-\beta}$. Moreover, $\sigma_{\zeta_M}^2 = \sum_{i>M} i^{-\beta}$.

The polynomial decay of eigenvalues and the ground truth has been widely considered to study the scaling laws for linear models like random feature model (Bahri et al., 2024; Bordelon et al., 2024; Paquette et al., 2024) and infinite dimensional linear regression (Lin et al., 2024), based on empirical observations of NTK spectral decompositions on the realistic dataset (Bahri et al., 2024; Bordelon &

---

**Algorithm 1** Stochastic Gradient Descent (SGD)

---

**Input:** Initial weight $\mathbf{v}_0 = \Omega(\min\{1, M^{-(\beta-\alpha)/4}\})\mathbf{1}_M$, initial step-size $\eta$, total sample size $T$, middle phase length $h$, decaying phase length $T_1 = \lfloor (T-h)/\log(T-h) \rfloor$.
**while** $t \leq T$ **do**
   **if** $t > h$ and $(t-h) \bmod T_1 = 0$ **then**
      $\eta \leftarrow \eta/2$.
   **end if**
   Sample a fresh data $(\mathbf{x}^{t+1}, y^{t+1}) \sim \mathcal{D}$.
   $\mathbf{v}^{t+1} \leftarrow \mathbf{v}^t - \frac{\eta}{2}\nabla_{\mathbf{v}} \left( f_{\mathbf{v}^t}(\mathbf{x}^{t+1}) - y^{t+1} \right)^2$.
**end while**

---

Pehlevan, 2021). It is used in slope functional regression (Cai & Hall, 2006), and also analogous to the capacity and source conditions in RKHS (Wainwright, 2019; Bietti & Mairal, 2019). Given that the optimization trajectory of linear models is intrinsically aligned with the principal directions of the covariate feature space, this alignment motivates us to adopt analogous assumptions for our model, thereby enabling direct comparison of learning dynamics through feature space decomposition.

### 3.4 ALGORITHM

We employ SGD with a geometric decay of step size to train the quadratically parameterized predictor $f_{\mathbf{v}}$ to minimize the objective equation 1. Starting at $\mathbf{v}^0$, the iteration of parameter vector $\mathbf{v} \in \mathbb{R}^M$ can be represented explicitly as follows:

$$\begin{aligned}
\mathbf{v}^t &= \mathbf{v}^{t-1} - \eta_t \left( f_{\mathbf{v}^{t-1}}\left(\mathbf{x}^t\right) - y^t \right)\left(\mathbf{v}^{t-1} \odot \mathbf{S}\mathbf{x}^t\right) \\
&= \mathbf{v}^{t-1} - \eta_t \left( \left\langle \mathbf{S}\mathbf{x}^t, \left(\mathbf{v}^{t-1}\right)^{\odot 2} \right\rangle - y^t \right)\left(\mathbf{v}^{t-1} \odot \mathbf{S}\mathbf{x}^t\right),
\end{aligned}$$

for $t = 1, \ldots, T$, where $\{(\mathbf{x}^t, y^t)\}_{t=1}^T$ are independent samples from distribution $\mathcal{D}$ and $\{\eta_t\}_{t=1}^T$ are the step sizes.

We use the tail geometric decay of step size schedule as describe in Wu et al. (2022). The step size remains constant for the first $T_1 + h$ iterations where $h$ denotes the middle phase length and $T_1 := \lfloor (T-h)/\log(T-h) \rfloor$. Then the step size halves every $T_1$ steps. Specifically, the decay schedule of step size is given by:

$$\eta_t = \begin{cases} \eta, & 0 \leq t \leq T_1 + h, \\ \eta/2^l, & T_1 + h < t \leq T, \ l = \lfloor (t-h)/T_1 \rfloor, \end{cases}$$

The integration of warm-up with subsequent learning rate decay has become a prevalent technique in deep learning optimization (Goyal, 2017). Within the decay stage, geometric decay schedules have demonstrated superior empirical efficiency compared to polynomial alternatives, as geometric decay achieves adaptively balancing aggressive early-stage learning with stable late-stage refinement (Ge et al., 2019). Motivated by these established advantages, our step size schedule design strategically combines an initial constant stage with a subsequent geometrically decaying stage. This hybrid approach inherits the computational benefits of geometric decay while maintaining the stability benefits of warm-up initialization, creating synergistic effects that polynomial decay schedules cannot achieve (Bubeck et al., 2015).

The algorithm is summarized as Algorithm 1. The initial point $\mathbf{v}_0$ and the initial step size $\eta$ are hyperparameters of Algorithm 1, and they play a crucial role in determining whether the algorithm can escape saddle points and converge to the optimal solution. Starting at an initial point near zero, the constant step size stage allows the algorithm to adaptively extract the important features without explicitly setting the truncation dimensions while keeping the remaining variables close to zero. The subsequent geometric decay of the step size guarantees fast convergence to the ground truth.

## 4 CONVERGENCE ANALYSIS

The upper bound of last iterate instantaneous risk for Algorithm 1 can be summarized by the following theorem, which provides the guarantee of global convergence for last iterate SGD with tail geometrically decaying stepsize and a sufficiently small initialization.

**Theorem 4.1.** *Under Assumptions 3.1 and 3.3, we consider a predictor trained by Algorithm 1 with total sample size $T$ and middle phase length $h = \lceil T/\log(T) \rceil$. Let $D \asymp \min\{T^{1/\max\{\beta,(\alpha+\beta)/2\}}, M\}$ and $\eta \asymp D^{\min\{0,(\alpha-\beta)/2\}}$. The error of output can be bounded from above by*

$$\mathcal{R}_M(\mathbf{v}^T) - \mathbb{E}[\xi^2] \asymp \underbrace{\frac{1}{M^{\beta-1}}}_{\text{approximation}} + \underbrace{\frac{\sigma^2 D}{T}}_{\text{variance}} + \underbrace{\frac{D}{T} + \frac{1}{D^{\beta-1}}\mathbb{1}_{D<M}}_{\text{bias}}, \tag{2}$$

*with probability at least 0.95, where $\sigma^2 := \sigma_\xi^2 + \sigma_{\zeta_M}^2$.*

Our bound exhibits two key properties: (1) Dimension-free: equation 3 depends on the effective dimension $D$ rather than ambient dimension $M$. (2) Problem-adaptive: $D$ is governed by the spectral structure of $\text{diag}\{\lambda_1(\mathbf{v}_1^*)^2, \cdots, \lambda_M(\mathbf{v}_M^*)^2\}$, which is induced by the multiplicative coupling between the data covariance matrix and optimal solution determined by the problem. The risk bound in equation 3 consists of three components: (1) approximation error term, (2) bias error term originating from $\mathbf{v}^{T_1} - \mathbf{v}_{1:M}^*$ at iteration $T_1 = \lceil (T-h)/\log(T-h) \rceil$, and (3) variance error term stemming from the multiplicative coupling between additive noise $\xi + \sum_{i \geq M+1} \mathbf{x}_i(\mathbf{v}_i^*)^2$ and matrix $\text{diag}\{\mathbf{v}_{1:M}^*\}$. The step size configuration in Theorem 4.1 is strategically designed to achieve faster convergence.

For larger $M$, Corollary 4.1 establishes the convergence rate for Algorithm 1 via Theorem 4.1.

**Corollary 4.1.** *Under the setting of the parameters in Theorem 4.1, if $T^{1/\max\{\beta,(\alpha+\beta)/2\}} \asymp D < M$, we have*

$$\begin{cases} \mathcal{R}_M(\mathbf{v}^T) - \mathbb{E}[\xi^2] \asymp \frac{1}{M^{\beta-1}} + \frac{\sigma^2+1}{T^{1-1/\beta}}, & \text{if } \beta \geq \alpha > 1, \\ \mathcal{R}_M(\mathbf{v}^T) - \mathbb{E}[\xi^2] \asymp \frac{1}{M^{\beta-1}} + \frac{\sigma^2+1}{T^{(2\beta-2)/(\alpha+\beta)}}, & \text{if } \alpha > \beta > 1, \end{cases}$$

*with probability at least 0.95.*

Corollary 4.1 demonstrates that under Assumptions 3.1 and 3.3, when the model size $M$ is sufficiently large, the last iterate instantaneous risk of Algorithm 1 exhibits distinct behaviors in two regimes: (I) $\beta \geq \alpha > 1$ and (II) $\alpha \geq \beta > 1$. We consider the total computational budget as $B = MT$, reflecting that Algorithm 1 queries $M$-dimensional gradients $T$ times.

**Given $B$:** If $\beta > \alpha > 1$, the optimal last iterate risk is attained with parameter configurations: $M = \widetilde{\Omega}(B^{\frac{1}{1+\beta}})$ and $T = \widetilde{\Omega}(B^{\frac{\beta}{1+\beta}})$. If $\alpha \geq \beta > 1$, the optimal last iterate risk is attained with parameter configurations: $M = \widetilde{\Omega}(B^{\frac{1}{1+(\alpha+\beta)/2}})$ and $T = \widetilde{\Omega}(B^{\frac{(\alpha+\beta)/2}{1+(\alpha+\beta)/2}})$.

**Given Total Sample Size $T$:** So as long as $M \gtrsim T^{1/\max\{\beta,(\alpha+\beta)/2\}}$, Corollary 4.1 implicates that the risk can be effectively reduced by increasing the model size $M$ as much as possible.

For smaller $M$, Corollary 4.2 provides the convergence rate for Algorithm 1 through Theorem 4.1.

**Corollary 4.2.** *Under the setting of the parameters in Theorem 4.1, if $M \lesssim T^{1/\max\{\beta,(\alpha+\beta)/2\}}$, we have*

$$\mathcal{R}_M(\mathbf{v}^T) - \mathbb{E}[\xi^2] \asymp \frac{1}{M^{\beta-1}} + \frac{(\sigma^2+1)M}{T},$$

*with probability at least 0.95.*

The risk bound $\mathcal{R}_M(\cdot)$ in Corollary 4.2 decreases monotonically with increasing $M$. So as long as $M \lesssim T^{1/\max\{\beta,(\alpha+\beta)/2\}}$, our analysis implies to increase the model size $M$ until reaching the computational budget.

*Remark* 4.1. For any (random) algorithm $\hat{\mathbf{v}}$ based on i.i.d. data $\{(\mathbf{x}_i, y_i)\}_{i=1}^T$ from the true parameter $\mathbf{v}_* \in \mathcal{V}$, the worst-case excess risk convergence rate is limited by the information-theoretic lower bound. The scaling law, however, describes the excess risk trajectory of a specific algorithm in a given context during training. Under the covariate distribution assumptions 3.1, and the ground truth assumption 3.3, prior work (Zhang et al., 2024a) established the info-theoretic lower bound as $T^{-\frac{1}{\beta}}$. Our analysis shows two distinct regimes: When $\alpha \leq \beta$, SGD in linear and quadratically parameterized models hits the lower bound, proving statistical optimality. When $\alpha > \beta$, SGD in both misses the bound, yet the quadratically parameterized model has better excess risk than the linear one. This shows a capacity gap between the two model types, highlighting the importance of feature learning and model adaptation.

When the covariance matrix of $\mathbf{Sx}$ is a general PSD matrix $\mathbf{A}$, we first need to obtain an estimate of $\mathbf{A}$ given by $\widetilde{\mathbf{U}}\widetilde{\mathbf{U}}^\top$, with $\widetilde{\mathbf{U}} \in \mathbb{R}^{M \times M}$. Then, based on the SVD of $\widetilde{\mathbf{U}} = \mathbf{Q}_{\widetilde{\mathbf{U}}} \cdot \operatorname{diag}\{\gamma_i\}_{i \in [M]} \cdot \mathbf{P}_{\widetilde{\mathbf{U}}}^\top$, the form of the predictor $f_{\mathbf{v}}(\mathbf{x})$ in Algorithm 1 is modified to:

$$f_{\mathbf{v}}(\mathbf{x}) = \left\langle \mathbf{Q}_{\widetilde{\mathbf{U}}}^\top \mathbf{Sx}, \mathbf{v}^{\odot 2} \right\rangle.$$

To establish convergence of SGD under this setting, we assume that the estimator $\widetilde{\mathbf{U}}$ satisfies the following accuracy condition:

**Assumption 4.1.** Defining $\mathbf{U_A} := \mathbf{Q_A}\mathbf{\Sigma}^{1/2}$ where $\mathbf{\Sigma} := \operatorname{diag}\{\lambda_i\}_{i \in [M]}$, then the following inequalities hold

$$\left\|\widetilde{\mathbf{U}}\widetilde{\mathbf{U}}^\top - \mathbf{A}\right\| \leq \lambda_M^2 \min\left\{\frac{1}{D}, \frac{D^{\max\{\beta, (\alpha+\beta)/2\} - 1}}{T}\right\},$$

$$\min_{\substack{\mathbf{R} \in \mathbb{R}^{M \times M} \\ \mathbf{RR}^\top = \mathbf{I}_M}} \left\|\widetilde{\mathbf{U}}\mathbf{R} - \mathbf{U_A}\right\| \leq \lambda_M \cdot \min\left\{M^{-\max\left\{0, \frac{\beta-\alpha}{2}\right\}}, \frac{1}{D}, \frac{D^{\max\left\{\beta, \frac{\alpha+\beta}{2}\right\} - 1}}{T}\right\}.$$

For a PSD matrix $\mathbf{A}$, numerous existing works (Stöger & Soltanolkotabi, 2021; Zhuo et al., 2024; Zhang et al., 2021; 2023; Xiong et al., 2023; Li et al., 2018; Tu et al., 2016) design algorithms using the parametrization $\mathbf{U}\mathbf{U}^\top$ with $\mathbf{U} \in \mathbb{R}^{M \times M}$ to achieve convergence in $\left\|\mathbf{U}\mathbf{U}^\top - \mathbf{A}\right\|$ or $\operatorname{dist}(\mathbf{U}, \mathbf{U_A})$ which is defined as follows:

$$\operatorname{dist}(\mathbf{U}, \mathbf{U_A}) := \min_{\substack{\mathbf{R} \in \mathbb{R}^{M \times M} \\ \mathbf{RR}^\top = \mathbf{I}_M}} \left\|\mathbf{U}\mathbf{R} - \mathbf{U_A}\right\|.$$

In our setting, we have access to random matrices $\mathbf{Sx}(\mathbf{Sx})^\top$, where $\mathbb{E}[\mathbf{Sx}(\mathbf{Sx})^\top] = \mathbf{A}$. Compared to the deterministic matrix factorization problem (Stöger & Soltanolkotabi, 2021; Zhuo et al., 2024; Zhang et al., 2021; 2023), this only introduces an additional zero-mean random noise. Consequently, by appropriately modifying existing algorithms for stochastic matrix factorization (e.g., those in Xiong et al. (2023); Li et al. (2018); Tu et al. (2016)), we can technically obtain an estimator $\widetilde{\mathbf{U}}$ that satisfies Assumption 4.1.

**Theorem 4.2.** *Under Assumptions 3.2, 3.3 and 4.1, we consider a predictor trained by Algorithm 1 with total sample size $T$ and middle phase length $h = \lceil T/\log(T) \rceil$. Let $D \asymp \min\{T^{1/\max\{\beta,(\alpha+\beta)/2\}}, M\}$ and $\eta \asymp D^{\min\{0,(\alpha-\beta)/2\}}$. The error of output can be bounded from above by*

$$\mathcal{R}_M(\mathbf{v}^T) - \mathbb{E}[\xi^2] \lesssim \underbrace{\frac{1}{M^{\beta-1}}}_{\text{approximation}} + \underbrace{\frac{\sigma^2 D}{T}}_{\text{variance}} + \underbrace{\frac{D}{T} + \frac{1}{D^{\beta-1}}\mathbb{1}_{D<M}}_{\text{bias}}$$

$$+ \left\|\widetilde{\mathbf{U}}\widetilde{\mathbf{U}}^\top - \mathbf{A}\right\| + \operatorname{dist}\left(\widetilde{\mathbf{U}}, \mathbf{U_A}\right), \tag{3}$$

*with probability at least 0.95, where $\sigma^2 := \sigma_\xi^2 + \sigma_{\zeta_M}^2$.*

The proof of Theorem 4.2 follows a similar line of reasoning and technique as that of Theorem 4.1. This is because, after applying $\mathbf{Q}_{\widetilde{\mathbf{U}}}$ to the sketched covariate vector, the covariance matrix of $\mathbf{Q}_{\widetilde{\mathbf{U}}}^\top \mathbf{Sx}$ does not deviate significantly from a diagonal matrix. In fact, the difference between the dynamics

of the parameter $\mathbf{v}$ in Theorem 4.1 and those in Theorem 4.2 can be controlled by the distance metric $\text{dist}(\widetilde{\mathbf{U}}, \mathbf{U_A})$ and the norm $\|\widetilde{\mathbf{U}}\widetilde{\mathbf{U}}^\top - \mathbf{A}\|$, as detailed in section D. Since Theorem 4.1 is the core result of this paper and the proof of Theorem 4.2 does not differ substantially from that of Theorem 4.1, we provide only a proof sketch for Theorem 4.1 in the next section.

# 5 PROOF SKETCH OF THEOREM 4.1

In this section, we introduce the proof techniques sketch of our main result Theorem 4.1, while a more detailed version is available in section A. The dynamics and analysis of SGD can be divided into two phases. In **Phase I (Adaptation)**, SGD autonomously truncates the top $D$ coordinates as $\mathcal{S}$ (i.e. $\mathcal{S} := [D]$) without requiring explicit selection of $D$. Algorithm 1 can converge these coordinates to a neighborhood of their optimal solutions within $T_1$ iterations with high probability. The core theorem in this phase is Theorem 5.1:

**Theorem 5.1.** *Under Assumption 3.1, consider a predictor trained via Algorithm 1 with initialization $\mathbf{v}^0$. Let the step size $\eta \leq \eta(D, c_1)$, for the effective dimension $D$ and the scaling constant $c_1 \in (0, 1)$. The iteration number $T_1$ requires:*

$$T_1 \in \begin{cases} [T_l(D, c_1), T_u(D, c_1)], & \text{if } D < M, \\ [T_l(M, c_1), \infty), & \text{otherwise.} \end{cases}$$

*Then, with high probability, we have*

$$\begin{cases} \mathbf{v}_i^{T_1} \in [(1 - c_1)\mathbf{v}_i^*, (1 + c_1)\mathbf{v}_i^*], & \text{if } i \in \mathcal{S}, \\ \mathbf{v}_i^{T_1} \in \left[0, \frac{3}{2}\mathbf{v}_i^*\right], & \text{otherwise .} \end{cases} \tag{4}$$

In **Phase II (Estimation)**, global convergence to the risk minimizer is achieved over $T_2 := T - T_1$ iterations, which can be approximated as SGD with geometrically decaying step sizes applied to a linear regression problem in the reparameterized feature space $\mathbf{Sx} \odot \mathbf{v}^*$. This implies that for each coordinate $i \in \mathcal{S}$, the step size in Algorithm 1 is scaled by a certain magnitude of $\mathbf{v}_i^*$. The core theorem in this phase is Theorem 5.2:

**Theorem 5.2.** *Suppose Assumptions 3.1 and 3.3 hold. By selecting an appropriate step size $\eta_0 = \eta(D)$ and middle phase length $h$, we obtain*

$$\mathcal{R}_M(\mathbf{v}^T) \lesssim \mathcal{R}_M(\mathbf{v}^*) + \frac{\sigma^2 D}{T} + \sigma^2 \eta_0^2 T \operatorname{tr}\left(\mathbf{H}_{D+1:M}^2\right) + \frac{D}{T} + \eta_0^2 T \operatorname{tr}\left(\mathbf{H}_{D+1:M}^2\right)$$

$$+ \left\langle \frac{1}{\eta_0 T}\mathbf{I}_{1:D} + \mathbf{H}_{D+1:M}, \left(\mathbf{I} - \eta_0 \widehat{\mathbf{H}}\right)^{\frac{2T}{\log(T)}} \mathbf{B}^0 \right\rangle,$$

*with probability at least 0.95, where*

$$\mathbf{H} := \operatorname{diag}\{\lambda_i(\mathbf{v}_i^*)^2\}_{i=1}^M \quad and \quad \widehat{\mathbf{H}} := \operatorname{diag}\{\lambda_1(\mathbf{v}_1^*)^2, \cdots, \lambda_N(\mathbf{v}_N^*)^2, \mathbf{0}_{\mathbf{M}-\mathbf{D}}\}.$$

For the *lower bound* (see Appendix C), our analysis reveals that for coordinates $j \geq \widetilde{\mathcal{O}}(D)$, the slow ascent rate inherently prevents $\mathbf{v}_j^t$ from approaching the optimal solution $\mathbf{v}_j^*$ upon algorithm termination. This phenomenon induces bias error's scaling as $\widetilde{\Omega}(D^{-\beta+1})$, matching our upper bound characterization, up to logarithmic factors.

# 6 CONCLUSIONS

In this paper, we construct the theoretical analysis for the dynamic of quadratically parameterized model under decaying ground truth and anisotropic gradient noise. Our technique is based on the precise analysis of two-stage dynamic of SGD, with adaptive selection of the effective dimension set in the first stage and the approximation of linear model in the second stage. Our analysis characterizes the feature learning and model adaptation ability with clear separations for convergence rates in the canonical linear model.

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

CONTENTS

## A  DETAILED PROOF SKETCH OF THEOREM 4.1

### A.1  PHASE I: ADAPTATION

During the "adaptation" phase, Algorithm 1 implicitly identifies the first $D$ coordinates as the effective dimension set $\mathcal{S} := [D]$. For each $i \in \mathcal{S}$, $\mathbf{v}_i^{T_1}$ converges with high probability to a rectangular neighborhood centered at $\mathbf{v}_i^*$ with half-width $c_1 \mathbf{v}_i^*$. Here, $c_1 \in (0, 1)$ denotes a scaling constant. For each $i \in \mathcal{S}^c := [M] \setminus \mathcal{S}$, $\mathbf{v}_i^{T_1}$ remains bounded above by $\frac{3}{2}\mathbf{v}_i^*$ with high probability.

To characterize the mainstream dynamic, our analysis employs a probabilistic sequence synchronization technique. That is, from the sequence $\{\mathbf{v}^t\}_{t=0}^{T_1}$ generated by Algorithm 1, we construct a control sequence $\{\mathbf{q}^t\}_{t=0}^{T_1}$ to rule out some low-probabilistic unbounded trajectories in $\{\mathbf{v}^t\}_{t=0}^{T_1}$. We first establish Lemmas A.1–A.3 for the control sequence.

In the analysis of **Phase I**, we need to delve into the dynamic processes of the two-part parameters separated by the effective dimension $D$. It is non-trivial because in the traditional analysis of prior work to recover the sparse ground truth (HaoChen et al., 2021), it is unnecessary to introduce $D$. In Lemma A.1, utilizing a constructive supermartingale, we formally characterize the one-step iterative behavior of $\mathbf{q}_i^t$ when $\mathbf{q}_i^t > \mathbf{v}_i^*$, which approximately satisfies: $\mathbf{q}_i^{t+1} - \mathbf{v}_i^* \lesssim (1 - \eta\mathcal{O}(\lambda_i(\mathbf{v}_i^*)^2))(\mathbf{q}_i^t - \mathbf{v}_i^*)$. Then we show the last iterate $\mathbf{q}^{T_1}$ satisfies a high-probability upper bound, matching the bound in Theorem 5.1.

**Lemma A.1.** *Under the setting of Theorem 5.1, both $\mathbf{q}_i^{T_1} \leq (1+c_1)\mathbf{v}_i^*$ for any $i \in \mathcal{S}$ and $\mathbf{q}_i^{T_1} \leq \frac{3}{2}\mathbf{v}_i^*$ for any $i \in \mathcal{S}^c$ occur with high probability.*

Lemmas A.2 and A.3 collectively address the lower bound of $\mathbf{q}^{T_1}$ in Theorem 5.1. To establish Lemma A.2, for any $i \in \mathcal{S}$, we construct a submartingale to formally analyze the one-step iterative behavior of $\mathbf{q}_i^t$ when $\mathbf{q}_i^t < (1 - c_1/2)\mathbf{v}_i^*$, which approximately satisfies: $\mathbf{q}_i^{t+1} \gtrsim (1 + \eta c_1 \mathcal{O}(\lambda_i(\mathbf{v}_i^*)^2))\mathbf{q}_i^t$. According to the concentration inequalities, we obtain the following conclusion.

**Lemma A.2.** *Under the setting of Theorem 5.1, with high probability, either $\max_{t \leq T_1} \mathbf{q}_i^t \geq (1 - c_1/2)\mathbf{v}_i^*$ for any $i \in \mathcal{S}$, or at least one of the following statements fails: $\mathbf{q}_i^{T_1} \leq (1 + c_1)\mathbf{v}_i^*$ for any $i \in \mathcal{S}$ and $\mathbf{q}_i^{T_1} \leq \frac{3}{2}\mathbf{v}_i^*$ for any $i \in \mathcal{S}^c$.*

Lemma A.3 establishes the lower bound for $\mathbf{q}_i^{T_1}$ ($i \in \mathcal{S}$). The proof mirrors that of Lemma A.1.

**Lemma A.3.** *Under the setting of Theorem 5.1, for any $i \in \mathcal{S}$, with high probability, either $\max_{t \leq T_1} \mathbf{q}_i^t < (1 - c_1/2)\mathbf{v}_i^*$ or $\mathbf{q}_i^{T_1} \geq (1 - c_1)\mathbf{v}_i^*$.*

According to the high-probability equivalence between $\{\mathbf{q}^t\}_{t=0}^{T_1}$ and $\{\mathbf{v}^t\}_{t=0}^{T_1}$, Lemmas A.1–A.3's conclusions transfer to $\mathbf{v}^{T_1}$ with high-probability guarantees. Therefore, we obtain Theorem 5.1.

### A.2  PHASE II: ESTIMATION

We now start the analysis of **Phase II** for Algorithm 1. The main idea stems from approximating Algorithm 1's iterations as SGD running over a linear model with rescaled features $\mathbf{Sx} \odot \mathbf{v}^*$. The adaptive rescale size $\mathbf{v}^*$ enables the quadratic model to achieve accelerated convergence rates compared to its linear counterpart.

The proof of Theorem 5.2 is structured in two key parts. In **Part I**, Theorem A.1 establishes that Algorithm 1 iterates remain within an uniform neighborhood of $\mathbf{v}^*$ (equation 5) with high probability.

**Theorem A.1.** *Under Assumption 3.1, we consider the iterative process of Algorithm 1, beginning from step $T_1$ with the same step size $\eta$ as in Theorem 5.1. If $D < M$, let $1 \leq T_2 \leq T_u(D)$ where $T_u(D) \in \mathbb{N}_+$ depends on $D$; otherwise, let $T_2 \geq 1$. Then, with high probability, we have*

$$\begin{cases} \mathbf{v}_i^{T_1+t} \in \left[\frac{1}{2}\mathbf{v}_i^*, \frac{3}{2}\mathbf{v}_i^*\right], & \text{if } i \in [D], \\ \mathbf{v}_i^{T_1+t} \in [0, 2\mathbf{v}_i^*], & \text{otherwise}, \end{cases} \quad \forall t \in [T_2]. \tag{5}$$

Let $c_1 = \frac{1}{4}$. According to Theorem 5.1, $\mathbf{v}^{T_1}$ satisfies equation 4 with high probability. By employing the same construction method as that in Lemmas A.1 and A.1, we derive a family of compressed

supermartingales to characterize the dynamics of $\{\mathbf{q}_t\}_{t=T_1}^T$. Combining the supermartingales concentration inequality, we obtain equation 5.

In **Part II**, we construct an auxiliary bounded sequence $\{\mathbf{w}^t\}_{t=1}^{T_2}$ which is the truncation of $\{\mathbf{v}^{T_1+t}\}_{t=1}^{T_2}$. The novelty and ingenuity of our analysis based on auxiliary sequence construction lie in the alignment of $\{\mathbf{w}^t\}_{t=1}^{T_2}$ and $\{\mathbf{v}^{T_1+t}\}_{t=1}^{T_2}$ as $\mathbf{w}^{T_2} = \mathbf{v}^T$ with high probability by Theorem A.1. Thus our proposed the last iterate risk for $\mathbf{w}^{T_2}$ can be extended to $\mathbf{v}^T$. Specifically, the update rule of $\mathbf{w}^t$ satisfies the following formula with high probability:

$$\mathbf{w}^{t+1} = \mathbf{w}^t - \eta_t \mathbf{H}^t(\mathbf{w}^t - \mathbf{v}^*) + \eta_t \mathbf{R}^t \mathbf{S} \mathbf{x}^t, \tag{6}$$

where $\mathbf{H}^t \in \mathbb{R}^{M \times M}$ depends on $\mathbf{w}^t$ and $\mathbf{x}^t$, and $\mathbf{R}^t \in \mathbb{R}^{M \times M}$ depends on $\mathbf{w}^t, \zeta_M^t$ and $\xi^t$. Combining equation 6 with the constraint of $\{\mathbf{w}^t\}_{t=1}^{T_2}$, we observe that the update process of $\mathbf{w}^t$ approximates that of SGD in traditional linear regression problems (Wu et al., 2022) with reparameterized features $\mathbf{S}\mathbf{x} \odot \mathbf{v}^*$. The SGD iteration in linear model exhibits structural similarity to equation 6, but differs in that its $\mathbf{H}^t$ and $\mathbf{R}^t$ are independent on iterative variables; this independence eliminates the need for truncated sequences in analytical treatments. Our analysis innovatively introduces the truncated sequence $\{\mathbf{w}^t\}_{t=1}^{T_2}$ to maintain analytical tractability of $\mathbf{H}^t$ and $\mathbf{R}^t$. According to equation 6, we decompose the risk $\mathcal{R}_M(\mathbf{w}^{T_2})$ as follows:

$$\mathbb{E}\left[\mathcal{R}_M(\mathbf{w}^{T_2})\right] - \mathcal{R}_M(\mathbf{v}^*) \lesssim \underbrace{\langle \mathbf{H}, \mathbf{B}^{T_2} \rangle}_{\text{bias error}} + \underbrace{\langle \mathbf{H}, \mathbf{V}^{T_2} \rangle}_{\text{variance error}}. \tag{7}$$

For any $t \in [T_2]$, $\mathbf{B}^t$ and $\mathbf{V}^t$ are $M \times M$ matrices, derived from the bias and variance terms induced by $\mathbf{w}^t - \mathbf{v}^*$, respectively. Since $\mathbf{H}^t$ and $\mathbf{R}^t$ in equation 6 are both dependent on $\mathbf{w}^t$, it is a challenge to directly establish the full-matrix recursion between $\mathbf{V}^{t+1}$ and $\mathbf{V}^t$ (or $\mathbf{B}^{t+1}$ and $\mathbf{B}^t$) under the SGD iteration process like the similar techniques in linear models (Wu et al., 2022). To resolve this challenge, we novelly consider the recursive relations between diagonal elements of $\{\mathbf{V}^t\}_{t=0}^{T_2}$ and $\{\mathbf{B}^t\}_{t=0}^{T_2}$ across discrete time steps, thereby obtaining the estimation for both variance and bias errors for our linear approximation.

# B   PROOFS OF UPPER BOUND (THEOREM 4.1)

In this section, we introduce our proof techniques to prove our main result Theorem B.4 on the upper bound of the last-iteration instantaneous risk of Algorithm 1. As shown in Section 5, the dynamic of SGD and our analysis can be basically divided into two phases. In the **Phase I** named "adaption" phase, we demonstrate that SGD can adaptively identify the first $D$ coordinates as the optimal set $\mathcal{S}$ without explicit selection of $D$, and bound such $D$ coordinates near the corresponding optimal solutions by $T_1$ iterations with high probability (refer to Theorem B.1). The analysis of **Phase I** can be further separated into two parts:

1. We construct a high-probability upper bound of $\mathbf{v}^{T_1}$. That is for any $i \in \mathcal{S}$, $\mathbf{v}_i^{T_1} \leq (1 + c_1)\mathbf{v}_i^*$ and for any $i \in \mathcal{S}^c$, $\mathbf{v}_i^{T_1} \leq \frac{3}{2}\mathbf{v}_i^*$ (refer to Lemma B.1).

2. We delve into the lower bound of $\max_{t \leq T_1} \mathbf{v}_i^t$ during $T_1$ iterations. With high probability, for any $i \in \mathcal{S}$, $\max_{t \leq T_1} \mathbf{v}_i^t$ converges to a neighborhood of $\mathbf{v}_i^*$ (refer to Lemma B.2). When $\max_{t \leq T_1} \mathbf{v}_i^t$ resides within the $\mathbf{v}_i^*$-neighborhood, the lower bound satisfies $\mathbf{v}_i^{T_1} \geq (1 - c_1)\mathbf{v}_i^*$ with high probability (refer to Lemma B.3).

Then we turn to the following **Phase II** with $T_2$ iterations named "estimation" phase where we establish the global convergence of Algorithm 1 for risk minimization (refer to Theorem B.2). The analysis of Algorithm 1's iterations can be approximated to SGD with geometrically decaying step sizes on a linear regression problem with reparameterized features $\mathbf{S}\mathbf{x} \odot \mathbf{v}^*$. It can also be separated into two parts:

1. We demonstrate that $\{\mathbf{v}^t\}_{t=T_1+1}^{T_1+T_2}$ remain confined within the neighborhood $\prod_{i=1}^D [\frac{1}{2}\mathbf{v}_i^*, \frac{3}{2}\mathbf{v}_i^*] \times \prod_{i=D+1}^M [0, 2\mathbf{v}_i^*]$ with high probability (refer to Lemma B.3).

2. We construct an auxiliary sequence $\{\mathbf{w}^t\}_{t=1}^{T_2}$ aligned to $\{\mathbf{v}^{T_1+t}\}_{t=1}^{T_2}$ with high probability. We approximate the update process of $\{\mathbf{w}^t\}_{t=1}^{T_2}$ to SGD in traditional linear regression, with separated bounds of variance term (refer to Lemma B.5) and bias term (refer to Lemma B.12).

We propose our proof process step by step according to the above sketch. First, for clarity, we formally define some of the notations to use. We let bold lowercase letters, for example, $\mathbf{x} \in \mathbb{R}^d$, denote vectors, and bold uppercase letters, for example, $\mathbf{A} \in \mathbb{R}^{m \times n}$, denote matrices. We apply scalar operators to vectors as the coordinate-wise operators of vectors. For vector $\mathbf{x} \in \mathbb{R}^d$, denote $|\mathbf{x}| \in \mathbb{R}^d$ with $|\mathbf{x}|_j = |\mathbf{x}_j|$. For two vectors $\mathbf{x}, \mathbf{y} \in \mathbb{R}^d$, denote $\mathbf{x} \le \mathbf{y}$, if for all $j \in [d]$, $\mathbf{x}_j \le \mathbf{y}_j$. Additionally, we use $\langle \mathbf{x}, \mathbf{y} \rangle_{-i}$ to denote $\sum_{\substack{j=1 \\ j \ne i}}^d \mathbf{x}_j \mathbf{y}_j$. For a sequence of real numbers $\{v^t\}_{t=t_1}^{t_2}$ and $a, b \in \mathbb{R}$ with $a \le b$, denote $v^{t_1:t_2} \in [a, b]$ to represent that $v^t \in [a, b]$ for all $t \in [t_1, t_2]$.

Considering Assumption 3.1, the random variable $\mathbf{Sx} \in \mathbb{R}^M$ satisfies the sub-Gaussian condition with parameter $\lambda_i^{1/2}$ for all $i \in [M]$, and the noise $\xi$ is zero-mean sub-Gaussian with parameter $\sigma_\xi$. For any $D \in \mathbb{N}_+$, for simplification, we define

$$\sigma_{\min}(D) := \min_{j \in [D]} \lambda_j (\mathbf{v}_j^*)^2, \quad \bar{\sigma}_{\min}(D) := \min_{j \in [D]} (\mathbf{v}_j^*)^2,$$

$$\hat{\sigma}_{\max}(D) := \max_{j \in [D]} \log^{-1}(\mathbf{v}_j^0), \quad \tilde{\sigma}_{\max}(D) := \max_{j \in [D+1:M]} \lambda_j.$$

We also denote the matrix $\text{diag}\{\lambda_1, \ldots, \lambda_M\}$ as $\Lambda_{1:M}$. For $\mathbf{b} \in \mathbb{R}_+^M$, we define $\mathcal{M}(\mathbf{b}) = (\sum_{j=1}^M \lambda_j \mathbf{b}_j^4)^{1/2}$ and $\sigma^2 = \sigma_\xi^2 + \sigma_{\zeta_M}^2$, where $\sigma_{\zeta_M} = (\sum_{j=M+1}^\infty \lambda_j (\mathbf{v}_j^*)^4)^{1/2}$. We denote

$$\mathcal{F}^t = \sigma\{\mathbf{v}^0, (\mathbf{Sx}^1, \zeta_M^1, \xi^1), \cdots, (\mathbf{Sx}^t, \zeta_M^t, \xi^t)\}$$

as the filtration involving the full information of all the previous $t$ iterations with $\sigma\{\cdot\}$.

## B.1 High-Probability Results Guarantee

Before the analyses of the two phases, we first introduce the guarantee of our high-probability results. We formally define a series of events for each iteration of Algorithm 1. We demonstrate that these events occur with high probability throughout the whole $T$ iterations, which indicates that the control sequence $\{\mathbf{q}^t\}_{t=0}^T$ we define is aligned with the original sequence $\{\mathbf{v}^t\}_{t=0}^T$ with high probability. This fact is the basis of our high-probability results.

At the $t$-th iteration, Algorithm 1 requires sampling $(\mathbf{Sx}^{t+1}, y^{t+1})$, where $y^{t+1} = \langle \mathbf{Sx}^{t+1}, \mathbf{v}^* \rangle + \zeta_M^{t+1} + \xi^{t+1}$. For simplicity, we denote $\mathbf{Sx}$ as $\mathbf{x}$. In order to simply rule out some low-probabilistic unbounded cases, for each iteration $t$, we define the following four events as:

$$\left\{ \begin{array}{ll} \mathcal{E}_1^{j,t} := \left\{ |\mathbf{x}_j^t| \le \lambda_j^{1/2} R \right\}, & \forall j \in [M], \\ \mathcal{E}_2^{j,t}(\mathbf{v}) := \left\{ |\langle \mathbf{v}^{\odot 2} - \mathbf{v}^{*\odot 2}, \mathbf{x}^t \rangle_{-j}| \le r_j(\mathbf{v}) R \right\}, & \forall j \in [M], \\ \mathcal{E}_3^t := \left\{ |\zeta_M^t| \le \sigma_{\zeta_M} R \right\}, & \\ \mathcal{E}_4^t := \left\{ |\xi^t| \le \sigma_\xi R \right\}, & \end{array} \right\}$$

where $R := \mathcal{O}(\log(MT/\delta))$ and $r_j(\mathbf{v}) := \mathcal{O}(\sum_{i \ne j} \lambda_i [(\mathbf{v}_i)^4 + (\mathbf{v}_i^*)^4])^{1/2}$ for any $\mathbf{v} \in \mathbb{R}^M$.

In Algorithm 1, the original sequence $\{\mathbf{v}^t\}_{t=0}^T$ follows the coordinate-wise update rule as

$$\begin{aligned} \mathbf{v}_j^{t+1} &= \mathbf{v}_j^t - \eta_t \left( \langle \mathbf{v}^{t \odot 2}, \mathbf{x}^{t+1} \rangle - y^{t+1} \right) \mathbf{x}_j^{t+1} \mathbf{v}_j^t \\ &= \mathbf{v}_j^t - \eta_t \langle \mathbf{v}^{t \odot 2} - \mathbf{v}^{*\odot 2}, \mathbf{x}^{t+1} \rangle \mathbf{x}_j^{t+1} \mathbf{v}_j^t + \eta_t \left( \zeta_M^{t+1} + \xi^{t+1} \right) \mathbf{x}_j^{t+1} \mathbf{v}_j^t, \end{aligned}$$

for any $j \in [M]$. Based on Assumption 3.1 and Proposition E.1, we have

$$\min \left\{ \mathbb{P}\left( \mathcal{E}_1^{j,t} \right), \mathbb{P}\left( \mathcal{E}_2^{j,t}(\mathbf{v}^t) \right), \mathbb{P}\left( \mathcal{E}_3^t \right), \mathbb{P}\left( \mathcal{E}_4^t \right) \right\} \ge 1 - \mathcal{O}\left( \frac{\delta}{MT^2} \right),$$

for any $j \in [M]$ and $t \in [T]$. Then we define the compound event as

$$
\mathcal{E} := \left\{ \bigcap_{t=1}^{T} \left( \left( \bigcap_{j=1}^{M} \mathcal{E}_1^{j,t} \right) \bigwedge \left( \bigcap_{j=1}^{M} \mathcal{E}_2^{j,t}(\mathbf{v}^t) \right) \bigwedge \mathcal{E}_3^t \bigwedge \mathcal{E}_4^t \right) \right\}.
$$

We can directly obtain the probability union bound as follows:

$$
\mathbb{P}(\mathcal{E}) = 1 - \mathbb{P}(\mathcal{E}^c) \geq 1 - \sum_{t=1}^{T} \left( 2 - \mathbb{P}(\mathcal{E}_3^t) - \mathbb{P}(\mathcal{E}_4^t) + \sum_{j=1}^{M} \left( 2 - \mathbb{P}\left( \mathcal{E}_1^{j,t} \right) - \mathbb{P}\left( \mathcal{E}_2^{j,t}(\mathbf{v}^t) \right) \right) \right)
$$

$$
\geq 1 - \mathcal{O}\left( \frac{\delta}{T} \right). \tag{8}
$$

The high-probability occurrence of event $\mathcal{E}$ guarantees our analysis of the coordinate-wise update dynamics for the control sequence $\{\mathbf{q}^t\}_{t=0}^{T}$ defined in $\mathbb{R}^M$ as

$$
\mathbf{q}_j^{t+1} = \mathbf{q}_j^t - \eta_t \left( \left( \mathbf{q}_j^t \right)^2 - \left( \mathbf{v}_j^* \right)^2 \right) \left( \mathbf{x}_j^{t+1} \right)^2 \mathbb{1}_{|\mathbf{x}_j^{t+1}| \leq \lambda_j^{1/2} R} \mathbf{q}_j^t
$$

$$
- \eta_t \left\langle \mathbf{q}^{t \odot 2} - \mathbf{v}^{* \odot 2}, \mathbf{x}^{t+1} \right\rangle_{-j} \mathbb{1}_{|\langle \mathbf{q}^{t \odot 2} - \mathbf{v}^{* \odot 2}, \mathbf{x}^{t+1} \rangle_{-j}| \leq r_j(\mathbf{q}^t) R} \mathbf{x}_j^{t+1} \mathbb{1}_{|\mathbf{x}_j^{t+1}| \leq \lambda_j^{1/2} R} \mathbf{q}_j^t
$$

$$
+ \eta_t \left( \zeta_M^{t+1} \mathbb{1}_{|\zeta_M^t| \leq \sigma_{\zeta_M} R} + \xi^{t+1} \mathbb{1}_{|\xi^t| \leq \sigma_\xi R} \right) \mathbf{x}_j^{t+1} \mathbb{1}_{|\mathbf{x}_j^{t+1}| \leq \lambda_j^{1/2} R} \mathbf{q}_j^t, \tag{9}
$$

for any $j \in [M]$ with initialization $\mathbf{q}^0 = \mathbf{v}^0$ is consistent with the analysis of $\{\mathbf{v}^t\}_{t=0}^{T}$ with high probability as Proposition B.1.

**Proposition B.1.** *For any $t \in [T]$, we have $\mathbf{v}^t = \mathbf{q}^t$ with probability at least $1 - \delta/T$.*

To simplify the representation of $\{\mathbf{q}^t\}_{t=0}^{T}$, we introduce four truncated random variables as:

1. $\widehat{\mathbf{x}} \in \mathbb{R}^M$ with entries $\widehat{\mathbf{x}}_j = \mathbf{x}_j \mathbb{1}_{|\mathbf{x}_j| \leq \lambda_j^{1/2} R}$ for any $j \in [M]$,

2. $\widehat{\mathbf{z}}(\mathbf{q}) \in \mathbb{R}^M$ with entries $\widehat{\mathbf{z}}_j(\mathbf{q}) = \left\langle \mathbf{q}^{\odot 2} - \mathbf{v}^{* \odot 2}, \mathbf{x} \right\rangle_{-j} \mathbb{1}_{|\langle \mathbf{q}^{\odot 2} - \mathbf{v}^{* \odot 2}, \mathbf{x} \rangle_{-j}| \leq r_j(\mathbf{q}) R}$

3. $\widehat{\zeta}_M = \zeta_M \mathbb{1}_{\zeta_M \leq \sigma_{\zeta_M} R}$,

4. $\widehat{\xi} = \xi \mathbb{1}_{\xi \leq \sigma_\xi R}$.

Thus, the coordinate-wise update dynamics for $\{\mathbf{q}^t\}_{t=0}^{T}$ in equation 9 can be represented as:

$$
\mathbf{q}_j^{t+1} = \mathbf{q}_j^t - \eta_t \left( \left( \mathbf{q}_j^t \right)^2 - \left( \mathbf{v}_j^* \right)^2 \right) \left( \widehat{\mathbf{x}}_j^{t+1} \right)^2 \mathbf{q}_j^t - \eta_t \widehat{\mathbf{z}}_j^{t+1}(\mathbf{q}^t) \widehat{\mathbf{x}}_j^{t+1} \mathbf{q}_j^t
$$

$$
+ \eta_t \left( \widehat{\zeta}_M^{t+1} + \widehat{\xi}^{t+1} \right) \widehat{\mathbf{x}}_j^{t+1} \mathbf{q}_j^t, \tag{10}
$$

for any $j \in [M]$.

## B.2 Proof of Phase I

In this section, we formally propose the proof techniques of **Phase I** in Theorem B.1. Theorem B.1 establishes that Algorithm 1 adaptively selects a effective dimension $D \in \mathbb{N}_+$ with the following convergence properties: (1) for $j \leq D$, $\mathbf{v}_j^{T_1}$ converges to an adaptive neighborhood of $\mathbf{v}_j^*$; (2) for $j > D$, $\mathbf{v}_j^{T_1}$ is bounded by $\frac{3}{2} \max\{\mathbf{v}_j^*, 2\mathbf{v}_j^0\}$. Theorem B.1 specifies the intrinsic relationship between Algorithm 1's key parameters: the recommended step size $\eta$, effective dimension $D$, and total sample size $T$. Furthermore, under Assumption 3.3, Phase II analysis demonstrates the optimality of the effective dimension $D$ selected in Theorem B.1.

**Theorem B.1.** *[Formal version of Theorem 5.1 ] Under Assumption 3.1, consider the dynamic generated via Algorithm 1 with initialization $\mathbf{v}_0$. Denote (1) the threshold vector $\widehat{\mathbf{v}}^* \in \mathbb{R}^M$ with coordinate $\widehat{\mathbf{v}}_j^* = \max\left\{ \frac{3}{2}\mathbf{v}_j^*, 3\mathbf{v}_j^0 \right\}$ for any $j \in [M]$; (2) the composite vector $\mathbf{b} = ((1 + c_1)(\mathbf{v}_{1:D}^*)^\top, (\widehat{\mathbf{v}}_{D+1:M}^*)^\top)^\top$, where the scaling constant $c_1 \in (0, 1/2)$. Let the step size $\eta$*

*satisfy* $\eta \leq \widetilde{\Omega}\left(\frac{c_1^2 \bar{\sigma}_{\min}(\max\{D,M\})}{[\sigma^2 + \mathcal{M}^2(\mathbf{b})]^2}\right)$ *for the given effective dimension* $D \in \mathbb{N}_+$. *If the iteration number* $T_1$ *requires:*

$$T_1 \in \begin{cases} \left[\widetilde{\mathcal{O}}\left(\frac{\sigma^2 + \mathcal{M}^2(\mathbf{b})}{c_1^2 \eta \sigma_{\min}(D)}\right) : \widetilde{\Omega}\left(\frac{\tilde{\sigma}_{\max}^{-1}(D)}{\eta^2 [\sigma^2 + \mathcal{M}^2(\mathbf{b})]}\right)\right], & \text{if } D < M, \\ \left[\widetilde{\mathcal{O}}\left(\frac{\sigma^2 + \mathcal{M}^2(\mathbf{b})}{c_1^2 \eta \sigma_{\min}(M)}\right) : \infty\right), & \text{otherwise,} \end{cases}$$

*then the dynamic satisfies the following convergence property:*

$$\mathbf{v}_j^{T_1} \in \begin{cases} \left[\mathbf{v}_j^* - c_1 \mathbf{v}_j^*, \mathbf{v}_j^* + c_1 \mathbf{v}_j^*\right], & \text{if } j \in [D], \\ \left[0, \frac{3}{2} \max\{\mathbf{v}_j^*, 2\mathbf{v}_j^0\}\right], & \text{otherwise,} \end{cases} \tag{11}$$

*with probability at least* $1 - \delta$.

Before the beginning of our proof, we define the **b**-capped coupling processes used in the following lemmas as below.

**Definition B.1** (b-capped coupling). *Let* $\{\mathbf{q}^t\}_{t=0}^T$ *be a Markov chain in* $\mathbb{R}_+^M$ *adapted to filtration* $\{\mathcal{F}^t\}_{t=0}^T$. *Given threshold vector* $\mathbf{b} \in \mathbb{R}_+^M$, *the* **b**-*capped coupling process* $\{\bar{\mathbf{v}}^t\}_{t=0}^T$ *with initialization* $\bar{\mathbf{v}}^0 = \mathbf{q}^0 \leq \mathbf{b}$ *evolves as:*

1. *Updating state: If* $\bar{\mathbf{v}}^t \leq \mathbf{b}$, *let* $\bar{\mathbf{v}}^{t+1} = \mathbf{q}^{t+1}$,

2. *Absorbing state: Otherwise, maintain* $\bar{\mathbf{v}}^{t+1} = \bar{\mathbf{v}}^t$.

### B.2.1 PART I: THE COORDINATE-WISE UPPER BOUNDS OF $\mathbf{v}^{T_1}$.

In this part, we establish coordinate-wise upper bounds for $\mathbf{v}^{T_1}$ in Lemma B.1. For each coordinate $i \in [M]$, we develop a geometrically compensated supermartingale $\{u_i^t := (1 - \eta\Theta(\lambda_i(\mathbf{v}_i^*)^2))^{-t}(\bar{\mathbf{v}}_i^t - \mathbf{v}_i^*)\}_{t=1}^{T_1}$ using the **b**-capped coupling sequence $\{\bar{\mathbf{v}}^t\}_{t=0}^{T_1}$ derived from the control sequence $\{\mathbf{q}^t\}_{t=0}^{T_1}$. We precisely calculate the sub-Gussian parameters of the supermartingale increments through geometric series summation over $\mathcal{S}$ and linear summation over $\mathcal{S}^c$. The analysis enables the application of Bernstein-type inequalities to establish the claimed concentration results in Lemma B.1.

**Lemma B.1.** *[Formal version of Lemma A.1] Under the setting of Theorem B.1, let* $\{\mathbf{q}^t\}_{t=0}^{T_1}$ *be a Markov chain with its* **b**-*capped coupling process* $\{\bar{\mathbf{v}}^t\}_{t=0}^{T_1}$. *When* $\eta \leq \widetilde{\Omega}\left(\frac{1}{\sigma^2 + \mathcal{M}^2(\mathbf{b})}\right)$, *the inequality* $\bar{\mathbf{v}}^t \geq \mathbf{0}$ *holds for any* $t \in [T_1]$. *For any* $\mathbf{v} \in \mathbb{R}^M$, *define the truncation event* $\mathcal{A}(\mathbf{v}) := \{\mathbf{v} \leq \mathbf{b}\}$. *For* $\delta \in (0,1)$, *the following conditions guarantee that* $\mathcal{A}(\bar{\mathbf{v}}^{T_1})$ *holds with probability at least* $1 - \frac{\delta}{6}$:

1. *Dominant coordinates condition:* $\bar{\sigma}_{\min}(D) \geq \frac{\eta}{c_1^2}\mathcal{O}([\sigma^2 + \mathcal{M}^2(\mathbf{b})]\log^5(MT_1/\delta))$,

2. *Residual spectrum condition:* $\tilde{\sigma}_{\max}(D) \geq T_1\eta^2\mathcal{O}([\sigma^2 + \mathcal{M}^2(\mathbf{b})]\log(\max\{M - D, 0\}T_1/\delta)\log^4(MT_1/\delta))$.

*Proof.* Define the random variable

$$p_j^{t+1} := \left(\left((\bar{\mathbf{v}}_j^t)^2 - (\mathbf{v}_j^*)^2\right)\hat{\mathbf{x}}_j^{t+1} + \hat{\mathbf{z}}_j^{t+1}(\bar{\mathbf{v}}^t) - \widehat{\zeta}_M^{t+1} - \hat{\xi}^{t+1}\right)\hat{\mathbf{x}}_j^{t+1}$$

for any $j \in [M]$ and $t \in [0 : T_1 - 1]$. Then in the updating state of $\{\bar{\mathbf{v}}^t\}_{t=0}^{T_1}$, we have

$$\bar{\mathbf{v}}_j^{t+1} = (1 - \eta p_j^{t+1})\bar{\mathbf{v}}_j^t, \quad \forall j \in [M]. \tag{12}$$

Based on the boundedness of $p_j^{t+1}$ and the appropriately chosen step size $\eta \leq \widetilde{\Omega}\left(\frac{1}{\sigma^2 + \mathcal{M}^2(\mathbf{b})}\right)$, if $\bar{\mathbf{v}}^t > \mathbf{0}$, then we have $\bar{\mathbf{v}}^{t+1} \geq \frac{1}{2}\bar{\mathbf{v}}^t$. Since $\bar{\mathbf{v}}^0 > 0$, we have $\bar{\mathbf{v}}^t > 0$ for any $t \in [T_1]$ by induction. Let $\bar{\tau}_{\mathbf{b}}$ be the stopping time when $\bar{\mathbf{v}}_j^{\bar{\tau}_{\mathbf{b}}} > \mathbf{b}_j$ for a certain coordinate $j \in [M]$, i.e.,

$$\bar{\tau}_{\mathbf{b}} = \inf_t \left\{t : \exists j \in [M], \text{ s.t. } \bar{\mathbf{v}}_j^t > \mathbf{b}_j\right\}.$$

For each coordinate $1 \leq j \leq M$, let $\bar{\tau}_{\mathbf{b},j}$ be the stopping time when $\bar{\mathbf{v}}_j^{\bar{\tau}_{\mathbf{b},j}} > \mathbf{b}_j$, i.e.,

$$\bar{\tau}_{\mathbf{b},j} = \inf_t \left\{ t : \bar{\mathbf{v}}_j^t > \mathbf{b}_j \right\}.$$

Based on Definition B.1, when the stopping time $\bar{\tau}_{\mathbf{b}} = t_2$ occurs for some $t_2 \in [T_1]$, the coupling process satisfies $\bar{\mathbf{v}}^t = \bar{\mathbf{v}}^{t_2}$ for all $t > t_2$. We categorize the following two cases and analyze the probability bound respectively.

**Case I:** Suppose there exists $j \in [D]$ such that $\bar{\tau}_{\mathbf{b},j} = t_2$. That is, the event $\mathcal{A}\left(\bar{\mathbf{v}}^t\right)$ holds for all $t \in [0 : t_2 - 1]$. The boundedness of $p_j^{t+1}$ and the dominant coordinates condition of $\eta$ in Lemma B.1 indicate that $\bar{\mathbf{v}}_j^t$ must traverse in and out of the threshold interval $\left[\frac{1}{1+c_1}\mathbf{b}_j, \mathbf{b}_j\right]$ before exceeding $\mathbf{b}_j$. We aim to estimate the following probability for coordinates $j \in [D]$ and time pairs $t_1 < t_2 \in [T_1]$:

$$\mathbb{P}\left(\mathcal{B}_{t_1}^{\bar{\tau}_{\mathbf{b},j}=t_2}(j) = \left\{\bar{\mathbf{v}}_j^{t_1} \leq \frac{1+c_1/2}{1+c_1}\mathbf{b}_j \bigwedge \bar{\mathbf{v}}_j^{t_1:t_2-1} \in \left[\frac{1}{1+c_1}\mathbf{b}_j, \mathbf{b}_j\right] \bigwedge \bar{\mathbf{v}}_j^{t_2} > \mathbf{b}_j\right\}\right).$$

For any $t \in [t_1 : t_2 - 1]$, we have

$$\begin{aligned}
\mathbb{E}\left[\bar{\mathbf{v}}_j^{t+1} - \mathbf{v}_j^* \mid \mathcal{F}^t\right] &= \mathbb{E}_{\mathbf{x}_{1:M}^{t+1}, \xi^{t+1}, \zeta_M^{t+1}}\left[\bar{\mathbf{v}}_j^t - \mathbf{v}_j^* - \eta p_j^{t+1}\bar{\mathbf{v}}_j^t\right] \\
&\overset{(a)}{\leq} \left(1 - \frac{1}{2}\eta\lambda_j\bar{\mathbf{v}}_j^t(\bar{\mathbf{v}}_j^t + \mathbf{v}_j^*)\right)(\bar{\mathbf{v}}_j^t - \mathbf{v}_j^*) \\
&\leq \left(1 - \frac{1+c_1/2}{(1+c_1)^2}\eta\lambda_j(\mathbf{v}_j^*)^2\right)(\bar{\mathbf{v}}_j^t - \mathbf{v}_j^*),
\end{aligned} \tag{13}$$

where (a) is due to Assumption 3.1 and Lemma E.2. By applying Lemma E.1 to $p_j^t$, we demonstrate that $p_j^t$ satisfies the sub-Gaussian property for all $t \in [0 : T_1 - 1]$. Thus we have

$$\mathbb{E}\left[\exp\left\{\lambda\left(\bar{\mathbf{v}}_j^{t+1} - \mathbb{E}[\bar{\mathbf{v}}_j^{t+1} \mid \mathcal{F}^t]\right)\right\} \mid \mathcal{F}^t\right] \leq \exp\left\{\frac{\lambda^2\eta^2\lambda_j(\mathbf{v}_j^*)^2\mathcal{O}\left(\left[\sigma^2 + \mathcal{M}^2(\mathbf{b})\right]\log^4(MT_1/\delta)\right)}{2}\right\},$$

for any $\lambda \in \mathbb{R}$. Combining Lemma E.3 with equation 13, we can establish the probability bound for event $\mathcal{B}_{t_1}^{\bar{\tau}_{\mathbf{b},j}=t_2}(j)$ for any time pair $t_1 < t_2 \in [T_1]$ as

$$\mathbb{P}\left(\mathcal{B}_{t_1}^{\bar{\tau}_{\mathbf{b},j}=t_2}(j)\right) \leq \exp\left\{-\frac{c_1^2(\mathbf{v}_j^*)^2}{\eta\mathcal{O}\left(\left[\sigma^2 + \mathcal{M}^2(\mathbf{b})\right]\log^4(MT_1/\delta)\right)}\right\}. \tag{14}$$

**Case II:** Suppose there exists $j \in [D+1 : M]$ such that $\bar{\tau}_{\mathbf{b},j} = t_2$. Similarly, $\bar{\mathbf{v}}_j^t$ must traverse in and out of the threshold interval $[\frac{2}{3}\mathbf{b}_j, \mathbf{b}_j]$ before exceeding $\mathbf{b}_j$. Therefore, we aim to estimate the following probability for coordinates $j \in [D+1 : M]$ and time pairs $t_1 < t_2 \in [T_1]$:

$$\mathbb{P}\left(\mathcal{C}_{t_1}^{\bar{\tau}_{\mathbf{b},j}=t_2}(j) = \left\{\bar{\mathbf{v}}_j^{t_1} \leq \frac{3}{4}\mathbf{b}_j \bigwedge \bar{\mathbf{v}}_j^{t_1:t_2-1} \in \left[\frac{2}{3}\mathbf{b}_j, \mathbf{b}_j\right] \bigwedge \bar{\mathbf{v}}_j^{t_2} > \mathbf{b}_j\right\}\right).$$

For any $t \in [t_1 : t_2 - 1]$, we have

$$\begin{aligned}
\mathbb{E}\left[\bar{\mathbf{v}}_j^{t+1} - \mathbf{v}_j^* \mid \mathcal{F}^t\right] &= \mathbb{E}_{\mathbf{x}_{1:M}^{t+1}, \xi^{t+1}, \zeta_M^{t+1}}\left[\bar{\mathbf{v}}_j^t - \mathbf{v}_j^* - \eta p_j^{t+1}\bar{\mathbf{v}}_j^t\right] \\
&\leq \left(1 - \frac{1}{2}\eta\lambda_j\bar{\mathbf{v}}_j^t(\bar{\mathbf{v}}_j^t + \mathbf{v}_j^*)\right)(\bar{\mathbf{v}}_j^t - \mathbf{v}_j^*) \\
&\leq \bar{\mathbf{v}}_j^t - \mathbf{v}_j^*.
\end{aligned} \tag{15}$$

Similarly, based on Lemma E.1, we have

$$\mathbb{E}\left[\exp\left\{\lambda(\bar{\mathbf{v}}_j^{t+1} - \mathbb{E}[\bar{\mathbf{v}}_j^{t+1} \mid \mathcal{F}^t])\right\} \mid \mathcal{F}^t\right] \leq \exp\left\{\frac{\lambda^2\eta^2\lambda_j(\mathbf{b}_j^*)^2\mathcal{O}\left(\left[\sigma^2 + \mathcal{M}^2(\mathbf{b})\right]\log^4(MT_1/\delta)\right)}{2}\right\},$$

for any $\lambda \in \mathbb{R}$. Combining Lemma E.3 with equation 15, we can establish the probability bound for event $\mathcal{C}_{t_1}^{\bar{\tau}_{\mathbf{b},j}=t_2}(j)$ for any time pair $t_1 < t_2 \in [T_1]$ as

$$\mathbb{P}\left(\mathcal{C}_{t_1}^{\bar{\tau}_{\mathbf{b},j}=t_2}(j)\right) \leq \exp\left\{-\frac{1}{T\eta^2\lambda_j\mathcal{O}\left(\left[\sigma^2 + \mathcal{M}^2(\mathbf{b})\right]\log^2(MT_1/\delta)\right)}\right\}. \tag{16}$$

Finally, combining the probability bounds equation 14 and equation 16 with the dominant coordinates condition and residual spectrum condition in Lemma B.1, we obtain the following probability bound for complement event $\mathcal{A}^c(\bar{\mathbf{v}}^{T_1})$:

$$
\begin{aligned}
\mathbb{P}\left(\mathcal{A}^c(\bar{\mathbf{v}}^{T_1})\right) &\leq \sum_{j=1}^{D} \sum_{1 \leq t_1 < t_2 \leq T_1} \mathbb{P}\left(\mathcal{B}_{t_1}^{\bar{\tau}_{\mathbf{b},j}=t_2}(j)\right) + \sum_{j=D+1}^{M} \sum_{1 \leq t_1 < t_2 \leq T_1} \mathbb{P}\left(\mathcal{C}_{t_1}^{\bar{\tau}_{\mathbf{b},j}=t_2}(j)\right) \\
&\leq \frac{NT_1^2}{2} \exp\left\{-\frac{c_1^2 \min_{1 \leq j \leq D}(\mathbf{v}_j^*)^2}{\eta \mathcal{O}\left([\sigma^2 + \mathcal{M}^2(\mathbf{b})] \log^4(MT_1/\delta)\right)}\right\} \\
&\quad + \max\{M - D, 0\} T_1 \exp\left\{-\frac{\min_{D+1 \leq j \leq M} \lambda_j^{-1}}{T_1 \eta^2 \mathcal{O}\left([\sigma^2 + \mathcal{M}^2(\mathbf{b})] \log^4(MT_1/\delta)\right)}\right\} \\
&\leq \frac{\delta}{12}.
\end{aligned}
\tag{17}
$$

$\square$

Lemma B.1 establishes the adaptive high-probability upper bounds for each coordinate of $\bar{\mathbf{v}}^{T_1}$. According to the construction methodology of the coupling process $\{\bar{\mathbf{v}}^t\}_{t=0}^{T_1}$, these bounds can be naturally extended to $\mathbf{q}^{T_1}$. Moreover, the high-probability consistency between control sequence $\{\mathbf{q}^t\}_{t=0}^{T}$ and original sequence $\{\mathbf{v}^t\}_{t=0}^{T}$ (refer to Proposition B.1) allows the direct application of Lemma B.1 to $\mathbf{v}^{T_1}$. It similarly holds for Lemmas B.2 and B.3, respectively.

### B.2.2 PART II: THE COORDINATE-WISE LOWER BOUNDS OF $\bar{\mathbf{v}}^{T_1}$

Deriving a direct high-probability lower bound for $\bar{\mathbf{v}}^{T_1}$ proves to be a challenge. We turn to the lower bound of $\max_{t \leq T_1} \bar{\mathbf{v}}_j^t$ during $T_1$ iterations. First we propose Lemma B.2 to construct such bounds for $\max_{t \leq T_1} \bar{\mathbf{v}}_j^t$ adaptively over $j \in [D]$. We derive a subcoupling sequence $\{\breve{\mathbf{v}}^{i,t}\}_{t=0}^{T_1}$ from the original coupling sequence $\{\bar{\mathbf{v}}^t\}_{t=0}^{T_1}$ for any $i \in \mathcal{S}$. Each subcoupling sequence undergoes logarithmic transformation to generate a linearly compensated submartingale $\{-t \log(1 + \eta \mathcal{O}(\lambda_i(\mathbf{v}_i^*)^2)) + \log(\breve{\mathbf{v}}_i^{i,t})\}_{t=1}^{T_1}$. These $|\mathcal{S}|$ submartingales exhibit monotonic growth with sub-Gaussian increments. Applying Bernstein-type concentration inequalities, we obtain $\max_{t \leq T_1} \mathbf{v}_i^t \geq (1 - c_1/2)\mathbf{v}_i^*$ with high probability for any $i \in \mathcal{S}$ in Lemma B.2.

**Lemma B.2.** *[Formal version of Lemma A.2] Under the setting of Lemma B.1, let*

$$
\eta \leq \frac{c_1 \log^{-4}(MT_1/\delta) \min_{j \in [D]}(\mathbf{v}_j^*)^2}{\mathcal{O}(\sigma^2 + (1+C)\mathcal{M}^2(\mathbf{b}))}
$$

*and*

$$
T_1 \geq \max\left\{\frac{\mathcal{O}\left(\max_{j \in [D]} -\log(\mathbf{v}_j^0)\right)}{c_1 \eta \sigma_{\min}(D)}, \frac{\mathcal{O}\left([\sigma^2 + \mathcal{M}^2(\mathbf{b})] \log^8(MT_1/\delta)\right)}{c_1^2 \min_{j \in [D]}(\lambda_j(\mathbf{v}_j^*)^4)}\right\}
$$

*The combined event set satisfies* $\mathbb{P}\left(\left(\bigcap_{j=1}^{D} \mathcal{E}_{1,j}\right) \bigcup \mathcal{E}_2\right) \geq 1 - \frac{\delta}{6}$. *where*

$$
\mathcal{E}_{1,j} := \left\{\max_{t \leq T_1} \bar{\mathbf{v}}_j^t \geq \frac{1 - c_1/2}{1 + c_1} \mathbf{b}_j\right\}, \quad \forall j \in [D],
$$

*and* $\mathcal{E}_2 := \left\{\mathcal{A}^c(\bar{\mathbf{v}}^{T_1})\right\}$.

*Proof.* For a fixed $j \in [D]$, we define the subcoupling $\{\breve{\mathbf{v}}^t\}_{t=0}^{T_1}$ with initialization $\breve{\mathbf{v}}^0 = \bar{\mathbf{v}}^0$ as follows:

1. Updating state: If event $\mathcal{B}_t(j) = \left\{\mathcal{A}(\breve{\mathbf{v}}^t) \bigwedge \breve{\mathbf{v}}_j^t < \frac{1-c_1/2}{1+c_1} \mathbf{b}_j\right\}$ holds, let $\breve{\mathbf{v}}^{t+1} = \bar{\mathbf{v}}^{t+1}$,

2. Multiplicative scaling state: Otherwise, let $\breve{\mathbf{v}}^{t+1} = \left(1 + \frac{c_1(1-c_1)\eta}{2} \lambda_j(\mathbf{v}_j^*)^2\right) \breve{\mathbf{v}}^t$.

We aim to demonstrate that $-t \log(1 + \frac{c_1(1-c_1)\eta}{2}\lambda_j(\mathbf{v}_j^*)^2) + \log(\breve{\mathbf{v}}_j^t)$ is a submartingale. If event $\mathcal{B}_t^c(j)$ holds, we directly obtain $\mathbb{E}[\log(\breve{\mathbf{v}}_j^{t+1}) \mid \mathcal{F}^t] \geq \log(1 + \frac{c_1(1-c_1)\eta}{2}\lambda_j(\mathbf{v}_j^*)^2) + \log(\breve{\mathbf{v}}_j^t)$. Otherwise, letting

$$w_j^t := \hat{z}_j^t(\bar{\mathbf{v}}^{t-1}) - \widehat{\zeta}_M^t - \hat{\xi}^t, \quad \forall t \in [T_1],$$

we have

$$
\begin{aligned}
\mathbb{E}\left[\log(\breve{\mathbf{v}}_j^{t+1}) \mid \mathcal{F}^t\right] &= \mathbb{E}\left[\log(\bar{\mathbf{v}}_j^{t+1}) \mid \mathcal{F}^t\right] \\
&= \mathbb{E}_{\mathbf{x}_{1:M}^{t+1}, \xi^{t+1}, \zeta_M^{t+1}}\left[\log\left(1 - \eta\left((\bar{\mathbf{v}}_j^t)^2 - (\mathbf{v}_j^*)^2\right)(\hat{\mathbf{x}}_j^{t+1})^2 - \eta w_j^{t+1}\hat{\mathbf{x}}_j^{t+1}\right)\right] \\
&\quad + \log(\bar{\mathbf{v}}_j^t) \\
&\overset{(a)}{\geq} \log\left(1 + \frac{3c_1(1-c_1)\eta}{4}\lambda_j(\mathbf{v}_j^*)^2\right) \\
&\quad - \eta^2\lambda_j\mathcal{O}\left(\left[\sigma^2 + (1+C)\mathcal{M}^2(\mathbf{b})\right]\log^4(MT_1/\delta)\right) + \log(\bar{\mathbf{v}}_j^t) \\
&\overset{(b)}{\geq} \log\left(1 + \frac{c_1(1-c_1)\eta}{2}\lambda_j(\mathbf{v}_j^*)^2\right) + \log(\bar{\mathbf{v}}_j^t) \\
&\overset{(c)}{=} \log\left(1 + \frac{c_1(1-c_1)\eta}{2}\lambda_j(\mathbf{v}_j^*)^2\right) + \log(\breve{\mathbf{v}}_j^t),
\end{aligned}
$$

where (a) is based on the following three facts: 1) the Taylor expansion of $\log(a + \cdot)$ with $a = 1 + \eta((\mathbf{v}_j^*)^2 - (\bar{\mathbf{v}}_j^t)^2)\mathbb{E}[(\hat{\mathbf{x}}_j^{t+1})^2]$; 2) the property that $Y_j^{t+1}$ is zero-mean and independent of $\hat{\mathbf{x}}_j^{t+1}$; and 3) the step size $\eta \leq \frac{c_1(\mathbf{v}_j^*)^2\log^{-4}(MT_1/\delta)}{\mathcal{O}(\sigma^2 + (1+C)\mathcal{M}^2(\mathbf{b}))}$ ensures that $1 - \tau\eta((\bar{\mathbf{v}}_j^t)^2 - (\mathbf{v}_j^*)^2)[(\hat{\mathbf{x}}_j^{t+1})^2 - \mathbb{E}[(\hat{\mathbf{x}}_j^{t+1})^2]] - \tau\eta w_j^{t+1}\hat{\mathbf{x}}_j^{t+1} \geq 1/2$ for any $\tau \in [0, 1]$, (b) is due to the inequality $\log(1 + \frac{c_1(1-c_1)\eta}{16}\lambda_j(\mathbf{v}_j^*)^2) \geq \eta^2\lambda_j\mathcal{O}([\sigma^2 + (1+C)\mathcal{M}^2(\mathbf{b})]\log^4(MT_1/\delta))$, and (c) relies on the temporal exclusivity property that if event $\mathcal{B}_t^c(j)$ occurs at time $t$, then $\mathcal{B}_t(j)$ is permanently excluded for all subsequent times $t' > t$. Therefore, based on the submartingale, we obtain

$$
\begin{aligned}
&\mathbb{P}\left\{\breve{\mathbf{v}}_j^{T_1} < \frac{1 - c_1/2}{1 + c_1}\mathbf{b}_j\right\} \\
&\overset{(d)}{\leq} \exp\left\{-\frac{2\left(T_1\log\left(1 + \frac{c_1(1-c_1)\eta}{2}\lambda_j(\mathbf{v}_j^*)^2\right) + \log(v_j^0) - \log\left(\frac{1-c_1/2}{1+c_1}\mathbf{b}_j\right)\right)^2}{T_1\eta^2\lambda_j\mathcal{O}\left([\sigma^2 + \mathcal{M}^2(\mathbf{b})]\log^6(MT_1/\delta)\right)}\right\} \\
&\overset{(e)}{\leq} \exp\left\{-\frac{T_1\log^2\left(1 + \frac{c_1(1-c_1)\eta}{2}\lambda_j(\mathbf{v}_j^*)^2\right)}{\eta^2\lambda_j\mathcal{O}\left([\sigma^2 + \mathcal{M}^2(\mathbf{b})]\log^6(MT_1/\delta)\right)}\right\} \\
&\overset{(f)}{\leq} \frac{\delta}{12N},
\end{aligned}
\tag{18}
$$

where (d) is derived from Azuma's inequality and the estimation of $\left|\log(\breve{\mathbf{v}}_j^{t+1}) - \log(\breve{\mathbf{v}}_j^t)\right|$ below:

$$\left|\log(\breve{\mathbf{v}}_j^{t+1}) - \log(\breve{\mathbf{v}}_j^t)\right| \leq \eta\lambda_j^{1/2}\mathcal{O}\left(\left[\sigma^2 + \mathcal{M}^2(\mathbf{b})\right]^{1/2}\log^4(MT_1/\delta)\right),\tag{19}$$

which implies that

$$\left|\log(\breve{\mathbf{v}}_j^{t+1}) - \log\left(1 + \frac{c_1(1-c_1)\eta}{2}\lambda_j(\mathbf{v}_j^*)^2\right) - \log(\breve{\mathbf{v}}_j^t)\right|^2 \leq \eta^2\lambda_j\mathcal{O}\left([\sigma^2 + \mathcal{M}^2(\mathbf{b})]\log^8(MT_1/\delta)\right).$$

Moreover, since $T_1\log(1 + \frac{c_1(1-c_1)\eta}{2}\lambda_j(\mathbf{v}_j^*)^2)/4 \geq -\log(v_j^0)$ and $c_1^2T_1\lambda_j(\mathbf{v}_j^*)^4 \geq \mathcal{O}([\sigma^2 + \mathcal{M}^2(\mathbf{b})]\log^8(MT_1/\delta))$, we obtain inequalities (e) and (f). If $\mathcal{A}(\bar{\mathbf{v}}^{T_1})$ holds, equation 18 illustrates that $\mathbb{P}(\mathcal{E}_{1,j}^c) \leq \frac{\delta}{12N}$. Thus, we have $\mathbb{P}(\mathcal{E}_{1,j}^c \bigcap \mathcal{E}_2^c) \leq \frac{\delta}{6N}$. $\qquad\square$

Second, we construct the high-probability lower bound for $\bar{\mathbf{v}}_j^{T_1}$ for any $j \in [D]$ in Lemma B.3. The proof technique of Lemma B.3 mirrors that of Lemma B.1. By contrast, we construct geometrically compensated supermartingale $\{-u_i^t\}_{t=1}^{T_1}$ for each $i \in \mathcal{S}$. The proof is finished by applying Bernstein-type concentration inequalities to these constructed supermartingales, yielding the required probabilistic bounds.

**Lemma B.3.** *[Formal version of Lemma A.3] Under the setting of Lemma B.1, let*

$$\eta \leq \frac{c_1^2 \bar{\sigma}_{\min}(D)}{\mathcal{O}([\sigma^2 + \mathcal{M}^2(\mathbf{b})] \log^4(MT_1/\delta))}.$$

*The combined event set satisfies* $\mathbb{P}\left(\bigcap_{j=1}^{D}\left(\bigcup_{k=3,4}\mathcal{E}_{k,j}\right)\right) \geq 1 - \frac{\delta}{6}$, *where*

$$\mathcal{E}_{3,j} := \left\{\max_{t \leq T_1} \bar{\mathbf{v}}_j^t < \frac{1 - c_1/2}{1 + c_1}\mathbf{b}_j\right\}, \quad \mathcal{E}_{4,j} := \left\{\bar{\mathbf{v}}_j^{T_1} \geq \frac{1 - c_1}{1 + c_1}\mathbf{b}_j\right\}, \quad \forall j \in [D].$$

*Proof.* For any $j \in [D]$, if $\mathcal{E}_{3,j}^c$ occurs, there exists $t \in [T_1]$ such that $\bar{\mathbf{v}}_j^t \geq \frac{1-c_1/2}{1+c_1}\mathbf{b}_j$. Define $\tau_{0,j}$ as the stopping time satisfying $\bar{\mathbf{v}}_j^{\tau_{0,j}} \geq \frac{1-c_1/2}{1+c_1}\mathbf{b}_j$ as:

$$\tau_{0,j} = \inf_t\left\{t : \bar{\mathbf{v}}_j^t \geq \frac{1 - c_1/2}{1 + c_1}\mathbf{b}_j\right\}.$$

We also define $\tau_{1,j}$ as the stopping time satisfying $\bar{\mathbf{v}}_j^{\tau_{1,j}} < \frac{1-c_1}{1+c_1}\mathbf{b}_j$ after $\tau_{0,j}$ as:

$$\tau_{1,j} = \inf_{t > \tau_{0,j}}\left\{t : \bar{\mathbf{v}}_j^t < \frac{1 - c_1}{1 + c_1}\mathbf{b}_j\right\}.$$

Based on the definition of $\{\bar{\mathbf{v}}^t\}_{t=0}^{T_1}$, once the event $\mathcal{A}^c(\bar{\mathbf{v}}^t)$ occurs, the coupling process satisfies $\bar{\mathbf{v}}^{t'} = \bar{\mathbf{v}}^t$ for any $t' > t$. Therefore, $\mathcal{A}(\bar{\mathbf{v}}^t)$ holds for all $t \leq \tau_{1,j}$. Moreover, $\bar{\mathbf{v}}_j^t$ must traverse in and out of the threshold interval $\left[\frac{1-c_1}{1+c_1}\mathbf{b}_j, \frac{1}{1+c_1}\mathbf{b}_j\right]$ before subceeding $\frac{1-c_1}{1+c_1}\mathbf{b}_j$. We aim to estimate the following probability for coordinates $j \in [D]$ and time pairs $t_0 < t_1 \in [T_1]$:

$$\mathbb{P}\left(\bar{\mathcal{D}}_{\tau_0=t_0}^{\tau_1=t_1}(j) = \left\{\bar{\mathbf{v}}_j^{t_0} \geq \frac{1 - c_1/2}{1 + c_1}\mathbf{b}_j \bigwedge \bar{\mathbf{v}}_j^{t_0:t_1-1} \in \left[\frac{1 - c_1}{1 + c_1}\mathbf{b}_j, \frac{1}{1 + c_1}\mathbf{b}_j\right] \bigwedge \bar{\mathbf{v}}_j^{t_1} < \frac{1 - c_1}{1 + c_1}\mathbf{b}_j\right\}\right).$$

For any $t \in [t_0 : t_1 - 1]$, we have

$$\mathbb{E}\left[\mathbf{v}_j^* - \bar{\mathbf{v}}_j^{t+1} \mid \mathcal{F}^t\right] = \mathbb{E}_{\mathbf{x}_{1:M}^{t+1}, \xi^{t+1}, \varsigma_{M+1:\infty}^{t+1}}\left[\mathbf{v}_j^* - \bar{\mathbf{v}}_j^t + \eta\left(((\bar{\mathbf{v}}_j^t)^2 - (\mathbf{v}_j^*)^2)\hat{\mathbf{x}}_j^{t+1} + \hat{\mathbf{z}}_j^{t+1}(\bar{\mathbf{v}}^t)\right.\right.$$
$$\left.\left. - \widehat{\zeta}_M^{t+1} - \hat{\xi}^{t+1}\right)\hat{\mathbf{x}}_j^{t+1}\bar{\mathbf{v}}_j^{t+1}\right] \tag{20}$$
$$\leq \left(1 - \frac{1 - c_1}{(1 + c_1)^2}\eta\lambda_j(\mathbf{v}_j^*)^2\right)(\mathbf{v}_j^* - \bar{\mathbf{v}}_j^t).$$

Applying Lemma E.1 to $(((\bar{\mathbf{v}}_j^t)^2 - (\mathbf{v}_j^*)^2)\hat{\mathbf{x}}_j^{t+1} + \hat{\mathbf{z}}_j^{t+1}(\bar{\mathbf{v}}^t) - \widehat{\zeta}_M^{t+1} - \hat{\xi}^{t+1})\hat{\mathbf{x}}_j^{t+1}$, we have

$$\mathbb{E}\left[\exp\left\{\lambda\left(\mathbb{E}[\bar{\mathbf{v}}_j^{t+1} \mid \mathcal{F}^t] - \bar{\mathbf{v}}_j^{t+1}\right)\right\} \mid \mathcal{F}^t\right] \leq \exp\left\{\frac{\lambda^2\eta^2\lambda_j(\mathbf{v}_j^*)^2\mathcal{O}\left([\sigma^2 + \mathcal{M}^2(\mathbf{b})]\log^4(MT_1/\delta)\right)}{2}\right\},$$

for any $\lambda \in \mathbb{R}$. Therefore, combining Lemma E.3 with equation 20, we establish the probability bound for event $\bar{\mathcal{D}}_{\tau_0=t_0}^{\tau_1=t_1}(j)$ with any time pair $t_0 < t_1 \in [T_1]$ as

$$\mathbb{P}\left(\bar{\mathcal{D}}_{\tau_0=t_0}^{\tau_1=t_1}(j)\right) \leq \exp\left\{\frac{-c_1^2(\mathbf{v}_j^*)^2}{\eta\mathcal{O}\left([\sigma^2 + \mathcal{M}^2(\mathbf{b})]\log^4(MT_1/\delta)\right)}\right\}.$$

Notice that the occurrence of $\mathcal{E}_{3,j}^c \bigwedge \mathcal{E}_{4,j}^c$ implies $\bar{\mathcal{D}}_{\tau_0=t_0}^{\tau_1=t_1}(j)$ must hold for certain $t_0 < t_1 \in [T_1]$. Therefore, we have

$$\mathbb{P}\left(\mathcal{E}_{3,j}^c \bigwedge \mathcal{E}_{4,j}^c\right) \leq \sum_{1 \leq t_1 < t_2 \leq T_1} \mathbb{P}\left(\bar{\mathcal{D}}_{\tau_0=t_0}^{\tau_1=t_1}(j)\right)$$
$$\leq \frac{T_1^2}{2}\exp\left\{\frac{-c_1^2(\mathbf{v}_j^*)^2}{\eta\mathcal{O}\left([\sigma^2 + \mathcal{M}^2(\mathbf{b})]\log^4(MT_1/\delta)\right)}\right\}$$
$$\leq \frac{T_1^2}{2}\exp\left\{\frac{-c_1^2\min_{j \in [D]}(\mathbf{v}_j^*)^2}{\eta\mathcal{O}\left([\sigma^2 + \mathcal{M}^2(\mathbf{b})]\log^4(MT_1/\delta)\right)}\right\}$$

$$\leq \frac{\delta}{6N}. \tag{21}$$

□

Combining Lemma B.1 in **Part I** and Lemma B.2, Lemma B.3 in **Part II**, we have now completed the proof of Theorem B.1.

*Proof of Theorem B.1.* First, we notice that in the setting of Theorem B.1,

$$\eta \leq \frac{c_1^2 \bar{\sigma}_{\min}(D)}{\mathcal{O}\left([\sigma^2 + \mathcal{M}^2(\mathbf{b})] \log^4(MT_1/\delta)\right)}, \tag{22}$$

and

$$
\begin{cases}
\frac{\mathcal{O}\left([\sigma^2+\mathcal{M}^2(\mathbf{b})]\log^8(MT_1/\delta)-\min_{j\in[D]}\log(\mathbf{v}_j^0)\right)}{c_1^2\eta\sigma_{\min}(D)} \leq T_1 \leq \frac{\log^{-4}(MT_1/\delta)\log((M-D)T_1/\delta)}{\eta^2\bar{\sigma}_{\max}(D)\mathcal{O}([\sigma^2+\mathcal{M}^2(\mathbf{b})])}, & \text{if } M > D, \\
\frac{\mathcal{O}\left([\sigma^2+\mathcal{M}^2(\mathbf{b})]\log^8(MT_1/\delta)-\min_{j\in[D]}\log(\mathbf{v}_j^0)\right)}{c_1^2\eta\sigma_{\min}(D)} \leq T_1, & \text{otherwise .}
\end{cases}
\tag{23}
$$

satisfy all assumptions in Lemmas B.1-B.3. Thus we can use all results in Lemma B.1-B.3. B.1 yields $\mathbb{P}\{\bar{\mathbf{v}}^{T_1} > \mathbf{b}\} \leq \frac{\delta}{6}$. Lemma B.2 implies that $\mathbb{P}\{\min_{j\in[D]} \max_{t\leq T_1}(\bar{\mathbf{v}}_j^t - \frac{1-c_1/2}{1+c_1}\mathbf{b}_j) < 0 \bigwedge \bar{\mathbf{v}}^{T_1} \leq \mathbf{b}\} \leq \frac{\delta}{6}$. Combining Lemma B.1 and B.2, we have $\mathbb{P}\{\min_{j\in[D]} \max_{t\leq T_1}(\bar{\mathbf{v}}_j^t - \frac{1-c_1/2}{1+c_1}\mathbf{b}_j) < 0\} \leq \frac{\delta}{3}$. Moreover, Lemma B.3 indicates that $\mathbb{P}\{\min_{j\in[D]} \max_{t\leq T_1}(\bar{\mathbf{v}}_j^t - \frac{1-c_1/2}{1+c_1}\mathbf{b}_j) \geq 0 \bigwedge \min_{j\in[D]}(\bar{\mathbf{v}}_j^{T_1} - \frac{1-c_1}{1+c_1}\mathbf{b}_j) < 0\} \leq \frac{\delta}{6}$. Combining these results, we establish the final probability bound: $\mathbb{P}\{|\bar{\mathbf{v}}_{1:D}^{T_1} - \mathbf{v}_{1:D}^*| \leq \frac{c_1}{1+c_1}\mathbf{b}_{1:D} \bigwedge \bar{\mathbf{v}}_{D+1:M}^{T_1} \leq \mathbf{b}_{D+1:M}\} \geq 1 - \frac{2}{3}\delta$, and this bound can be extended to $\mathbf{q}^{T_1}$ by the definition of capped coupling process in Definition B.1. By Proposition B.1, we complete the proof. □

### B.3 PROOF OF PHASE II

In this section, we introduce the proof techniques of **Phase II** in Theorem B.2, where we construct the global convergence analysis of Algorithm 1 for risk minimization. We demonstrate that after Phase I (i.e., $t > T_1$), the iterations of $\mathbf{v}^t$ are confined within a neighborhood of $\mathbf{v}^*$ with high probability. Therefore, the SGD dynamics for the quadratic model can be well approximated by the dynamics for the linear model with high probability. Therefore, we can extend the analytical techniques for SGD in the linear model to obtain the conclusion of Theorem B.2.

Theorem B.1 illustrates that the output of Algorithm 1 after $T_1$ iterations lies in the neighborhood of the ground truth within a constant factor, namely, $|\mathbf{v}_{1:D} - \mathbf{v}_{1:D}^*| \leq c_1\mathbf{v}_{1:D}^*$. Thus, we use $\mathbf{v}^{T_1}$, which satisfies equation 11, as the initial point for the SGD iterations in **Phase II**, and set the annealing learning rate to guarantee the output of Algorithm 1 fully converges to $\mathbf{v}^*$. Before we formal propose Theorem B.2, we preliminarily introduce some of the coupling process, auxiliary function, and notations used for our statement of Theorem B.2 and analysis in Phase II. We introduce the truncated coupling $\{\widehat{\mathbf{v}}^t\}_{t=0}^{T_2}$ as follows:

$$
\begin{cases}
\widehat{\mathbf{v}}^{t+1} = \mathbf{v}^{T_1+t+1}, & \text{if } \mathcal{G}(\widehat{\mathbf{v}}^t) \text{ occurs },\\
\widehat{\mathbf{v}}^{\tau+1} = \frac{13}{4}\mathbf{v}^*, & \forall\tau \geq t, \quad \text{otherwise },
\end{cases}
$$

with initialization $\widehat{\mathbf{v}}^0 = \mathbf{v}^{T_1}$ which satisfies equation 11, where event

$$\mathcal{G}(\mathbf{v}) := \left\{\mathbf{v}_j \in \left[\frac{1}{2}\mathbf{v}_j^*, \frac{3}{2}\mathbf{v}_j^*\right], \ \forall j \in [D] \bigwedge \mathbf{v}_j \in \left[0, 2\mathbf{v}_j^*\right], \ \forall j \in [D+1 : M]\right\}, \tag{24}$$

for any $\mathbf{v} \in \mathbb{R}^M$ and $T_2 = T - T_1$. Moreover, we define the auxiliary function $\psi : \mathbb{R}^M \to \mathbb{R}^M$ as:

$$\psi(\mathbf{v}) = \begin{cases} \mathbf{v}, & \text{if } \mathcal{G}(\mathbf{v}) \text{ occurs },\\ \mathbf{v}^*, & \text{otherwise.} \end{cases}$$

Thus we construct the truncated sequence $\{\mathbf{w}^t = \psi(\widehat{\mathbf{v}}^t)\}_{t=0}^{T_2}$. In this phase, our analysis primarily focuses on the trajectory of $\mathbf{w}^t$. Based on the generation mechanism of the sequence $\{\mathbf{w}^t\}_{t=0}^{T_2}$, the update from $\mathbf{w}^t$ to $\mathbf{w}^{t+1}$ can be categorized into two cases: Case I) $\mathbf{w}^{t+1}$ remains updated, with its iteration closely approximating SGD updates in linear models (Wu et al., 2022); Case II) For any $\tau \geq t$, $\mathbf{w}^{\tau+1}$ does not update and remains constant at $\mathbf{v}^*$.

We also define some notations for simplifying the representation. For any $\mathbf{v}, \mathbf{u} \in \mathbb{R}^d$, we define $\mathbf{v} \odot \mathbf{u} = (\mathbf{v}_1 \mathbf{u}_1, \cdots, \mathbf{v}_d \mathbf{u}_d)^\top$ and $\mathrm{diag}\{\mathbf{v}\} = \mathrm{diag}\{\mathbf{v}_1, \cdots, \mathbf{v}_d\} \in \mathbb{R}^{d \times d}$. Let $\mathbf{H} = \frac{25}{4}\mathbf{\Lambda}\,\mathrm{diag}\{\widehat{\mathbf{b}} \odot \widehat{\mathbf{b}}\}$ with $\widehat{\mathbf{b}}^\top = ((\mathbf{v}_{1:D}^*)^\top, (\widehat{\mathbf{v}}_{D+1:M}^*)^\top)$ and $\hat{\mathbf{v}}^*$ satisfies $\hat{\mathbf{v}}_j^* = \max\left\{\frac{3}{2}\mathbf{v}_j^*, 3\mathbf{v}_j^0\right\}$. We also denote $\mathbf{H}_{\mathbf{w}}^t = (\mathbf{w}^t \odot \mathbf{x}^{t+1}) \otimes ((\mathbf{w}^t + \mathbf{v}^*) \odot \mathbf{x}^{t+1})$ and $\mathbf{R}_{\mathbf{w}}^t = (\xi^{t+1} + \zeta_M^{t+1})\,\mathrm{diag}\{\mathbf{w}^t\}$ for simplicity. We denote the following linear operators that will be used in the proof:

$$\mathcal{I} := \mathbf{I} \otimes \mathbf{I}, \quad \mathcal{H}_{\mathbf{w}}^t := \mathbb{E}_t\left[\mathbf{H}_{\mathbf{w}}^t \otimes (\mathbf{H}_{\mathbf{w}}^t)^\top\right], \quad \widetilde{\mathcal{H}}_{\mathbf{w}}^t := \mathbb{E}_t\left[\mathbf{H}_{\mathbf{w}}^t\right] \otimes \mathbb{E}_t\left[\mathbf{H}_{\mathbf{w}}^t\right],$$

$$\mathcal{G}_{\mathbf{w}}^t := \mathbb{E}_t\left[\mathbf{H}_{\mathbf{w}}^t\right] \otimes \mathbf{I} + \mathbf{I} \otimes \mathbb{E}_t\left[\mathbf{H}_{\mathbf{w}}^t\right] - \eta_t \mathcal{H}_{\mathbf{w}}^t, \quad \widetilde{\mathcal{G}}_{\mathbf{w}}^t := \mathbb{E}_t\left[\mathbf{H}_{\mathbf{w}}^t\right] \otimes \mathbf{I} + \mathbf{I} \otimes \mathbb{E}_t\left[\mathbf{H}_{\mathbf{w}}^t\right] - \eta_t \widetilde{\mathcal{H}}_{\mathbf{w}}^t,$$

where $\mathbb{E}_t[\cdot] = \mathbb{E}[\cdot \mid \mathcal{F}^t]$. For any operator $\mathcal{A}$, we use $\mathcal{A} \circ \mathbf{A}$ to denote $\mathcal{A}$ acting on a symmetric matrix $\mathbf{A}$. It's easy to directly verify the following rules for above operators acting on a symmetric matrix $\mathbf{A}$:

$$\mathcal{I} \circ \mathbf{A} = \mathbf{A}, \quad \mathcal{H}_{\mathbf{w}}^t \circ \mathbf{A} = \mathbb{E}_t\left[\mathbf{H}_{\mathbf{w}}^t \mathbf{A}(\mathbf{H}_{\mathbf{w}}^t)^\top\right], \quad \widetilde{\mathcal{H}}_{\mathbf{w}}^t \circ \mathbf{A} = \mathbb{E}_t\left[\mathbf{H}_{\mathbf{w}}^t\right]\mathbf{A}\mathbb{E}_t\left[\mathbf{H}_{\mathbf{w}}^t\right],$$

$$\left(\mathcal{I} - \eta_t \mathcal{G}_{\mathbf{w}}^t\right) \circ \mathbf{A} = \mathbb{E}_t\left[\left(\mathbf{I} - \eta_t \mathbf{H}_{\mathbf{w}}^t\right)\mathbf{A}\left(\mathbf{I} - \eta_t \mathbf{H}_{\mathbf{w}}^t\right)\right],$$

$$\left(\mathcal{I} - \eta_t \widetilde{\mathcal{G}}_{\mathbf{w}}^t\right) \circ \mathbf{A} = \left(\mathbf{I} - \eta_t \mathbb{E}_t\left[\mathbf{H}_{\mathbf{w}}^t\right]\right)\mathbf{A}\left(\mathbf{I} - \eta_t \mathbb{E}_t\left[\mathbf{H}_{\mathbf{w}}^t\right]\right).$$

The following is the formalized expression of the iteration process for $\mathbf{w}^t$. For all $t \in [0 : T_2 - 1]$, if $\mathbf{w}^{t+1} = \mathbf{v}^{T_1+t+1}$ (i.e., event $\mathcal{G}(\mathbf{v}^{T_1+t+1})$ occurs), $\mathbf{w}^{t+1}$ follows the update rule as:

$$\mathbf{w}^{t+1} - \mathbf{v}^* = \mathbf{w}^t - \mathbf{v}^* - \eta_t \mathbf{H}_{\mathbf{w}}^t\left(\mathbf{w}^t - \mathbf{v}^*\right) + \eta_t \mathbf{R}_{\mathbf{w}}^t \mathbf{x}^t. \tag{25}$$

Otherwise, we have

$$\mathbf{w}^{\tau+1} = \mathbf{v}^*, \quad \forall \tau \geq t.$$

Since $\mathbf{w}^{t+1} = \mathbf{v}^{T_1+t+1}$ implies $\mathbf{w}^t = \mathbf{v}^{T_1+t}$, but the converse does not necessarily hold, we derive the recurrence process as:

$$\mathbb{E}\left[\left(\mathbf{w}^{t+1} - \mathbf{v}^*\right)^{\otimes 2}\right] \preceq \mathbb{E}\left[\left(\mathbf{w}^t - \mathbf{v}^* - \eta_t \mathbf{H}_{\mathbf{w}}^t\left(\mathbf{w}^t - \mathbf{v}^*\right) + \eta_t \mathbf{R}_{\mathbf{w}}^t \mathbf{x}^t\right)^{\otimes 2} \mathbb{1}_{\mathbf{w}^t = \mathbf{v}^{T_1+t}}\right].$$

Define $\widehat{\mathbf{w}}^t := \mathbf{w}^t - \mathbf{v}^*$. The iterative update of $\widehat{\mathbf{w}}^t$ can be decomposed into two random processes,

$$\widehat{\mathbf{w}}^t = \mathbb{1}_{\mathbf{w}^t = \mathbf{v}^{T_1+t}} \cdot \widehat{\mathbf{w}}_{\mathrm{bias}}^t + \mathbb{1}_{\mathbf{w}^t = \mathbf{v}^{T_1+t}} \cdot \widehat{\mathbf{w}}_{\mathrm{variance}}^t, \quad \forall t \in [0 : T_2], \tag{26}$$

where $\{\widehat{\mathbf{w}}_{\mathrm{variance}}^t\}_{t=1}^{T_2}$ is recursively defined by

$$\begin{cases} \widehat{\mathbf{w}}_{\mathrm{variance}}^{t+1} = (\mathbf{I} - \eta_t \mathbf{H}_{\mathbf{w}}^t)\,\widehat{\mathbf{w}}_{\mathrm{variance}}^t + \eta_t \mathbf{R}_{\mathbf{w}}^t \mathbf{x}^t, & \text{if } \mathbf{w}^t = \mathbf{v}^{T_1+t}, \\ \widehat{\mathbf{w}}_{\mathrm{variance}}^{t+1} = \mathbf{0}, & \text{otherwise}, \end{cases}$$

for any $t \in [0 : T_2 - 1]$ with $\widehat{\mathbf{w}}_{\mathrm{variance}}^0 = \mathbf{0}$ and $\{\widehat{\mathbf{w}}_{\mathrm{bias}}^t\}_{t=1}^{T_2}$ is recursively defined by

$$\begin{cases} \widehat{\mathbf{w}}_{\mathrm{bias}}^{t+1} = (\mathbf{I} - \eta_t \mathbf{H}_{\mathbf{w}}^t)\,\widehat{\mathbf{w}}_{\mathrm{bias}}^t, & \text{if } \mathbf{w}^t = \mathbf{v}^{T_1+t}, \\ \widehat{\mathbf{w}}_{\mathrm{bias}}^{t+1} = \mathbf{0}, & \text{otherwise}, \end{cases}$$

for any $t \in [0 : T_2 - 1]$ with $\widehat{\mathbf{w}}_{\mathrm{bias}}^0 = \mathbf{w}^0 - \mathbf{v}^*$. We define the $t$-th step bias iteration as $\mathbf{B}^t = \mathbb{E}\left[\widehat{\mathbf{w}}_{\mathrm{bias}}^t \otimes \widehat{\mathbf{w}}_{\mathrm{bias}}^t\right]$ and $t$-th step variance iteration as $\mathbf{V}^t = \mathbb{E}\left[\widehat{\mathbf{w}}_{\mathrm{variance}}^t \otimes \widehat{\mathbf{w}}_{\mathrm{variance}}^t\right]$. Therefore, we can derive the following relations for $\{\mathbf{B}^t\}_{t=0}^{T_2}$ and $\{\mathbf{V}^t\}_{t=0}^{T_2}$:

$$\begin{cases} \mathbf{B}^{t+1} & \preceq \mathbb{E}\left[(\mathcal{I} - \eta_t \mathcal{G}_{\mathbf{w}}^t) \circ (\widehat{\mathbf{w}}_{\mathrm{bias}}^t \otimes \widehat{\mathbf{w}}_{\mathrm{bias}}^t)\right], \\ \mathbf{V}^{t+1} & \preceq \mathbb{E}\left[(\mathcal{I} - \eta_t \mathcal{G}_{\mathbf{w}}^t) \circ (\widehat{\mathbf{w}}_{\mathrm{variance}}^t \otimes \widehat{\mathbf{w}}_{\mathrm{variance}}^t)\right] + \eta_t^2 \mathbf{\Sigma}_{\mathbf{w}}^t, \end{cases} \quad \forall t \in [0 : T_2 - 1], \tag{27}$$

with $\mathbf{B}^0 = \left(\mathbf{w}^0 - \mathbf{v}^*\right)\left(\mathbf{w}^0 - \mathbf{v}^*\right)^\top$ and $\mathbf{V}^0 = \mathbf{0}$, where $\mathbf{\Sigma}_{\mathbf{w}}^t = \sigma^2 \mathbf{\Lambda} \mathbb{E}\left[\mathrm{diag}\{\mathbf{w}^t \odot \mathbf{w}^t\}\right]$.

We formally propose Theorem B.2 as below.

**Theorem B.2.** *[Formal version of Theorem 5.2] Suppose Assumption 3.1 and 3.3 hold, and let $T_1 = \lceil (T-h)/\log(T-h) \rceil$ and $h = \lceil T/\log(T) \rceil$. Under the following setting*

1. *There exists $D < M$ such that $\eta_0 \leq \widetilde{\Omega}(\min\{\mathrm{tr}^{-1}(\mathbf{H}), \bar{\sigma}_{\min}(D)\})$ and $T_1 = \widetilde{\mathcal{O}}(\frac{\sigma^2 + \mathcal{M}^2(\widehat{\mathbf{b}})}{\eta_0 \sigma_{\min}(D)})$,*

2. *Let $D = M$, $\eta_0 \leq \widetilde{\Omega}(\min\{\mathrm{tr}^{-1}(\mathbf{H}), \bar{\sigma}_{\min}(M)\})$, and $T_1 \geq \widetilde{\mathcal{O}}(\frac{\sigma^2 + \mathcal{M}^2(\widehat{\mathbf{b}})}{\eta_0 \sigma_{\min}(M)})$,*

*we have*

$$
\mathbb{E}\left[\mathcal{R}_M(\mathbf{w}^{T_2}) - \mathcal{R}_M(\mathbf{v}^*)\right] \lesssim \sigma^2 \left( \frac{N_0'}{K} + \eta_0 \sum_{i=N_0'+1}^{N_0} \lambda_i(\mathbf{v}_i^*)^2 \right)
$$
$$
+ \sigma^2 \eta_0^2 (h + T_1) \sum_{i=N_0+1}^{M} \lambda_i^2(\widehat{\mathbf{b}}_i^*)^4
$$
$$
+ \left\langle \frac{1}{\eta_0 T_1} \mathbf{I}_{1:N_1} + \mathbf{H}_{N_1+1:M}, \left(\mathbf{I} - \eta_0 \widehat{\mathbf{H}}\right)^{2h} \mathbf{B}^0 \right\rangle
$$
$$
+ \Gamma(\mathbf{H}) \left\langle \frac{1}{\eta_0 h} \mathbf{I}_{1:N_1'} + \mathbf{H}_{N_1'+1:M}, \mathbf{B}^0 \right\rangle, \tag{28}
$$

*for arbitrary $D \geq N_0 \geq N_0' \geq 0$ and $D \geq N_1 \geq N_1' \geq 0$, where $\Gamma(\mathbf{H}) := (\frac{625 N_1'}{T_1} + \frac{25 \eta_0 h}{T_1} \mathrm{tr}(\mathbf{H}_{N_1'+1:N_1}) + \eta_0^2 h \, \mathrm{tr}(\mathbf{H}_{N_1+1:M}^2))$ and $T_2 = T - T_1$. Specially, we have*

$$
\mathcal{R}_M(\mathbf{v}^T) - \mathcal{R}_M(\mathbf{v}^*) \lesssim \frac{\sigma^2 N}{T_1} + \sigma^2 \eta_0^2 (h + T_1) \sum_{i=D+1}^{M} \lambda_i^2(\widehat{\mathbf{b}}_i^*)^4
$$
$$
+ \left\langle \frac{1}{\eta_0 T_1} \mathbf{I}_{1:D} + \mathbf{H}_{D+1:M}, \left(\mathbf{I} - \eta_0 \widehat{\mathbf{H}}\right)^{2h} \mathbf{B}^0 \right\rangle \tag{29}
$$
$$
+ \left( \frac{D}{T_1} + \eta_0^2 h \, \mathrm{tr}(\mathbf{H}_{D+1:M}^2) \right) \left\langle \frac{1}{\eta_0 h} \mathbf{I}_{1:D} + \mathbf{H}_{D+1:M}, \mathbf{B}^0 \right\rangle,
$$

*with probability at least 0.95.*

Before the beginning of our proof, we define the $(c\mathbf{v}_{1:D}^*, \mathbf{b})$-neighbor coupling process which will be used in the following lemma as below.

**Definition B.2.** *[$(c\mathbf{v}_{1:D}^*, \mathbf{b})$-neighbor coupling] Let $\{\mathbf{q}^t\}_{t=0}^T$ be a Markov chain in $\mathbb{R}_+^M$ adapted to filtration $\{\mathcal{F}^t\}_{t=0}^T$. Given parameters: 1) Dimension index $D \in \mathbb{Z}_+$; 2) Tolerance $c > 0$; 3) Threshold vector $\mathbf{b} \in \mathbb{R}_+^{M-D}$. With initial condition $\bar{\mathbf{v}}^0 = \mathbf{q}^0$, $\left|\bar{\mathbf{v}}_{1:D}^0 - \mathbf{v}_{1:D}^*\right| \leq c\mathbf{v}_{1:D}^*$ and $0 \leq \bar{\mathbf{v}}_{D+1:M}^0 \leq \mathbf{b}$, the $(c\mathbf{v}_{1:D}^*, \mathbf{b})$-neighbor coupling process $\{\bar{\mathbf{v}}^t\}_{t=0}^T$ evolves as:*

1. *Updating state: If $\left|\bar{\mathbf{v}}_{1:D}^t - \mathbf{v}_{1:D}^*\right| \leq c\mathbf{v}_{1:D}^*$ and $0 \leq \bar{\mathbf{v}}_{D+1:M}^t \leq \mathbf{b}$, let $\bar{\mathbf{v}}^{t+1} = \mathbf{v}^{t+1}$,*

2. *Absorbing state: Otherwise, maintain $\bar{\mathbf{v}}^{t+1} = \bar{\mathbf{v}}^t$.*

### B.3.1 PART I: BOUND THE OUTPUT OF PHASE I

In this part, we demonstrate that the output of **Phase I** remains confirmed within the neighborhood of the ground truth with high probability in Lemma B.3. Specifically, by constructing similar supermartingales to that in the proofs of Lemma B.1 and Lemma B.3, we obtain a set of compressed supermartingales dependent on the coordinate $i \in [M]$. Combining the compression properties of these supermartingales with the sub-Gaussian property of their difference sequences, through concentration inequality, we obtain Lemma B.3 as below.

**Theorem B.3.** *[Formal version of Theorem A.1] Under Assumption 3.1, we consider the $T_1$-th step of Algorithm 1 and its subsequent iterative process. Let $D \in \mathbb{N}+$ represent the effective dimension.*

*Define $\eta_0 \leq \widetilde{\Omega}\left(\frac{\bar{\sigma}_{\min}(\max\{D,M\})}{\sigma^2 + \mathcal{M}^2(\mathbf{b})}\right)$, and let $\{\widetilde{\mathbf{v}}^t\}_{t=0}^{T_2}$ be an $(1/2, 2)$-$\mathbf{v}^*$ neighbor coupling process based on the control sequence $\{\mathbf{q}^{T_1+t}\}_{t=0}^{T_2}$. Recall the definition equation 24 of event $\mathcal{G}(\mathbf{v})$ for any $\mathbf{v} \in \mathbb{R}^M$. If $D < M$, set the iteration number $T_2 \in \left[\widetilde{\Omega}\left(\frac{\widetilde{\sigma}_{\max}^{-1}(D)}{\eta_0^2[\sigma^2 + \mathcal{M}^2(\mathbf{b})]}\right)\right]$. Otherwise, set $T_2$ be an arbitrary positive integer. Then, $\bigcap_{t=0}^{T_2} \mathcal{G}(\mathbf{v}^{T_1+t})$ holds with probability at least $1 - \delta$.*

*Proof.* Setting $c_1 = \frac{1}{4}$ in Theorem B.1, we have $|\mathbf{q}_{1:D}^{T_1} - \mathbf{v}_{1:D}^*| \leq \frac{1}{4}\mathbf{v}_{1:D}^*$ and $\mathbf{0}_{D+1:M} \leq \mathbf{q}_{D+1:M}^{T_1} \leq \frac{3}{2}\mathbf{v}_{D+1:M}^*$ with probability at least $1 - \delta/6$. Without loss of generality, we assume $\mathbf{q}^{T_1}$ satisfies $|\mathbf{q}_{1:D}^{T_1} - \mathbf{v}_{1:D}^*| \leq \frac{1}{4}\mathbf{v}_{1:D}^*$ and $\mathbf{0}_{D+1:M} \leq \mathbf{q}_{D+1:M}^{T_1} \leq \frac{3}{2}\mathbf{v}_{D+1:M}^*$. Let $\hat{\tau}$ be the stopping time satisfying $\mathcal{G}^c(\widetilde{\mathbf{v}}^{\hat{\tau}})$, i.e.,

$$\hat{\tau} = \inf\left\{t : \exists j \in [D], \text{ s.t. } \left|\widetilde{\mathbf{v}}_j^t - \mathbf{v}_j^*\right| > \frac{1}{2}\mathbf{v}_j^* \text{ or } \exists j \in [D+1 : M], \text{ s.t. } \widetilde{\mathbf{v}}_j^t > 2\mathbf{v}_j^*\right\},$$

For each coordinate $j \in [D]$, let $\hat{\tau}_{[D],j}^u$ and $\hat{\tau}_{[D],j}^l$ be the stopping time satisfying $\widetilde{\mathbf{v}}_j^{\hat{\tau}_{[D],j}^u} > \frac{3}{2}\mathbf{v}_j^*$ and $\widetilde{\mathbf{v}}_j^{\hat{\tau}_{[D+1:M],j}^l} < \frac{1}{2}\mathbf{v}_j^*$, respectively, i.e.,

$$\hat{\tau}_{[D],j}^u = \inf\left\{t : \widetilde{\mathbf{v}}_j^t > \frac{3}{2}\mathbf{v}_j^*\right\}, \quad \hat{\tau}_{[D],j}^l = \inf\left\{t : \widetilde{\mathbf{v}}_j^t < \frac{1}{2}\mathbf{v}_j^*\right\}.$$

For each coordinate $j \in [D+1 : M]$, let $\hat{\tau}_{[D+1:M],j}$ be the stopping time satisfying $\widetilde{\mathbf{v}}_j^{\hat{\tau}_{[D+1:M],j}} > 2\mathbf{v}_j^*$, i.e.,

$$\hat{\tau}_{[D+1:M],j} = \inf\left\{t : \widetilde{\mathbf{v}}_j^t > 2\mathbf{v}_j^*\right\}.$$

Based on Defnition B.2, once the stopping time $\hat{\tau} = t_2$ occurs for certain $t_2 \in [T_2]$, the coupling process satisfies $\widetilde{\mathbf{v}}^t = \widetilde{\mathbf{v}}^{t_2}$ for all $t > t_2$. Suppose there exists a certain $j \in [D]$ such that $\hat{\tau}_{[D],j}^u = t_2$. Thus, the event $\mathcal{G}(\widetilde{\mathbf{v}}^t)$ holds for all $t \in [0 : t_2 - 1]$. Similar to the proof of Lemma B.1 and B.3, $\widetilde{\mathbf{v}}_j^t$ must traverse in and out of the threshold interval $[\mathbf{v}_j^*, \frac{3}{2}\mathbf{v}_j^*]$ before exceeding $\frac{3}{2}\mathbf{v}_j^*$. We aim to estimate the following probability for coordinates $j \in [D]$ and time pairs $t_1 < t_2 \in [0 : T_2]$ as:

$$\mathbb{P}\left(\mathcal{B}_{t_1}^{\hat{\tau}_{[D],j}^u = t_2}(j) = \left\{\widetilde{\mathbf{v}}_j^{t_1} \leq \frac{5}{4}\mathbf{v}_j^* \bigwedge \widetilde{\mathbf{v}}_j^{t_1:t_2-1} \in \left[\mathbf{v}_j^*, \frac{3}{2}\mathbf{v}_j^*\right]\right\}\right).$$

For any $t \in [t_1 : t_2 - 1]$, we have

$$\mathbb{E}\left[\widetilde{\mathbf{v}}_j^{t+1} - \mathbf{v}_j^* \mid \mathcal{F}^t\right] = \mathbb{E}_{\mathbf{x}_{1:M}^{t+1}, \xi^{t+1}, \zeta_M^{t+1}}\left[\widetilde{\mathbf{v}}_j^t - \mathbf{v}_j^* - \eta\left(((\widetilde{\mathbf{v}}_j^t)^2 - (\mathbf{v}_j^*)^2)\hat{\mathbf{x}}_j^{t+1} + \hat{\mathbf{z}}_j^{t+1}(\widetilde{\mathbf{v}}^t)\right.\right.$$
$$\left.\left. - \widehat{\zeta}_M^{t+1} - \hat{\xi}^{t+1}\right)\hat{\mathbf{x}}_j^{t+1}\widetilde{\mathbf{v}}_j^*\right] \quad (30)$$
$$\leq \left(1 - \frac{3\eta_t}{8}\lambda_j(\mathbf{v}_j^*)^2\right)(\widetilde{\mathbf{v}}_j^t - \mathbf{v}_j^*).$$

Applying Lemma E.1 to $(((\widetilde{\mathbf{v}}_j^t)^2 - (\mathbf{v}_j^*)^2)\hat{\mathbf{x}}_j^{t+1} + \hat{\mathbf{z}}_j^{t+1}(\widetilde{\mathbf{v}}^t) - \widehat{\zeta}_M^{t+1} - \hat{\xi}^{t+1})\hat{\mathbf{x}}_j^{t+1}\widetilde{\mathbf{v}}_j^{t+1}$, we obtain

$$\mathbb{E}\left[\exp\left\{\lambda\left(\widetilde{\mathbf{v}}_j^{t+1} - \mathbb{E}\left[\widetilde{\mathbf{v}}_j^{t+1} \mid \mathcal{F}^t\right]\right)\right\} \mid \mathcal{F}^t\right] \leq \exp\left\{\frac{\lambda^2 \eta_t^2 \lambda_j(\mathbf{v}_j^*)^2 \mathcal{O}\left([\sigma^2 + \mathcal{M}^2(\mathbf{b})]\log^4(MT_2/\delta)\right)}{2}\right\},$$

for any $\lambda \in \mathbb{R}$. Therefore, based on Lemma E.3 and equation 30, we establish the probability bound for event $\mathcal{B}_{t_1}^{\hat{\tau}_{[D],j}^u = t_2}(j)$ for any time pair $t_1 < t_2 \in [0 : T_2]$ as:

$$\mathbb{P}\left\{\mathcal{B}_{t_1}^{\hat{\tau}_{[D],j}^u = t_2}(j)\right\} \leq \exp\left\{-\frac{(\mathbf{v}_j^*)^2}{V_j}\right\}, \quad (31)$$

where $V_j$ is denoted as

$$V_j = \lambda_j(\mathbf{v}_j^*)^2 \mathcal{O}\left([\sigma^2 + \mathcal{M}^2(\mathbf{b})]\log^4(MT_2/\delta)\right)\sum_{t=0}^{T_2-1}\left(\prod_{i=t+1}^{T_2-1}(1 - \frac{3\eta_i}{4}\lambda_j(\mathbf{v}_j^*)^2)^2\right)(\eta_t)^2.$$

By Lemma E.4, we have $V_j \leq \mathcal{O}(\eta_0[\sigma^2 + \mathcal{M}^2(\mathbf{b})]\log^4(MT_2/\delta))$. Therefore, using equation 31, we can derive

$$\mathbb{P}\left\{\mathcal{B}_{t_1}^{\hat{\tau}_{[D],j}^u=t_2}(j)\right\} \leq \exp\left\{-\frac{(\mathbf{v}_j^*)^2}{\eta_0\mathcal{O}\left([\sigma^2 + \mathcal{M}^2(\mathbf{b})]\log^4(MT_2/\delta)\right)}\right\}. \tag{32}$$

Similarly, suppose there exists a certain $j \in [D]$ such that $\hat{\tau}_{[D],j}^l = t_2$. Thus, the event $\mathcal{G}(\widetilde{\mathbf{v}}^t)$ holds for all $t \in [0 : t_2 - 1]$. $\widetilde{\mathbf{v}}_j^t$ must traverse in and out of the threshold interval $[\frac{1}{2}\mathbf{v}_j^*, \mathbf{v}_j^*]$ before subceeding $\frac{1}{2}\mathbf{v}_j^*$. We aim to estimate the following probability for coordinates $j \in [D]$ and time pairs $t_1 < t_2 \in [T_2]$:

$$\mathbb{P}\left(\mathcal{C}_{t_1}^{\hat{\tau}_{[D],j}^l=t_2}(j) = \left\{\widetilde{\mathbf{v}}_j^{t_1} \geq \frac{3}{4}\mathbf{v}_j^* \bigwedge \widetilde{\mathbf{v}}_j^{t_1:t_2-1} \in \left[\frac{1}{2}\mathbf{v}_j^*, \mathbf{v}_j^*\right]\right\}\right).$$

For any $t \in [t_1 : t_2 - 1]$, we have

$$\mathbb{E}\left[\mathbf{v}_j^* - \widetilde{\mathbf{v}}_j^{t+1} \mid \mathcal{F}^t\right] \leq \left(1 - \frac{3\eta_t}{8}\lambda_j(\mathbf{v}_j^*)^2\right)(\mathbf{v}_j^* - \widetilde{\mathbf{v}}_j^t).$$

Based on Lemmas E.1, E.3, and E.4 sequentially, we obtain the probability bound for event $\mathcal{C}_{t_1}^{\hat{\tau}_{[D],j}^l=t_2}(j)$ for any time pair $t_1 < t_2 \in [0 : T_2]$ as:

$$\mathbb{P}\left\{\mathcal{C}_{t_1}^{\hat{\tau}_{[D],j}^l=t_2}(j)\right\} \leq \exp\left\{-\frac{(\mathbf{v}_j^*)^2}{V_j}\right\} \leq \exp\left\{-\frac{(\mathbf{v}_j^*)^2}{\eta_0\mathcal{O}\left([\sigma^2 + \mathcal{M}^2(\mathbf{b})]\log^4(MT_2/\delta)\right)}\right\}. \tag{33}$$

For the third stopping time, we also suppose there exists a certain $j \in [D+1 : M]$ such that $\hat{\tau}_{[D+1:M],j} = t_2$. Thus, the event $\mathcal{G}(\widetilde{\mathbf{v}}^t)$ holds for all $t \in [0 : t_2 - 1]$. Similarly, $\widehat{\mathbf{v}}_j^t$ must traverse in and out of the threshold interval $[\mathbf{v}_j^*, 2\mathbf{v}_j^*]$ before exceeding $2\mathbf{v}_j^*$. We aim to estimate the following probability for coordinates $j \in [D+1 : M]$ and time pairs $t_1 < t_2 \in [0 : T_2]$ as:

$$\mathbb{P}\left(\mathcal{D}_{t_1}^{\hat{\tau}_{[D+1:M],j}=t_2}(j) = \left\{\widetilde{\mathbf{v}}_j^{t_1} \leq \frac{3}{2}\mathbf{v}_j^* \bigwedge \widetilde{\mathbf{v}}_j^{t_1:t_2-1} \in [\mathbf{v}_j^*, 2\mathbf{v}_j^*]\right\}\right).$$

For any $t \in [t_1 : t_2 - 1]$, we have

$$\mathbb{E}\left[\widetilde{\mathbf{v}}_j^{t+1} - \mathbf{v}_j^* \mid \mathcal{F}^t\right] \leq \widetilde{\mathbf{v}}_j^t - \mathbf{v}_j^*. \tag{34}$$

Applying Lemma E.1 to $(((\widetilde{\mathbf{v}}_j^t)^2 - (\mathbf{v}_j^*)^2)\hat{\mathbf{x}}_j^{t+1} + \hat{\mathbf{z}}_j^{t+1}(\widetilde{\mathbf{v}}^t) - \hat{\zeta}_M^{t+1} - \hat{\xi}^{t+1})\hat{\mathbf{x}}_j^{t+1}\widetilde{\mathbf{v}}_j^{t+1}$, we obtain

$$\mathbb{E}\left[\exp\left\{\lambda\left(\widetilde{\mathbf{v}}_j^{t+1} - \mathbb{E}\left[\widetilde{\mathbf{v}}_j^{t+1} \mid \mathcal{F}^t\right]\right)\right\} \mid \mathcal{F}^t\right] \leq \exp\left\{\frac{\lambda^2\eta_t^2\lambda_j(\mathbf{v}_j^*)^2\mathcal{O}\left([\sigma^2 + \mathcal{M}^2(\mathbf{b})]\log^4(MT_2/\delta)\right)}{2}\right\},$$

for any $\lambda \in \mathbb{R}$. Based on Lemma E.3 and equation 34, we establish the probability bound for the event $\mathcal{D}_{t_1}^{\hat{\tau}_{[D+1:M],j}=t_2}(j)$ for any time pair $t_1 < t_2 \in [0 : T_2]$ as:

$$\mathbb{P}\left\{\mathcal{D}_{t_1}^{\hat{\tau}_{[D+1:M],j}=t_2}(j)\right\} \leq \exp\left\{-\frac{(\mathbf{v}_j^*)^2}{V_j}\right\} \overset{(a)}{\leq} \exp\left\{-\frac{\log^{-4}(MT_2/\delta)}{T_2\eta_0^2\lambda_j\mathcal{O}\left([\sigma^2 + \mathcal{M}^2(\mathbf{b})]\right)}\right\}, \tag{35}$$

where (a) is derived from $V_j \leq T_2\eta_0^2\lambda_j\mathcal{O}\left([\sigma^2 + \mathcal{M}^2(\mathbf{b})]\log^4(MT_2/\delta)\right)$.

Then, it is easy to notice that $\mathcal{G}^c(\widetilde{\mathbf{v}}^{T_2})$ indicates that one of the following situation happens:

1. For a certain coordinate $j \in [D]$ and time pairs $t_1 < t_2 \in [0 : T_2]$, either $\mathcal{B}_{t_1}^{\hat{\tau}_{[D],j}^u=t_2}(j)$ or $\mathcal{C}_{t_1}^{\hat{\tau}_{[D],j}^l=t_2}(j)$ occurs,

2. For a certain coordinate $j \in [D]$ and time pairs $t_1 < t_2 \in [0 : T_2]$, $\mathcal{D}_{t_1}^{\hat{\tau}_{[D+1:M],j}=t_2}(j)$ occurs.

Therefore, by the setting of $\eta_0$ in Lemma B.3, we derive the following probability bound of event $\mathcal{G}^c(\widetilde{\mathbf{v}}^{T_2})$:

$$\mathbb{P}\{\mathcal{G}^c(\widetilde{\mathbf{v}}^{T_2})\} \leq \sum_{t_1<t_2} \left[ \sum_{j\in[D]} \left( \mathbb{P}\left\{ \mathcal{B}_{t_1}^{\hat{\tau}_{[D],j}^u=t_2}(j) \right\} + \mathbb{P}\left\{ \mathcal{C}_{t_1}^{\hat{\tau}_{[D],j}^l=t_2}(j) \right\} \right) \right.$$

$$\left. + \sum_{j\in[D+1:M]} \mathbb{P}\left\{ \mathcal{D}_{t_1}^{\hat{\tau}_{[D+1:M],j}=t_2}(j) \right\} \right]$$

$$\leq 2T_2^2 N \exp\left\{ -\frac{\min_{j\in\mathcal{N}}(\mathbf{v}_j^*)^2}{\eta_0 \mathcal{O}\left([\sigma^2 + \mathcal{M}^2(\mathbf{b})]\log^4(MT_2)\right)} \right\}$$

$$+ T_2^2(\max\{M,D\}-D) \exp\left\{ -\frac{\log^{-4}(MT_2/\delta)}{T_2\eta_0^2 \max_{j\in\mathcal{N}} \lambda_j \mathcal{O}\left([\sigma^2 + \mathcal{M}^2(\mathbf{b})]\right)} \right\}$$

$$\leq \delta/2.$$

According to the construction of the coupling process $\{\widetilde{\mathbf{v}}^t\}_{t=0}^{T_2}$ in Definition B.2, we have $\bigcap_{t=T_1}^{T_1+T_2} \mathcal{G}(\mathbf{q}^t)$ holds with probability at least $1 - \delta/2$. By Proposition B.1, the proof is completed. $\qquad\square$

### B.3.2 PART II: LINEAR APPROXIMATION OF THE DYNAMIC

In part I, we have proved that $\bigcap_{t=0}^{T_2} \mathcal{G}(\mathbf{v}^{T_1+t})$ occurs with high probability, which implies the truncated sequence $\{\mathbf{w}^t\}_{t=1}^{T_2}$ aligned to $\{\mathbf{v}^{T_1+t}\}_{t=1}^{T_2}$ with high probability. Then we approximate the update process of $\{\mathbf{w}^t\}_{t=1}^{T_2}$ to SGD in traditional linear regression, with respective bounds of variance term and bias term.

We estimate the risk between the last-step function value and the ground truth as:

$$\mathbb{E}\left[\mathcal{R}_M(\mathbf{w}^{T_2}) - \mathcal{R}_M(\mathbf{v}^*)\right] \overset{\text{(a)}}{\leq} \left\langle \mathbf{H}, \mathbb{E}\left[\widehat{\mathbf{w}}^{T_2} \otimes \widehat{\mathbf{w}}^{T_2}\right]\right\rangle \leq 2\left\langle \mathbf{H}, \mathbf{B}^{T_2}\right\rangle + 2\left\langle \mathbf{H}, \mathbf{V}^{T_2}\right\rangle, \qquad (36)$$

where $\mathbf{H} = \frac{25}{4}\mathbf{\Lambda}\operatorname{diag}\{\widehat{\mathbf{b}} \odot \widehat{\mathbf{b}}\}$ and $\widehat{\mathbf{b}}^\top = \left((\mathbf{v}_{1:D}^*)^\top, (\widehat{\mathbf{v}}_{D+1:M}^*)^\top\right)$. Here, (a) is derived from combining

$$\mathbb{E}\left[\mathcal{R}_M(\mathbf{w}^{T_2}) - \mathcal{R}_M(\mathbf{v}^*)\right] = \mathbb{E}\left[\sum_{i=1}^M \lambda_i(\mathbf{w}_i^{T_2} + \mathbf{v}_i^*)^2(\mathbf{w}_i^{T_2} - \mathbf{v}_i^*)^2\right],$$

with the uniform boundedness of $\mathbf{w}^t$ over $t \in [0:T_2]$. According to the definitions of $\mathbf{w}^t$ and $\mathbf{H}_{\mathbf{w}}^t$, we have $\mathbb{E}[\mathbf{H}_{\mathbf{w}}^t] \preceq \mathbf{H}$. Use $\widehat{\mathbf{H}}$ to denote $\frac{1}{4}\mathbf{\Lambda}\operatorname{diag}\{\overline{\mathbf{b}} \odot \overline{\mathbf{b}}\}$ where $\overline{\mathbf{b}}^\top = \left((\mathbf{v}_{1:D}^*)^\top, \mathbf{0}^\top\right)$, and define $\widehat{\mathcal{G}} := \widehat{\mathbf{H}} \otimes \mathbf{I} + \mathbf{I} \otimes \widehat{\mathbf{H}} - \eta\widehat{\mathbf{H}} \otimes \widehat{\mathbf{H}}$. For simplicity, we let $K = T_1$. Moreover, we use $C$ to denote the constant such that $\mathbb{E}[|\mathbf{x}_i|^4] \leq C\mathbb{E}[|\mathbf{x}_i|^2]$ for any $i \geq 1$. Then we respectively bound the variance and bias to obtain the estimation of $\mathcal{R}_M(\mathbf{v}^{T_2}) - \mathcal{R}_M(\mathbf{v}^*)$.

**Bound of Variance**: Lemma B.4 provides a uniform upper bound for $\mathbf{V}^t$ over $t \in [0:T_2]$.

**Lemma B.4.** *Suppose Assumption 3.1 holds. Under the setting of Theorem B.2, for any $t \in [0:T_2]$, we obtain*

$$\mathbf{V}_{\text{diag}}^t \precsim \eta_0\sigma^2\mathbf{I}. \qquad (37)$$

*Proof.* The definition of $\mathbf{\Sigma}_{\mathbf{w}}^t$ and the boundedness of $\mathbf{w}^t$ implicate that $\mathbf{\Sigma}_{\mathbf{w}}^t \preceq \sigma^2\mathbb{E}[\mathbf{H}_{\mathbf{w}}^t] \preceq \mathbf{H}$ given $\mathbf{v}^* \geq \mathbf{0}$. The proof relies on induction. At $t = 0$, it follows that $\mathbf{V}_{\text{diag}}^0 = \mathbf{0} \precsim \eta_0\sigma^2\mathbf{I}$. Assuming $\mathbf{V}_{\text{diag}}^\tau \precsim \eta_0\sigma^2\mathbf{I}$ for any $\tau \leq t$, we proceed to estimate $\mathbf{V}^{t+1}$ by combining equation 27 as,

$$\mathbf{V}_{\text{diag}}^{t+1} \preceq \left(\mathbb{E}\left[(\mathcal{I} - \eta_t\mathcal{G}_{\mathbf{w}}^t) \circ (\widehat{\mathbf{w}}_{\text{variance}}^t \otimes \widehat{\mathbf{w}}_{\text{variance}}^t)\right]\right)_{\text{diag}} + \eta_t^2\mathbf{\Sigma}_{\mathbf{w}}^t$$

$$\preceq \left(\mathcal{I} - \eta_t\widehat{\mathbf{H}} \otimes \mathbf{I} - \eta_t\mathbf{I} \otimes \widehat{\mathbf{H}}\right) \circ \mathbf{V}_{\text{diag}}^t$$

$$+ \eta_t^2 \left( \mathbb{E} \left[ \mathcal{H}_{\mathbf{w}}^t \circ \left( \widehat{\mathbf{w}}_{\text{variance}}^t \otimes \widehat{\mathbf{w}}_{\text{variance}}^t \right) \right] \right)_{\text{diag}} + \eta_t^2 \sigma^2 \mathbf{H}$$

$$\stackrel{(a)}{\preceq} \left( \mathbf{I} - 2\eta_t \widehat{\mathbf{H}} \right) \mathbf{V}_{\text{diag}}^t + \mathcal{O} \left( \eta_t^2 (C+2) \langle \mathbf{H}, \mathbf{V}_{\text{diag}}^t \rangle \mathbf{H} + \eta_t^2 \sigma^2 \mathbf{H} \right)$$

$$\preceq \left( \mathbf{I} - 2\eta_t \widehat{\mathbf{H}} \right) \mathbf{V}_{\text{diag}}^t + \widetilde{\mathcal{O}} \left( \eta_t^2 \eta_0 \sigma^2 (C+2) \operatorname{tr}(\mathbf{H}) \mathbf{H} + \eta_t^2 \sigma^2 \mathbf{H} \right),$$

where (a) is derived from Lemma E.6 with $\mathbf{A} = \operatorname{diag}\{\mathbf{v}^* + \mathbf{w}^t\}$ and $\mathbf{B} = \widehat{\mathbf{w}}_{\text{variance}}^t \otimes \widehat{\mathbf{w}}_{\text{variance}}^t$. For $i \in [D]$, we have

$$\left( \mathbf{V}_{\text{diag}}^{t+1} \right)_{i,i} \le \left( 1 - 2\eta_t \widehat{\mathbf{H}}_{i,i} \right) \left( \mathbf{V}_{\text{diag}}^t \right)_{i,i} + \widetilde{\mathcal{O}} \left( \eta_t^2 \sigma^2 \widehat{\mathbf{H}}_{i,i} \right). \tag{38}$$

The recursion given by equation 38 implies that $(\mathbf{V}_{\text{diag}}^{t+1})_{i,i} \lesssim \eta_0 \sigma^2$ for any $i \in [D]$, using Lemma E.4. For $i \in [D+1:M]$, we obtain

$$\left( \mathbf{V}_{\text{diag}}^{t+1} \right)_{i,i} \lesssim \sigma^2 \mathbf{H}_{i,i} \sum_{k=0}^t \eta_k^2 \lesssim \eta_0 \sigma^2. \tag{39}$$

Therefore, we complete the induction. $\qquad\square$

**Lemma B.5.** *Suppose Assumption 3.1 holds. Under the setting of Theorem B.2, we have*

$$\langle \mathbf{H}, \mathbf{V}^{T_2} \rangle \lesssim \sigma^2 \left( \frac{N_0'}{K} + \eta_0 \sum_{i=N_0'+1}^{N_0} \lambda_i (\mathbf{v}_i^*)^2 \right) + \sigma^2 \eta_0^2 (h+K) \sum_{i=N_0+1}^M \lambda_i^2 (\widehat{\mathbf{b}}_i^*)^4, \tag{40}$$

*for arbitrary $D \ge N_0 \ge N_0' \ge 0$.*

*Proof.* Applying equation 27, we obtain

$$\mathbf{V}_{\text{diag}}^{t+1} \preceq \left( \mathcal{I} - \eta_t \widehat{\mathcal{G}} \right) \circ \mathbf{V}_{\text{diag}}^t + \eta_t^2 \left( \mathcal{H}_{\mathbf{w}}^t \circ \left( \widehat{\mathbf{w}}_{\text{variance}}^t \otimes \widehat{\mathbf{w}}_{\text{variance}}^t \right) \right)_{\text{diag}} + \eta_t^2 \sigma^2 \mathbb{E}[\mathbf{H}_{\mathbf{w}}^t]$$

$$\stackrel{(a)}{\preceq} \left( \mathcal{I} - \eta_t \widehat{\mathcal{G}} \right) \circ \mathbf{V}_{\text{diag}}^t + \widetilde{\mathcal{O}} \left( \eta_t^2 \sigma^2 \eta_0 (C+2) \operatorname{tr}(\mathbf{H}) \mathbf{H} + \eta_t^2 \sigma^2 \mathbf{H} \right)$$

$$= \left( \mathcal{I} - \eta_t \widehat{\mathcal{G}} \right) \circ \mathbf{V}_{\text{diag}}^t + \widetilde{\mathcal{O}} \left( \eta_t^2 \sigma^2 \mathbf{H} \right), \tag{41}$$

where (a) is derived from Lemma B.4. Therefore, the recursion for $\mathbf{V}_{\text{diag}}^{T_2}$ can be directly derived by incorporating equation 41 as

$$\mathbf{V}_{\text{diag}}^{T_2} \precsim \sigma^2 \sum_{t=0}^{T_2} \eta_t^2 \prod_{i=t+1}^{T_2} \left( \mathcal{I} - \eta_i \widehat{\mathcal{G}} \right) \circ \mathbf{H} \stackrel{(b)}{\precsim} \sigma^2 \underbrace{\sum_{t=0}^{T_2} \eta_t^2 \prod_{i=t+1}^{T_2} \left( \mathbf{I} - \eta_i \widehat{\mathbf{H}} \right) \mathbf{H}}_{I}, \tag{42}$$

where (b) is based on the inequality $(1 - \eta c_2)^2 c_3 \le (1 - \eta c_2) c_3$, which holds for any $\eta \le c_2^{-1}$ given fixed constants $c_2, c_3 > 0$. According to the update rule for $\eta_t$ defined in Algorithm 1, we obtain

$$I = \eta_0^2 \sum_{i=1}^h \left( \mathbf{I} - \eta_0 \widehat{\mathbf{H}} \right)^{h-i} \prod_{j=1}^L \left( \mathbf{I} - \frac{\eta_0}{2^j} \widehat{\mathbf{H}} \right)^K \mathbf{H}$$

$$+ \sum_{l=1}^L \left( \frac{\eta_0}{2^l} \right)^2 \sum_{i=1}^K \left( \mathbf{I} - \frac{\eta_0}{2^l} \widehat{\mathbf{H}} \right)^{K-i} \prod_{j=l+1}^L \left( \mathbf{I} - \frac{\eta_0}{2^j} \widehat{\mathbf{H}} \right)^K \mathbf{H}$$

$$\preceq 4 \left( \left( \frac{\eta_0}{2} \right)^2 \sum_{i=1}^{h+K} \left( \mathbf{I} - \frac{\eta_0}{2} \widehat{\mathbf{H}} \right)^{h+K-i} \prod_{j=1}^{L-1} \left( \mathbf{I} - \frac{\eta_0}{2^{1+j}} \widehat{\mathbf{H}} \right)^K \mathbf{H} \right.$$

$$\left. + \sum_{l=1}^{L-1} \left( \frac{\eta_0}{2^{1+l}} \right)^2 \sum_{i=1}^K \left( \mathbf{I} - \frac{\eta_0}{2^{1+l}} \widehat{\mathbf{H}} \right)^{K-i} \prod_{j=l+1}^{L-1} \left( \mathbf{I} - \frac{\eta_0}{2^{1+j}} \widehat{\mathbf{H}} \right)^K \mathbf{H} \right)$$

$$\preceq 100 \left( \frac{\eta_0}{2} \left( \mathbf{I} - \left( \mathbf{I} - \frac{\eta_0}{2} \widehat{\mathbf{H}}_{1:D} \right)^{h+K} \right) \prod_{j=1}^{L-1} \left( \mathbf{I} - \frac{\eta_0}{2^{1+j}} \widehat{\mathbf{H}}_{1:D} \right)^{K} \right.$$

$$+ \sum_{l=1}^{L-1} \frac{\eta_0}{2^{1+l}} \left( \mathbf{I} - \left( \mathbf{I} - \frac{\eta_0}{2^{1+l}} \widehat{\mathbf{H}}_{1:D} \right)^{K} \right) \prod_{j=l+1}^{L-1} \left( \mathbf{I} - \frac{\eta_0}{2^{1+j}} \widehat{\mathbf{H}}_{1:D} \right)^{K} \right)$$

$$+ 2\eta_0^2 (h + K) \mathbf{H}_{D+1:M}. \tag{43}$$

Then, we define the following scalar function

$$f(x) := x \left( 1 - (1-x)^{h+K} \right) \prod_{j=1}^{L-1} \left( 1 - \frac{x}{2^j} \right)^{K} + \sum_{l=1}^{L-1} \frac{x}{2^l} \left( 1 - \left( 1 - \frac{x}{2^l} \right)^{K} \right) \prod_{j=l+1}^{L-1} \left( 1 - \frac{x}{2^j} \right)^{K},$$

as similar as that in [Lemma C.2, Wu et al. (2022)]. Moreover, the following inequality can be directly derived

$$f\left( \frac{\eta_0}{2} \widehat{\mathbf{H}}_{1:D} \right) \preceq \frac{8}{K} \mathbf{I}_{1:N_0'} + \eta_0 \widehat{\mathbf{H}}_{N_0'+1:N_0} + \frac{\eta_0^2}{2} (h+K) \widehat{\mathbf{H}}_{N_0+1:D}^2, \tag{44}$$

for arbitrary $D \geq N_0 \geq N_0' \geq 0$ by [Lemma C.3, Wu et al. (2022)]. Applying equation 44 to equation 43 and combining equation 42, we obtain

$$\mathbf{V}_{\mathrm{diag}}^{T_2} \precsim \sigma^2 \left( \frac{1}{K} \widehat{\mathbf{H}}_{1:N_0'}^{-1} + \eta_0 \mathbf{I}_{N_0'+1:N_0} + \eta_0^2 (h+K) \widehat{\mathbf{H}}_{N_0+1:D} + \eta_0^2 (h+K) \mathbf{H}_{D+1:M} \right). \tag{45}$$

Consequently, we have

$$\langle \mathbf{H}, \mathbf{V}^{T_2} \rangle \precsim \sigma^2 \left( \frac{N_0'}{K} + \eta_0 \operatorname{tr} \left( \widehat{\mathbf{H}}_{N_0'+1:N_0} \right) + \eta_0^2 (h+K) \operatorname{tr} \left( \widehat{\mathbf{H}}_{N_0+1:D}^2 \right) \right)$$

$$+ \sigma^2 \eta_0^2 (h+K) \operatorname{tr} \left( \mathbf{H}_{D+1:M}^2 \right)$$

$$\precsim \sigma^2 \left( \frac{N_0'}{K} + \eta_0 \sum_{i=N_0'+1}^{N_0} \lambda_i (\mathbf{v}_i^*)^2 \right) + \sigma^2 \eta_0^2 (h+K) \sum_{i=N_0+1}^{M} \lambda_i^2 (\widehat{\mathbf{b}}_i^*)^4. \tag{46}$$

$$\square$$

**Bound of Bias**: We begin with an analysis of the bias error during a single period of Algorithm 1, where the bias iterations are updated using a constant step size $\eta_t \equiv \eta$ over $\hat{T}$ steps. Based on equation 27, the bias iterations are updated according to the following rule:

$$\mathbf{B}^{t+1} \preceq \mathbb{E} \left[ (\mathcal{I} - \eta \mathcal{G}_{\mathbf{w}}^t) \circ \left( \widehat{\mathbf{w}}_{\mathrm{bias}}^t \otimes \widehat{\mathbf{w}}_{\mathrm{bias}}^t \right) \right], \quad \forall t \in [0 : \hat{T} - 1]. \tag{47}$$

Combining equation 47, we have

$$\mathbf{B}_{\mathrm{diag}}^{t+1} \preceq \left( \mathcal{I} - \eta \widehat{\mathcal{G}} \right) \circ \mathbf{B}_{\mathrm{diag}}^t + \eta^2 \mathbb{E} \left( \left[ \mathcal{H}_{\mathbf{w}}^t \circ \mathbf{B}^t \right] \right)_{\mathrm{diag}}$$

$$\preceq \prod_{i=0}^{t} \left( \mathcal{I} - \eta \widehat{\mathcal{G}} \right) \circ \mathbf{B}_{\mathrm{diag}}^0 + \eta^2 \sum_{i=0}^{t} \prod_{j=i+1}^{t} \left( \mathcal{I} - \eta \widehat{\mathcal{G}} \right) \circ \mathbb{E} \left( \left[ \mathcal{H}_{\mathbf{w}}^t \circ \mathbf{B}^t \right] \right)_{\mathrm{diag}}$$

$$\overset{(a)}{\preceq} \prod_{i=0}^{t} \left( \mathcal{I} - \eta \widehat{\mathcal{G}} \right) \circ \mathbf{B}_{\mathrm{diag}}^0 + (C+2) \eta^2 \sum_{i=0}^{t} \prod_{j=i+1}^{t} \left( \mathcal{I} - \eta \widehat{\mathcal{G}} \right) \circ \mathbf{H} \langle \mathbf{H}, \mathbf{B}^i \rangle. \tag{48}$$

where (a) is derived from Lemma E.6 by selecting $\mathbf{A} = \frac{5}{2} \operatorname{diag}\{\widehat{\mathbf{b}}\}$ and $\mathbf{B} = \mathbf{B}^i$. According to equation 48, we have

$$\mathbf{B}_{\mathrm{diag}}^{t+1} \preceq \left( \mathcal{I} - \eta \widehat{\mathcal{G}} \right)^{t+1} \circ \mathbf{B}_{\mathrm{diag}}^0 + (C+2) \eta^2 \sum_{i=0}^{t} \left( \mathbf{I} - \eta \widehat{\mathbf{H}} \right)^{2(t-i)} \mathbf{H} \langle \mathbf{H}, \mathbf{B}^i \rangle. \tag{49}$$

We utilize the following lemma to estimate $\left\langle \mathbf{H}, \mathbf{B}^{\hat{T}} \right\rangle$ under bias iteration defined in equation 47.

**Lemma B.6.** *Suppose Assumption 3.1 and Assumption 3.3 hold, and $\mathbf{B}^t$ is recursively defined by equation 47. Under the setting of Theorem B.2, letting $1 \leq \hat{T} \leq T$ and $\eta \leq \eta_0$, we have*

$$\left\langle \mathbf{H}, \mathbf{B}^{\hat{T}} \right\rangle \leq \frac{2}{1 - \widetilde{\mathcal{O}}(C+2)\eta \operatorname{tr}(\mathbf{H})} \left\langle \frac{25}{\eta \hat{T}} \mathbf{I}_{1:N_0} + \mathbf{H}_{N_0+1:M}, \mathbf{B}^0 \right\rangle, \tag{50}$$

*where $N_0 \in [0 : D]$ is an arbitrary integer.*

*Proof.* By Lemma E.5, we can derive $\eta(\mathbf{I} - \eta\widehat{\mathbf{H}})^{2t}\mathbf{H} \preceq \frac{25}{t+1}\mathbf{I}$. Applying this to equation 49, we obtain

$$\mathbf{B}_{\mathrm{diag}}^{t+1} \preceq \left(\mathcal{I} - \eta\widehat{\mathcal{G}}\right)^{t+1} \circ \mathbf{B}_{\mathrm{diag}}^0 + 25(C+2)\eta \sum_{i=0}^{t} \frac{\left\langle \mathbf{H}, \mathbf{B}^i \right\rangle}{t+1-i} \cdot \mathbf{I}, \tag{51}$$

for any $t \in [0 : \hat{T} - 1]$. Therefore, based on Lemma E.7, we have

$$\sum_{i=0}^{t} \frac{\left\langle \mathbf{H}, \mathbf{B}^i \right\rangle}{t+1-i} \leq \left\langle \sum_{i=0}^{t} \frac{(\mathbf{I} - \eta\widehat{\mathbf{H}})^{2i}\mathbf{H}}{t+1-i}, \mathbf{B}^0 \right\rangle + \widetilde{\mathcal{O}}(C+2)\eta \operatorname{tr}(\mathbf{H}) \sum_{i=0}^{t} \frac{\left\langle \mathbf{H}, \mathbf{B}^i \right\rangle}{t+1-i}, \tag{52}$$

for any $t \in [\hat{T}]$. equation 52 implies that

$$\sum_{t=0}^{\hat{T}-1} \frac{\left\langle \mathbf{H}, \mathbf{B}^t \right\rangle}{\hat{T}-t} \leq \frac{1}{1 - \widetilde{\mathcal{O}}(C+2)\eta \operatorname{tr}(\mathbf{H})} \left\langle \sum_{t=0}^{\hat{T}-1} \frac{(\mathbf{I} - \eta\widehat{\mathbf{H}})^{2t}\mathbf{H}}{\hat{T}-t}, \mathbf{B}^0 \right\rangle, \tag{53}$$

since $\widetilde{\mathcal{O}}\eta(C+2)\operatorname{tr}(\mathbf{H}) < 1$. Combining equation 51 with equation 53, we obtain

$$
\begin{aligned}
\left\langle \mathbf{H}, \mathbf{B}^{\hat{T}} \right\rangle &\leq \left\langle (\mathbf{I} - \eta\widehat{\mathbf{H}})^{2\hat{T}}\mathbf{H}, \mathbf{B}^0 \right\rangle + \frac{\mathcal{O}(C+2)\eta \operatorname{tr}(\mathbf{H})}{1 - \widetilde{\mathcal{O}}(C+2)\eta \operatorname{tr}(\mathbf{H})} \left\langle \sum_{t=0}^{\hat{T}-1} \frac{(\mathbf{I} - \eta\widehat{\mathbf{H}})^{2t}\mathbf{H}}{\hat{T}-t}, \mathbf{B}^0 \right\rangle \\
&\overset{(a)}{\leq} \left\langle (\mathbf{I} - \eta\widehat{\mathbf{H}})^{2\hat{T}}\mathbf{H}, \mathbf{B}^0 \right\rangle \\
&\quad + \frac{\mathcal{O}(C+2)\eta \operatorname{tr}(\mathbf{H})}{1 - \widetilde{\mathcal{O}}(C+2)\eta \operatorname{tr}(\mathbf{H})} \left\langle \frac{\mathbf{I}_{1:D} - (\mathbf{I}_{1:D} - \eta\widehat{\mathbf{H}}_{1:D})^{\hat{T}}}{\eta\hat{T}} + (\mathbf{I}_{1:D} - \eta\widehat{\mathbf{H}}_{1:D})^{\hat{T}}\widehat{\mathbf{H}}_{1:D}, \mathbf{B}^0 \right\rangle \\
&\quad + \frac{\mathcal{O}(C+2)\eta \operatorname{tr}(\mathbf{H})}{1 - \widetilde{\mathcal{O}}(C+2)\eta \operatorname{tr}(\mathbf{H})} \left\langle \mathbf{H}_{D+1:M}, \mathbf{B}^0 \right\rangle \\
&\overset{(b)}{\leq} \frac{2}{1 - \widetilde{\mathcal{O}}(C+2)\eta \operatorname{tr}(\mathbf{H})} \left\langle \frac{25}{\eta T} \mathbf{I}_{1:N_0} + \mathbf{H}_{N_0+1:M}, \mathbf{B}^0 \right\rangle, 
\end{aligned} \tag{54}
$$

where $N_0 \in [0 : D]$ is an arbitrary integer, (a) follows the technique in [Lemma C.4, Wu et al. (2022)], and (b) is derived from the invariant scaling relationship between $\widehat{\mathbf{H}}_{1:D}$ and $\mathbf{H}_{1:D}$. $\quad\square$

**Lemma B.7.** *Suppose Assumption 3.1 and Assumption 3.3 hold. Under the setting of Theorem B.2, letting $2 \leq \hat{T} \leq T$ and $\eta \leq \eta_0$, we have*

$$\mathbf{B}_{\mathrm{diag}}^{\hat{T}} \preceq \left(\mathbf{I} - \eta\widehat{\mathbf{H}}\right)^{\hat{T}} \mathbf{B}_{\mathrm{diag}}^0 \left(\mathbf{I} - \eta\widehat{\mathbf{H}}\right)^{\hat{T}} + \frac{\widetilde{\mathcal{O}}(C+2)\eta^2\hat{T}}{1 - \widetilde{\mathcal{O}}(C+2)\eta \operatorname{tr}(\mathbf{H})} \left\langle \widetilde{\mathbf{H}}^{\hat{T}}, \mathbf{B}^0 \right\rangle \overline{\mathbf{H}}^{\hat{T}}, \tag{55}$$

*where $\widetilde{\mathbf{H}}^t := \frac{25}{\eta t}\mathbf{I}_{1:N_0} + \mathbf{H}_{N_0+1:M}$, and $\overline{\mathbf{H}}^t := \frac{25}{\eta t}\mathbf{I}_{1:N_0'} + \mathbf{H}_{N_0'+1:M}$ for any $t \geq 1$, and $N_0, N_0' \in [0 : D]$ could be arbitrary integer.*

*Proof.* Applying Lemma B.6 into equation 49, we obtain

$$
\begin{aligned}
\mathbf{B}_{\mathrm{diag}}^{\hat{T}} &\preceq \left(\mathcal{I} - \eta\widehat{\mathcal{G}}\right)^{\hat{T}} \circ \mathbf{B}_{\mathrm{diag}}^0 + (C+2)\eta^2 \left(\mathbf{I} - \eta\widehat{\mathbf{H}}\right)^{2(\hat{T}-1)} \mathbf{H} \left\langle \mathbf{H}, \mathbf{B}^0 \right\rangle \\
&\quad + (C+2)\eta^2 \sum_{t=1}^{\hat{T}-1} \left(\mathbf{I} - \eta\widehat{\mathbf{H}}\right)^{2(\hat{T}-1-t)} \mathbf{H} \left\langle \mathbf{H}, \mathbf{B}^t \right\rangle
\end{aligned}
$$

$$\preceq \left(\mathcal{I} - \eta\widehat{\mathcal{G}}\right)^{\hat{T}} \circ \mathbf{B}_{\text{diag}}^0 + (C+2)\eta^2 \underbrace{\left(\mathbf{I} - \eta\widehat{\mathbf{H}}\right)^{2(\hat{T}-1)} \mathbf{H} \left\langle \mathbf{H}, \mathbf{B}^0\right\rangle}_{\mathcal{I}}$$

$$+ \frac{2(C+2)\eta^2}{1 - 2\widetilde{\mathcal{O}}(C+2)\eta \operatorname{tr}(\mathbf{H})} \underbrace{\sum_{t=1}^{\hat{T}-1} \left(\mathbf{I} - \eta\widehat{\mathbf{H}}\right)^{2(\hat{T}-1-t)} \mathbf{H} \left\langle \widetilde{\mathbf{H}}^t, \mathbf{B}^0\right\rangle}_{\mathcal{II}}, \tag{56}$$

We then provide a bound of term $\mathcal{II}$ as follows:

$$\mathcal{II} = \left(\sum_{t=1}^{\hat{T}-1} \left\langle \widetilde{\mathbf{H}}^t, \mathbf{B}^0\right\rangle\right) \mathbf{H}_{D+1:M} + 25 \sum_{t=1}^{\hat{T}-1} \left(\mathbf{I}_{1:D} - \eta\widehat{\mathbf{H}}_{1:D}\right)^{2(\hat{T}-1-t)} \widehat{\mathbf{H}}_{1:D} \left\langle \widetilde{\mathbf{H}}^t, \mathbf{B}^0\right\rangle$$

$$\preceq \hat{T}\log(\hat{T}) \left\langle \widetilde{\mathbf{H}}^{\hat{T}}, \mathbf{B}^0\right\rangle \mathbf{H}_{D+1:M} + 25 \left(\sum_{t=1}^{\hat{T}/2-1} \left(\mathbf{I}_{1:D} - \eta\widehat{\mathbf{H}}_{1:D}\right)^{\hat{T}} \widehat{\mathbf{H}}_{1:D} \left\langle \widetilde{\mathbf{H}}^t, \mathbf{B}^0\right\rangle\right.$$

$$+ \sum_{t=\hat{T}/2}^{\hat{T}-1} \left(\mathbf{I}_{1:D} - \eta\widehat{\mathbf{H}}_{1:D}\right)^{\hat{T}-1-t} \widehat{\mathbf{H}}_{1:D} \left\langle \widetilde{\mathbf{H}}^{\hat{T}/2}, \mathbf{B}^0\right\rangle\right)$$

$$= \hat{T}\log(\hat{T}) \left\langle \widetilde{\mathbf{H}}^{\hat{T}}, \mathbf{B}^0\right\rangle \mathbf{H}_{D+1:M} + 25 \left(\left(\mathbf{I}_{1:D} - \eta\widehat{\mathbf{H}}_{1:D}\right)^{\hat{T}} \widehat{\mathbf{H}}_{1:D} \left\langle \sum_{t=1}^{\hat{T}/2-1} \widetilde{\mathbf{H}}^t, \mathbf{B}^0\right\rangle\right.$$

$$+ \frac{\mathbf{I}_{1:D} - \left(\mathbf{I}_{1:D} - \eta\widehat{\mathbf{H}}_{1:D}\right)^{\hat{T}/2}}{\eta} \left\langle \widetilde{\mathbf{H}}^{\hat{T}/2}, \mathbf{B}^0\right\rangle\right)$$

$$\preceq \hat{T}\log(\hat{T}) \left\langle \widetilde{\mathbf{H}}^{\hat{T}}, \mathbf{B}^0\right\rangle \mathbf{H}_{D+1:M} + 25 \left(\hat{T}\log(\hat{T}) \left(\mathbf{I}_{1:D} - \eta\widehat{\mathbf{H}}_{1:D}\right)^{\hat{T}} \widehat{\mathbf{H}}_{1:D} \left\langle \widetilde{\mathbf{H}}^{\hat{T}}, \mathbf{B}^0\right\rangle\right.$$

$$+ 2\frac{\mathbf{I}_{1:D} - \left(\mathbf{I}_{1:D} - \eta\widehat{\mathbf{H}}_{1:D}\right)^{\hat{T}/2}}{\eta} \left\langle \widetilde{\mathbf{H}}^{\hat{T}}, \mathbf{B}^0\right\rangle\right)$$

$$\overset{(a)}{\preceq} \hat{T}\log(\hat{T}) \left\langle \widetilde{\mathbf{H}}^{\hat{T}}, \mathbf{B}^0\right\rangle \overline{\mathbf{H}}^{\hat{T}}, \tag{57}$$

where (a) follows the similar technique used in equation 54. We then proceed to establish bounds on $\mathcal{I}$. It's worth to notice that

$$\left(\mathbf{I} - \eta\widehat{\mathbf{H}}\right)^{2(\hat{T}-1)} \mathbf{H} \preceq \frac{25}{2\eta(\hat{T}-1)} \mathbf{I}_{1:N_0'} + 25\widehat{\mathbf{H}}_{N_0'+1:D} + \mathbf{H}_{D+1:M} \preceq \overline{\mathbf{H}}^{\hat{T}}. \tag{58}$$

Applying equation 58 to $\mathcal{I}$, we obtain

$$\mathcal{I} \preceq \overline{\mathbf{H}}^{\hat{T}} \left\langle \mathbf{H}, \mathbf{B}^0\right\rangle \preceq \hat{T}\overline{\mathbf{H}}^{\hat{T}} \left\langle \widetilde{\mathbf{H}}^{\hat{T}}, \mathbf{B}^0\right\rangle, \tag{59}$$

where the last inequality is derived from the condition $\eta < 1/(25\operatorname{tr}(\mathbf{H}))$, which ensures $\lambda_i(\mathbf{H}) < 1/\eta$ holds for all $i \in [N_0]$. Combining the estimation of $\mathcal{I}$ and $\mathcal{II}$ with equation 56, we have

$$\mathbf{B}_{\text{diag}}^{\hat{T}} \preceq \left(\mathcal{I} - \eta\widehat{\mathcal{G}}\right)^{\hat{T}} \circ \mathbf{B}_{\text{diag}}^0 + (C+2)\eta^2\hat{T}\widetilde{\mathbf{H}}^{\hat{T}} \left\langle \widetilde{\mathbf{H}}^{\hat{T}}, \mathbf{B}^0\right\rangle$$

$$+ \frac{2(C+2)\eta^2}{1 - \widetilde{\mathcal{O}}(C+2)\eta \operatorname{tr}(\mathbf{H})} \hat{T}\log(\hat{T}) \left\langle \widetilde{\mathbf{H}}^{\hat{T}}, \mathbf{B}^0\right\rangle \overline{\mathbf{H}}^{\hat{T}}$$

$$\preceq \left(\mathcal{I} - \eta\widehat{\mathcal{G}}\right)^{\hat{T}} \circ \mathbf{B}_{\text{diag}}^0 + \frac{\widetilde{\mathcal{O}}(C+2)\eta^2\hat{T}}{1 - \widetilde{\mathcal{O}}(C+2)\eta \operatorname{tr}(\mathbf{H})} \left\langle \widetilde{\mathbf{H}}^{\hat{T}}, \mathbf{B}^0\right\rangle \overline{\mathbf{H}}^T. \tag{60}$$

By the definition of $\widehat{\mathcal{G}}$, we complete the proof. $\qquad\qquad\square$

Notice that in **Phase II**, the step size $\eta_t$ decays geometrically. Thus, we define the bias iteration at the end of the step-size-decaying phase as:

$$\widetilde{\mathbf{B}}^l := \begin{cases} \mathbf{B}^h, & l = 0, \\ \mathbf{B}^{h+Kl}, & l \in [L]. \end{cases} \tag{61}$$

Based on the step-size iteration in Algorithm 1 and preceding definition, we formalize the iterative process of Algorithm 1 in Phase II as: 1) Phase when $l = 0$: Initialized from $\mathbf{B}^0$, Algorithm 1 runs $h$ iterations with step size $\eta_0$, yielding $\widetilde{\mathbf{B}}^0$; 2) Phase when $l \geq 1$: Initialized from $\widetilde{\mathbf{B}}^{l-1}$, Algorithm 1 runs $K$ iterations with step size $\eta_0/2^l$, yielding $\widetilde{\mathbf{B}}^l$. This multi-phase process terminates at $l = L$, with $\widetilde{\mathbf{B}}^L = \mathbf{B}^{T_2}$ as the final output.

**Lemma B.8.** *Suppose Assumption 3.1 and Assumption 3.3 hold. Under the setting of Theorem B.2, we have*

$$\left\langle \mathbf{H}, \widetilde{\mathbf{B}}^l \right\rangle \leq K_l := \begin{cases} 4 \left\langle \frac{25}{\eta_0 h} \mathbf{I}_{1:N_0} + \mathbf{H}_{N_0+1:M}, \mathbf{B}^0 \right\rangle, & \text{for } l = 0, \\ 4 \left\langle \frac{25 \cdot 2^l}{\eta_0 K} \mathbf{I}_{1:N_0} + \mathbf{H}_{N_0+1:M}, \widetilde{\mathbf{B}}^{l-1} \right\rangle, & \text{for } l \in [L], \end{cases} \tag{62}$$

*for arbitrary $N_0 \in [0 : D]$.*

*Proof.* For $\left\langle \mathbf{H}, \widetilde{\mathbf{B}}^0 \right\rangle$, we apply Lemma B.6 with $\eta = \eta_0$ and $\hat{T} = h$, and use the condition that $\widetilde{\mathcal{O}}(C+2)\eta \operatorname{tr}(\mathbf{H}) \leq 1/4$; For $\left\langle \mathbf{H}, \widetilde{\mathbf{B}}^l \right\rangle$ with $l \geq 2$, we apply Lemma B.6 with $\eta = \eta_0/2^l$, $\hat{T} = K$ and $\mathbf{B}^0 = \widetilde{\mathbf{B}}^{l-1}$, and use the condition that $\widetilde{\mathcal{O}}(C+2)\eta \operatorname{tr}(\mathbf{H}) \leq 1/4$. $\qquad \square$

**Lemma B.9.** *Suppose Assumption 3.1 and Assumption 3.3 hold. Under the setting of Theorem B.2, we have*

$$\widetilde{\mathbf{B}}^l_{\text{diag}} \preceq \mathbf{R}^l := \begin{cases} \left( \mathbf{I} - \eta_0 \widehat{\mathbf{H}} \right)^h \mathbf{B}^0_{\text{diag}} \left( \mathbf{I} - \eta_0 \widehat{\mathbf{H}} \right)^h + P_0 \overline{\mathbf{H}}^h_0, & \text{for } l = 0, \\ \left( \mathbf{I} - \frac{\eta_0}{2^l} \widehat{\mathbf{H}} \right)^h \widetilde{\mathbf{B}}^{l-1}_{\text{diag}} \left( \mathbf{I} - \frac{\eta_0}{2^l} \widehat{\mathbf{H}} \right)^h + P_l \overline{\mathbf{H}}^K_l, & \text{for } l \in [L], \end{cases} \tag{63}$$

*where $\overline{\mathbf{H}}^t_0 := \frac{25}{\eta_0 t} \mathbf{I}_{1:N'_0} + \mathbf{H}_{N'_0+1:M}$ and $\overline{\mathbf{H}}^t_l := \frac{25 \cdot 2^l}{\eta_0 t} \mathbf{I}_{1:N'_0} + \mathbf{H}_{N'_0+1:M}$ for any $t \geq 1$ and arbitrary $N'_0 \in [0 : D]$, and $P_0 := \widetilde{\mathcal{O}}(C+2)\eta_0^2 h \langle \widetilde{\mathbf{H}}^h_0, \mathbf{B}^0 \rangle$ with $\widetilde{\mathbf{H}}^h_0 := \frac{25}{\eta_0 h} \mathbf{I}_{1:N_0} + \mathbf{H}_{N_0+1:M}$ and $P_l := \widetilde{\mathcal{O}}(C+2)(\frac{\eta_0}{2^l})^2 K \langle \widetilde{\mathbf{H}}^K_l, \widetilde{\mathbf{B}}^{l-1} \rangle$ for $l \in [L]$ with $\widetilde{\mathbf{H}}^K_l := \frac{25 \cdot 2^l}{\eta_0 K} \mathbf{I}_{1:N_0} + \mathbf{H}_{N_0+1:M}$ for arbitrary $N_0 \in [0 : D]$.*

*Proof.* For $\widetilde{\mathbf{B}}^0$, we apply Lemma B.7 with $\eta = \eta_0$ and $\hat{T} = h$, and use the condition that $\widetilde{\mathcal{O}}(C+2)\eta \operatorname{tr}(\mathbf{H}) \leq 1/4$. For $\widetilde{\mathbf{B}}^l$ with $l \geq 2$, we apply Lemma B.7 with $\eta = \eta_0/2^l$, $\hat{T} = K$ and $\mathbf{B}^0 = \widetilde{\mathbf{B}}^{l-1}$, and use the condition that $\widetilde{\mathcal{O}}(C+2)\eta \operatorname{tr}(H) \leq 1/8$. $\qquad \square$

**Lemma B.10.** *Suppose Assumption 3.1 and Assumption 3.3 hold. Under the setting of Theorem B.2, we have*

$$\left\langle \mathbf{H}, \mathbf{B}^{T_2} \right\rangle = \left\langle \mathbf{H}, \widetilde{\mathbf{B}}^L \right\rangle \leq e \left\langle \mathbf{H}, \widetilde{\mathbf{B}}^1 \right\rangle \tag{64}$$

*Proof.* Consider $l \geq 1$. According to Lemma B.9, we obtain

$$\begin{aligned} \widetilde{\mathbf{B}}^l_{\text{diag}} &\preceq \left( \mathbf{I} - \frac{\eta_0}{2^l} \widehat{\mathbf{H}} \right)^h \widetilde{\mathbf{B}}^{l-1}_{\text{diag}} \left( \mathbf{I} - \frac{\eta_0}{2^l} \widehat{\mathbf{H}} \right)^h + P_l \widetilde{\mathbf{H}}^K_l \\ &\overset{(a)}{\preceq} \widetilde{\mathbf{B}}^{l-1}_{\text{diag}} + \widetilde{\mathcal{O}}(C+2) \log(K) \cdot \frac{\eta_0}{2^l} \cdot \left\langle \mathbf{H}, \widetilde{\mathbf{B}}^{l-1} \right\rangle \mathbf{I}. \end{aligned} \tag{65}$$

where (a) is derived from choosing $N'_0 = D$ and $N_0 = 0$ in $\overline{\mathbf{H}}^K_l$ and $\widetilde{\mathbf{H}}^K_l$ for any $l \in [L]$, respectively, and $\mathbf{H}_{D+1:M} \preceq \frac{\widetilde{\mathcal{O}}(2^l)}{\eta_0 K} \mathbf{I}_{D+1:M}$. equation 65 implies that

$$\left\langle \mathbf{H}, \widetilde{\mathbf{B}}^l \right\rangle \leq \left( 1 + \widetilde{\mathcal{O}}(C+2) \operatorname{tr}(\mathbf{H}) \log(K) \cdot \frac{\eta_0}{2^l} \right) \left\langle \mathbf{H}, \widetilde{\mathbf{B}}^{l-1} \right\rangle. \tag{66}$$

Therefore, we have following estimation of bias iterations using equation 66:

$$\left\langle \mathbf{H}, \widetilde{\mathbf{B}}^L \right\rangle \leq \prod_{l=1}^{L} \left(1 + \widetilde{\mathcal{O}}(C+2)\operatorname{tr}(\mathbf{H})\log(K) \cdot \frac{\eta_0}{2^l}\right) \left\langle \mathbf{H}, \widetilde{\mathbf{B}}^1 \right\rangle$$

$$\leq \exp\left\{\widetilde{\mathcal{O}}(C+2)\eta_0 \operatorname{tr}(\mathbf{H})\log(K) \sum_{l=1}^{L} 2^{-l}\right\} \left\langle \mathbf{H}, \widetilde{\mathbf{B}}^1 \right\rangle$$

$$\leq e \left\langle \mathbf{H}, \widetilde{\mathbf{B}}^1 \right\rangle. \tag{67}$$

$\square$

**Lemma B.11.** *Suppose Assumption 3.1 and Assumption 3.3 hold. Under the setting of Theorem B.2, we have*

$$\left\langle \mathbf{H}, \widetilde{\mathbf{B}}^1 \right\rangle \leq 8 \left\langle \frac{25}{\eta_0 K}\mathbf{I}_{1:N_0} + \mathbf{H}_{N_0+1:M}, \left(\mathbf{I} - \eta_0\widehat{\mathbf{H}}\right)^{2h}\mathbf{B}^0 \right\rangle$$

$$+ \widetilde{\mathcal{O}}(C+2)\Gamma_K(\mathbf{H}) \left\langle \frac{25}{\eta_0 h}\mathbf{I}_{1:N_0'} + \mathbf{H}_{N_0'+1:M}, \mathbf{B}^0 \right\rangle, \tag{68}$$

*where $\Gamma_K(\mathbf{H}) := \left(\frac{625 N_0'}{K} + \frac{25\eta_0 h}{K}\operatorname{tr}(\mathbf{H}_{N_0'+1:N_0}) + \eta_0^2 h \operatorname{tr}(\mathbf{H}_{N_0+1:M}^2)\right)$ for arbitrary $D \geq N_0 \geq N_0' \geq 0$.*

*Proof.* According to Lemma B.8, we have

$$\left\langle \mathbf{H}, \widetilde{\mathbf{B}}^1 \right\rangle \leq 8 \left\langle \frac{25}{\eta_0 K}\mathbf{I}_{1:N_0} + \mathbf{H}_{N_0+1:M}, \widetilde{\mathbf{B}}^0 \right\rangle,$$

for arbitrary $N_0 \in [0 : D]$. Then, choosing $N_0 = N_0'$ in Lemma B.9, we obtain

$$\widetilde{\mathbf{B}}_{\text{diag}}^0 \preceq \left(\mathbf{I} - \eta_0\widehat{\mathbf{H}}\right)^h \mathbf{B}_{\text{diag}}^0 \left(\mathbf{I} - \eta_0\widehat{\mathbf{H}}\right)^h$$

$$+ \widetilde{\mathcal{O}}(C+2)\eta_0^2 h \left\langle \frac{25}{\eta_0 h}\mathbf{I}_{1:N_0'} + \mathbf{H}_{N_0'+1:M}, \mathbf{B}^0 \right\rangle \left(\frac{25}{\eta_0 h}\mathbf{I}_{1:N_0'} + \mathbf{H}_{N_0'+1:M}\right).$$

Combining above two inequalities, we have

$$\left\langle \mathbf{H}, \widetilde{\mathbf{B}}^1 \right\rangle \leq 8 \left\langle \frac{25}{\eta_0 K}\mathbf{I}_{1:N_0} + \mathbf{H}_{N_0+1:M}, \left(\mathbf{I} - \eta_0\widehat{\mathbf{H}}\right)^{2h}\mathbf{B}^0 \right\rangle$$

$$+ \widetilde{\mathcal{O}}(C+2)\eta_0^2 h \left\langle \frac{25}{\eta_0 h}\mathbf{I}_{1:N_0'} + \mathbf{H}_{N_0'+1:M}, \mathbf{B}^0 \right\rangle$$

$$\times \left\langle \frac{25}{\eta_0 K}\mathbf{I}_{1:N_0} + \mathbf{H}_{N_0+1:M}, \frac{25}{\eta_0 h}\mathbf{I}_{1:N_0'} + \mathbf{H}_{N_0'+1:M} \right\rangle,$$

where

$$\left\langle \frac{25}{\eta_0 K}\mathbf{I}_{1:N_0} + \mathbf{H}_{N_0+1:M}, \frac{25}{\eta_0 h}\mathbf{I}_{1:N_0'} + \mathbf{H}_{N_0'+1:M} \right\rangle$$

$$\leq \frac{625 N_0'}{\eta_0^2 h K} + \frac{25}{\eta_0 K}\operatorname{tr}(\mathbf{H}_{N_0'+1:N_0}) + \operatorname{tr}(\mathbf{H}_{N_0+1:M}^2), \tag{69}$$

when $N_0 > N_0'$. $\square$

**Lemma B.12.** *Suppose Assumptions 3.1 and 3.3 hold. Under the setting of Theorem B.2, we have*

$$\left\langle \mathbf{H}, \mathbf{B}^{T_2} \right\rangle \lesssim \left\langle \frac{1}{\eta_0 K}\mathbf{I}_{1:N_0} + \mathbf{H}_{N_0+1:M}, \left(\mathbf{I} - \eta_0\widehat{\mathbf{H}}\right)^{2h}\mathbf{B}^0 \right\rangle$$

$$+ (C+2)\Gamma_K(\mathbf{H}) \left\langle \frac{1}{\eta_0 h}\mathbf{I}_{1:N_0'} + \mathbf{H}_{N_0'+1:M}, \mathbf{B}^0 \right\rangle, \tag{70}$$

*where $\Gamma_K(\mathbf{H}) := \left(\frac{625 N_0'}{K} + \frac{25\eta_0 h}{K}\operatorname{tr}(\mathbf{H}_{N_0'+1:N_0}) + \eta_0^2 h \operatorname{tr}(\mathbf{H}_{N_0+1:M}^2)\right)$ for arbitrary $D \geq N_0 \geq N_0' \geq 0$.*

*Proof.* Using Lemma B.10 and Lemma B.11 we directly obtain the results. ☐

Finally, we will finish the proof of Theorem B.2.

*Proof of Theorem B.2.* Combining Lemma B.5 with Lemma B.12, we derive equation 28. Based on Theorem B.3, the equality $\mathbf{w}^{T_2} = \mathbf{v}^{T_1+T_2}$ holds with probability at least $1 - \delta$. By setting $N_0' = N_0 = N_1' = N_1 = D$ in equation 28 and applying Markov's inequality, we obtain equation 29. ☐

### B.4 PROOF OF MAIN RESULTS

In this section, we finally complete the proof of main results for the global convergence of Algorithm 1 in Theorem B.4, based on the analysis of **Phase I** and **Phase II**. Before we propose the main Theorem B.4, we set the parameter as follows:

$$L_1 = \widetilde{\mathcal{O}}\left((\sigma^2 + \mathcal{M}^2(\mathbf{b}))^2 + \hat{\sigma}_{\max}(D)\right), \quad L_2 = \widetilde{\mathcal{O}}(\sigma^2 + \mathcal{M}^2(\mathbf{b})), \quad L_3 = 1 + \frac{L_1 \tilde{\sigma}_{\max}(D)\bar{\sigma}_{\min}(D)}{\sigma_{\min}(D)}, \tag{71}$$

**Theorem B.4.** *[Upper Bound in Theorem 4.1] Under Assumption 3.1 and 3.3, we consider a predictor trained by Algorithm 1 with total sample size $T$. Let $h < T$ and $T_1 := \lceil (T-h)/\log(T-h) \rceil$. Suppose there exists $D \le M$ such that $T_1 \in [\frac{L_1 L_3}{\sigma_{\min}(D)\bar{\sigma}_{\min}(D)}, \frac{L_2 L_3^2}{\tilde{\sigma}_{\max}(D)\bar{\sigma}_{\min}^2(D)}]$ with parameter setting equation 71 and let $\eta = \widetilde{\Omega}(\frac{\bar{\sigma}_{\min}(D)}{\sigma^2+\mathcal{M}^2(\mathbf{b})})$. Then we have*

$$\mathcal{R}_M(\mathbf{v}^T) - \mathcal{R}_M(\mathbf{v}^*) \lesssim \frac{\sigma^2 D}{T_1} + \sigma^2 \eta^2 (h + T_1) \sum_{i=D+1}^{M} \lambda_i^2 (\mathbf{v}_i^*)^4$$

$$+ \frac{1}{\eta T_1} \operatorname{tr}\left(\left(\mathbf{I}_{1:D} - \frac{\eta}{4}\mathbf{H}_{1:D}^*\right)^{2h} \operatorname{diag}\left\{(\mathbf{v}_{1:D}^*)^{\odot 2}\right\}\right)$$

$$+ \left\langle \mathbf{H}_{D+1:M}^*, \operatorname{diag}\left\{(\mathbf{v}_{D+1:M}^*)^{\odot 2}\right\}\right\rangle$$

$$+ \left(\frac{D}{T_1} + \eta^2 h \operatorname{tr}\left((\mathbf{H}_{D+1:M}^*)^2\right)\right)\left\langle \frac{1}{\eta h}\mathbf{I}_{1:D} + \mathbf{H}_{D+1:M}^*, \operatorname{diag}\left\{(\mathbf{v}^*)^{\odot 2}\right\}\right\rangle,$$

*with probability at least 0.95. Otherwise, let $T_1 \in [\frac{L_1 L_3}{\sigma_{\min}(M)\bar{\sigma}_{\min}(M)}, +\infty)$ with parameter setting equation 71 and $\eta = \widetilde{\Omega}(\frac{\bar{\sigma}_{\min}(M)}{\sigma^2+\mathcal{M}^2(\mathbf{b})})$. Then we have*

$$\mathcal{R}_M(\mathbf{v}^T) - \mathcal{R}_M(\mathbf{v}^*) \lesssim \frac{\sigma^2 M}{T_1} + \frac{1}{\eta T_1} \operatorname{tr}\left(\left(\mathbf{I} - \frac{\eta}{4}\mathbf{H}^*\right)^{2h} \operatorname{diag}\left\{(\mathbf{v}^*)^{\odot 2}\right\}\right)$$

$$+ \frac{M}{\eta h T_1} \operatorname{tr}\left(\operatorname{diag}\left\{(\mathbf{v}^*)^{\odot 2}\right\}\right),$$

*with probability at least 0.95.*

*Proof.* Combining Theorem B.1 and Theorem B.2, we complete the proof. ☐

## C PROOFS OF LOWER BOUND (THEOREM 4.1)

In this section, we introduce the proof of the lower bound in Theorem C.1. Let $\bar{\sigma}^2 := \mathbb{E}[\xi^2] + \sum_{i=M+1}^{\infty} \lambda_i (\mathbf{v}_i^*)^4$. Recall the analysis in Phase I, $\mathbf{v}^{T_1}$ satisfies the inequality $\overline{\mathbf{b}} \le \mathbf{v}^{T_1} \le \widehat{\mathbf{b}}$ with high probability. Here, $\widehat{\mathbf{b}}$ is defined as $\widehat{\mathbf{b}}^\top = (\frac{3}{2}(\mathbf{v}_{1:D}^*)^\top, 3(\mathbf{v}_{D+1:M}^*)^\top)$, while $\overline{\mathbf{b}}$ is defined as $\overline{\mathbf{b}}^\top = (\frac{1}{2}(\mathbf{v}_{1:D}^*)^\top, \mathbf{0}^\top)$. We begin with the required concepts as below. A Markov chain $\{\breve{\mathbf{v}}^t\}_{t=0}^{T_2}$ is constructed with initialization $\breve{\mathbf{v}}^0$ satisfying $\overline{\mathbf{b}} \le \breve{\mathbf{v}}^0 \le \widehat{\mathbf{b}}$. The update rule is defined by

$$\breve{\mathbf{v}}^{t+1} = \breve{\mathbf{v}}^t - \eta_t \mathbf{H}_{\breve{\mathbf{v}}}^t \left(\breve{\mathbf{v}}^t - \mathbf{v}^*\right) + \eta_t \mathbf{R}_{\breve{\mathbf{v}}}^t \mathbf{x}^t, \quad \forall t \in [0 : T_2 - 1],$$

where $\mathbf{H}_{\breve{\mathbf{v}}}^t$ and $\mathbf{R}_{\breve{\mathbf{v}}}^t$ satisfy:

1. If $\overline{\mathbf{b}} \leq \breve{\mathbf{v}}^t \leq \widehat{\mathbf{b}}$, $\mathbf{H}_{\breve{\mathbf{v}}}^t = (\breve{\mathbf{v}}^t \odot \mathbf{x}^t) \otimes ((\breve{\mathbf{v}}^t + \mathbf{v}^*) \odot \mathbf{x}^t)$ and $\mathbf{R}_{\breve{\mathbf{v}}}^t = (\xi^t + \sum_{i=M+1}^{\infty} \mathbf{x}_i^t (\mathbf{v}_i^*)^2) \operatorname{diag}\{\breve{\mathbf{v}}^t\}$,

2. Otherwise, for any $\tau \in [t : T_2 - 1]$, $\mathbf{H}_{\breve{\mathbf{v}}}^\tau = \frac{25}{4}(\mathbf{v}^* \odot \Pi_M \mathbf{x}^\tau) \otimes (\mathbf{v}^* \odot \Pi_M \mathbf{x}^\tau)$ and $\mathbf{R}_{\breve{\mathbf{v}}}^\tau = (\xi^\tau + \sum_{i=M+1}^{\infty} \mathbf{x}_i^\tau (\mathbf{v}_i^*)^2) \operatorname{diag}\{\overline{\mathbf{b}}\}$.

Let $\breve{\mathbf{w}}^t := \breve{\mathbf{v}}^t - \mathbf{v}^*$ be the error vector, and let $t_s := \inf\{t \mid \breve{\mathbf{v}}^t \not\leq \widehat{\mathbf{b}} \bigvee \breve{\mathbf{v}}^t \not\geq \overline{\mathbf{b}}\}$ be the stopping time. According to equation 72, $\{\breve{\mathbf{w}}^t\}_{t=1}^{T_2}$ is recursively defined by

$$\breve{\mathbf{w}}^{t+1} = \left(\mathbf{I} - \eta_t \mathbf{H}_{\breve{\mathbf{v}}}^t\right)\breve{\mathbf{w}}^t + \eta_t \mathbf{R}_{\breve{\mathbf{v}}}^t \mathbf{x}^t.$$

We define $\breve{\mathbf{V}}^t = \mathbb{E}\left[\breve{\mathbf{w}}^t \otimes \breve{\mathbf{w}}^t\right]$. By the definitions of $\mathcal{H}_\cdot^t$, $\widetilde{\mathcal{H}}_\cdot^t$, $\mathcal{G}_\cdot^t$, and $\widetilde{\mathcal{G}}_\cdot^t$ in Phase II, we derive the iterative relationship governing the sequence $\{\breve{\mathbf{V}}^t\}_{t=0}^{T_2}$:

$$\breve{\mathbf{V}}^{t+1} = \mathbb{E}\left[\left(\mathcal{I} - \eta_t \mathcal{G}_{\breve{\mathbf{v}}}^t\right) \circ \left(\breve{\mathbf{w}}^t \otimes \breve{\mathbf{w}}^t\right)\right] + \eta_t^2 \Sigma_{\breve{\mathbf{v}}}^t, \tag{72}$$

for $t \in [0 : T_2 - 1]$ with $\mathbf{V}^0 = \left(\mathbf{w}^0 - \mathbf{v}^*\right) \otimes \left(\mathbf{w}^0 - \mathbf{v}^*\right)$. If $t < t_s$, $\Sigma_{\breve{\mathbf{v}}}^t = \bar{\sigma}^2 \Lambda \mathbb{E}[\operatorname{diag}\{\breve{\mathbf{v}}^{t \odot 2}\}]$; otherwise, $\Sigma_{\breve{\mathbf{v}}}^\tau = \bar{\sigma}^2 \Lambda \operatorname{diag}\{\overline{\mathbf{b}}^{\odot 2}\}$ for any $\tau \geq t$. According to the definitions above, we obtain following estimation of the last-iteration function value:

$$\mathbb{E}\left[\mathcal{R}_M(\breve{\mathbf{w}}^{T_2}) - \mathcal{R}_M(\mathbf{v}^*)\right] \geq \frac{1}{24}\left\langle\breve{\mathbf{H}}, \mathbb{E}\left[\breve{\mathbf{w}}^{T_2} \otimes \breve{\mathbf{w}}^{T_2}\right]\right\rangle \geq \frac{1}{24}\left\langle\breve{\mathbf{H}}, \breve{\mathbf{V}}^{T_2}\right\rangle, \tag{73}$$

where $\breve{\mathbf{H}} := 12\Lambda \operatorname{diag}\{\mathbf{v}^* \odot \mathbf{v}^*\}$. We define $\breve{\mathcal{G}}^i := \breve{\mathbf{H}} \otimes \mathbf{I} + \mathbf{I} \otimes \breve{\mathbf{H}} - \eta_i \breve{\mathbf{H}} \otimes \breve{\mathbf{H}}$. We formally propose the lower bound of the estimate in Theorem C.1 as below.

**Theorem C.1.** *[Lower Bound in Theorem 4.1] Under Assumption 3.1 and 3.3, we consider a predictor trained by Algorithm 1 with iteration number $T$ and middle phase length $h > \lceil(T-h)/\log(T-h)\rceil$. Let $D \asymp \min\{T^{1/\max\{\beta,(\alpha+\beta)/2\}}, M\}$ and $\eta \asymp D^{\min\{0,(\alpha-\beta)/2\}}$. Then we have*

$$\mathbb{E}\left[\mathcal{R}_M(\mathbf{v}^T)\right] - \mathbb{E}[\xi^2] \gtrsim \frac{1}{M^{\beta-1}} + \frac{\bar{\sigma}^2 D}{T} + \frac{1}{D^{\beta-1}}\mathbb{1}_{M>D}, \tag{74}$$

*where $\bar{\sigma}^2 := \mathbb{E}[\xi^2] + \sum_{i=M+1}^{\infty} \lambda_i(\mathbf{v}_i^*)^4$. Moreover, we can also obtain*

$$\mathcal{R}_M(\mathbf{v}^T) - \mathbb{E}\left[\xi^2\right] \gtrsim \frac{1}{M^{\beta-1}} + \frac{\bar{\sigma}^2 D}{T} + \frac{1}{D^{\beta-1}}\mathbb{1}_{M>D}, \tag{75}$$

*with probability at least 0.95.*

*Proof.* The proof of Theorem C.1 is divided into two steps. **Step I** reveals that for coordinates $j \geq \widetilde{\mathcal{O}}(D)$, the slow ascent rate inherently prevents $\mathbf{v}_j^t$ from attaining close proximity to the optimal solution $\mathbf{v}_j^*$ upon algorithmic termination.

**Step I:** Let $M \gtrsim T^{1/\max\{\beta,(\alpha+\beta)/2\}}$, and define $T_1 := \lceil(T-h)/\log(T-h)\rceil$ and $D^\dagger := \mathcal{O}((\eta T)^{2/(\alpha+\beta)})$. Considering the **b**-capped coupling process $\{\bar{\mathbf{v}}^t\}_{t=0}^T$ mentioned in Phase I, we denote $\hat{\tau}_j$ as the stopping time when $\bar{\mathbf{v}}_j^{\hat{\tau}_j} \geq \frac{1}{4}\mathbf{v}_j^*$ for each coordinate $D^\dagger \leq j \leq M$, i.e.,

$$\hat{\tau}_j = \inf\left\{t : \bar{\mathbf{v}}_j^t \geq \frac{1}{4}\mathbf{v}_j^*\right\}.$$

We aim to estimate the following probability for coordinates $j \in [D^\dagger : M]$ and times $t_1 \in [T_1]$:

$$\mathbb{P}\left(\mathcal{J}^{\hat{\tau}_j=t_1}(j) = \left\{\bar{\mathbf{v}}_j^0 \leq \frac{1}{8}\mathbf{v}_j^* \bigwedge \bar{\mathbf{v}}_j^{0:t_1-1} \in \left[0 : \frac{1}{4}\mathbf{v}_j^*\right] \bigwedge \bar{\mathbf{v}}_j^{t_1} \geq \frac{1}{4}\mathbf{v}_j^*\right\}\right).$$

For fixed $j \in [D^\dagger : M]$ and any $t \in [0 : t_1 - 1]$, we have

$$\begin{aligned}
\mathbb{E}\left[\bar{\mathbf{v}}_j^{t+1} \mid \mathcal{F}^t\right] =& \mathbb{E}_{\mathbf{x}_{1:M}^{t+1}, \xi^{t+1}, \zeta_{M+1:\infty}^{t+1}}\left[\bar{\mathbf{v}}_j^t - \eta\left(\left((\bar{\mathbf{v}}_j^t)^2 - (\mathbf{v}_j^*)^2\right)\hat{\mathbf{x}}_j^{t+1} + \hat{\mathbf{z}}_j^{t+1}(\bar{\mathbf{v}}^t)\right.\right. \\
& \left.\left. - \hat{\zeta}_{M+1:\infty}^{t+1} - \hat{\xi}^{t+1}\right)\hat{\mathbf{x}}_j^{t+1}\bar{\mathbf{v}}_j^{t+1}\right] \tag{76} \\
\leq& \left(1 + \eta\lambda_j(\mathbf{v}_j^*)^2\right)\bar{\mathbf{v}}_j^t.
\end{aligned}$$

Similarly, based on Lemma E.1, we have

$$
\mathbb{E}\left[\exp\left\{\lambda(\bar{\mathbf{v}}_j^{t+1} - \mathbb{E}[\bar{\mathbf{v}}_j^{t+1} \mid \mathcal{F}^t])\right\} \mid \mathcal{F}^t\right] \leq \exp\left\{\frac{\lambda^2 \eta^2 \lambda_j (\mathbf{v}_j^*)^2 \mathcal{O}\left(\left[\bar{\sigma}^2 + \mathcal{M}^2(\mathbf{v}^*)\right]\log^4(MT_1/\delta)\right)}{2}\right\},
$$

for any $\lambda \in \mathbb{R}$. According to the setting of stepsize $\eta$, we have $(1 + \eta\lambda_i(\mathbf{v}_i^*)^2)^{T_1} \leq 2$ for any $i \in [D^\dagger : M]$. Utilizing Corollary E.1 and equation 76, we can establish the probability bound for event $\mathcal{J}^{\hat{\tau}_j = t_1}(j)$ for any time $t_1 \in [T_1]$ as

$$
\mathbb{P}\left(\mathcal{J}^{\hat{\tau}_j = t_1}(j)\right) \leq \exp\left\{-\frac{1}{T\eta^2\lambda_j \mathcal{O}\left(\left[\bar{\sigma}^2 + \mathcal{M}^2(\mathbf{v}^*)\right]\log^2(MT_1/\delta)\right)}\right\}. \tag{77}
$$

Finally, combining the probability bounds equation 77 with the setting of $\eta$, we obtain the following probability bound for complement event $\bigcup_{j=D^\dagger}^{M}\{\max_{t\in[T_1]}\bar{\mathbf{v}}_j^t \geq \frac{1}{4}\mathbf{v}_j^*\}$:

$$
\begin{aligned}
\mathbb{P}\left(\bigcup_{j=D^\dagger}^{M}\left\{\max_{t\in[T_1]}\bar{\mathbf{v}}_j^t \geq \frac{1}{4}\mathbf{v}_j^*\right\}\right) &\leq \sum_{j=D^\dagger}^{M}\sum_{t_1=1}^{T_1}\mathbb{P}\left(\mathcal{J}^{\hat{\tau}_j = t_1}(j)\right) \\
&\leq MT_1 \exp\left\{-\frac{\min_{D^\dagger \leq j \leq M}\lambda_j^{-1}}{T_1\eta^2\mathcal{O}\left(\left[\bar{\sigma}^2 + \mathcal{M}^2(\mathbf{v}^*)\right]\log^4(MT_1/\delta)\right)}\right\} \\
&\leq \frac{\delta}{2}. \tag{78}
\end{aligned}
$$

Therefore, we have $\bigcap_{j=D^\dagger}^{M}\{\max_{t\in[T_1]}\mathbf{v}_j^t < \frac{1}{4}\mathbf{v}_j^*\}$ with high probability.

Similar to **Phase II**'s analysis, **Step II** derives the lower bound estimate of the risk by constructing a recursive expression for $\{\breve{\mathbf{V}}_{\text{diag}}^t\}_{t=0}^{T_2}$ where $T_2 = T - T_1$. We continue to use $\mathbf{v}^{T_1}$, which satisfies equation 11, as the initial point for the SGD iterations in **Step II**. If $M \gtrsim T^{1/\max\{\beta,(\alpha+\beta)/2\}}$, we further require that $\mathbf{v}^{T_1}$ satisfies

$$
\mathbf{v}_j^{T_1} < \frac{1}{4}\mathbf{v}_j^*, \quad \forall j \in \left[\widetilde{\mathcal{O}}(T^{1/\max\{\beta,(\alpha+\beta)/2\}}), M\right].
$$

According to Theorem B.1 and the result of **Step I**, the assumption on $\mathbf{v}^{T_1}$ can be satisfied with high probability.

**Step II:** If $M \gtrsim T^{1/\max\{\beta,(\alpha+\beta)/2\}}$, assume that $\breve{\mathbf{v}}^0$ further satisfies $\breve{\mathbf{v}}_{D^\dagger:M}^0 \leq \frac{1}{4}\mathbf{v}_{D^\dagger:M}^*$. Setting $\eta_0 = \eta$ and $K = T_1$, we have

$$
\begin{aligned}
\breve{\mathbf{V}}_{\text{diag}}^{t+1} &= \left(\mathbb{E}\left[\left(\mathcal{I} - \eta_t\widetilde{\mathcal{G}}_{\breve{\mathbf{v}}}^t\right)\circ\left(\breve{\mathbf{w}}^t \otimes \breve{\mathbf{w}}^t\right)\right]\right)_{\text{diag}} + \eta_t^2\left(\mathbb{E}\left[\left(\mathcal{H}_{\breve{\mathbf{v}}}^t - \widetilde{\mathcal{H}}_{\breve{\mathbf{v}}}^t\right)\circ\left(\breve{\mathbf{w}}^t \otimes \breve{\mathbf{w}}^t\right)\right]\right)_{\text{diag}} + \eta_t^2\Sigma_{\breve{\mathbf{v}}}^t \\
&\succeq \left(\mathcal{I} - \eta_t\breve{\mathcal{G}}^t\right)\circ\breve{\mathbf{V}}_{\text{diag}}^t + \eta_t^2\bar{\sigma}^2\boldsymbol{\Lambda}\operatorname{diag}\left\{\overline{\mathbf{b}}^{\odot 2}\right\},
\end{aligned}
$$

for any $t \in [0 : T_2 - 1]$. According to the recursive step above, we obtain

$$
\begin{aligned}
\breve{\mathbf{V}}^{T_2} &\succeq \bar{\sigma}^2\sum_{t=1}^{T_2}\eta_t^2\prod_{i=t+1}^{T_2}\left(\mathbf{I} - \eta_i\breve{\mathbf{H}}\right)^2\boldsymbol{\Lambda}\operatorname{diag}\left\{\overline{\mathbf{b}}^{\odot 2}\right\} + \underbrace{\left(\mathbf{I} - \eta_0\breve{\mathbf{H}}\right)^{2T_2}\left(\breve{\mathbf{w}}^0 \otimes \breve{\mathbf{w}}^0\right)}_{\mathcal{II}} \\
&\succeq \underbrace{\bar{\sigma}^2\sum_{t=1}^{T_2}\eta_t^2\prod_{i=t+1}^{T_2}\left(\mathbf{I} - 2\eta_i\breve{\mathbf{H}}\right)\boldsymbol{\Lambda}\operatorname{diag}\left\{\overline{\mathbf{b}}^{\odot 2}\right\}}_{\mathcal{I}} + \mathcal{II}. \tag{79}
\end{aligned}
$$

Recalling the step size decay rule in Algorithm 1, we have

$$
\mathcal{I} = \eta_0^2\sum_{i=1}^{h}\left(\mathbf{I} - 2\eta_0\breve{\mathbf{H}}\right)^{h-i}\prod_{j=1}^{L-1}\left(\mathbf{I} - \frac{\eta_0}{2^{j-1}}\breve{\mathbf{H}}\right)^K\boldsymbol{\Lambda}\operatorname{diag}\left\{\overline{\mathbf{b}}^{\odot 2}\right\}
$$

$$+ \sum_{l=1}^{L-1} \left(\frac{\eta_0}{2^l}\right)^2 \sum_{i=1}^{K} \left(\mathbf{I} - \frac{\eta_0}{2^{l-1}}\breve{\mathbf{H}}\right)^{K-i} \prod_{j=l+1}^{L-1} \left(\mathbf{I} - \frac{\eta_0}{2^{j-1}}\breve{\mathbf{H}}\right)^{K} \mathbf{\Lambda} \operatorname{diag}\left\{\overline{\mathbf{b}}^{\odot 2}\right\}$$

$$= \frac{\eta_0^2}{12} \sum_{i=1}^{h} \left(\mathbf{I}_{1:D} - 2\eta_0\breve{\mathbf{H}}_{1:D}\right)^{h-i} \prod_{j=1}^{L-1} \left(\mathbf{I}_{1:D} - \frac{\eta_0}{2^{j-1}}\breve{\mathbf{H}}_{1:D}\right)^{K} \breve{\mathbf{H}}_{1:D}$$

$$+ \frac{1}{12} \sum_{l=1}^{L-1} \left(\frac{\eta_0}{2^l}\right)^2 \sum_{i=1}^{K} \left(\mathbf{I}_{1:D} - \frac{\eta_0}{2^{l-1}}\breve{\mathbf{H}}_{1:D}\right)^{K-i} \prod_{j=l+1}^{L-1} \left(\mathbf{I}_{1:D} - \frac{\eta_0}{2^{j-1}}\breve{\mathbf{H}}_{1:D}\right)^{K} \breve{\mathbf{H}}_{1:D}$$

$$= \frac{\eta_0}{24} \left(\mathbf{I}_{1:D} - \left(\mathbf{I}_{1:D} - 2\eta_0\breve{\mathbf{H}}_{1:D}\right)^{h}\right) \left(\prod_{j=1}^{L-1} \left(\mathbf{I}_{1:D} - \frac{\eta_0}{2^{j-1}}\breve{\mathbf{H}}_{1:D}\right)\right)^{K}$$

$$+ \sum_{l=1}^{L-1} \frac{\eta_0}{12 \cdot 2^{l+1}} \left(\mathbf{I}_{1:D} - \left(\mathbf{I}_{1:D} - \frac{\eta_0}{2^{l-1}}\breve{\mathbf{H}}_{1:D}\right)^{K}\right) \left(\prod_{j=l+1}^{L-1} \left(\mathbf{I}_{1:D} - \frac{\eta_0}{2^{j-1}}\breve{\mathbf{H}}_{1:D}\right)\right)^{K}$$

$$\overset{\text{(a)}}{\geq} \frac{\eta_0}{24} \left(\mathbf{I}_{1:D} - \left(\mathbf{I}_{1:D} - 2\eta_0\breve{\mathbf{H}}_{1:D}\right)^{h}\right) \left(\mathbf{I}_{1:D} - 2\eta_0\breve{\mathbf{H}}_{1:D}\right)^{K}$$

$$+ \sum_{l=1}^{L-1} \frac{\eta_0}{12 \cdot 2^{l+1}} \left(\mathbf{I}_{1:D} - \left(\mathbf{I}_{1:D} - \frac{\eta_0}{2^{l-1}}\breve{\mathbf{H}}_{1:D}\right)^{K}\right) \left(\mathbf{I}_{1:D} - \frac{\eta_0}{2^{l-1}}\breve{\mathbf{H}}_{1:D}\right)^{K}, \tag{80}$$

where (a) is derived from following inequality

$$\prod_{i=l+1}^{L-1} \left(\mathbf{I}_{1:D} - \frac{\eta_0}{2^{i-1}}\breve{\mathbf{H}}_{1:D}\right) \geq \mathbf{I}_{1:D} - \sum_{i=l+1}^{L-1} \frac{\eta_0}{2^{i-1}}\breve{\mathbf{H}} \geq \mathbf{I}_{1:D} - \frac{\eta_0}{2^{l-1}}\breve{\mathbf{H}}_{1:D}.$$

When $h > K$, we apply an auxiliary function analogous to [Lemma D.1, Wu et al. (2022)]'s:

$$f(x) := \frac{x}{2} \left(1 - (1 - 2x)^h\right)(1 - 2x)^K + \sum_{l=1}^{L-1} \frac{x}{2^{l+1}} \left(1 - \left(1 - \frac{x}{2^{l-1}}\right)^K\right)\left(1 - \frac{x}{2^{l-1}}\right)^K.$$

Then, we obtain

$$f(\eta_0\breve{\mathbf{H}}) \succeq \frac{1}{4800K}\mathbf{I}_{1:H_1} + \frac{\eta_0}{480}\breve{\mathbf{H}}_{H_1+1:H_2} + \frac{\eta_0^2 h}{480}\breve{\mathbf{H}}_{H_2+1:D}^2, \tag{81}$$

where $H_1 := \min\{D, \max\{i \mid \lambda_i(\mathbf{v}_i^*)^2 \geq \frac{1}{12\eta_0 K}\}\}$ and $H_2 := \min\{D, \max\{i \mid \lambda_i(\mathbf{v}_i^*)^2 \geq \frac{1}{12\eta_0 h}\}\}$.

For term $\mathcal{II}$, we have

$$\left\langle \breve{\mathbf{H}}, \mathcal{II} \right\rangle \gtrsim \begin{cases} \sum_{i=D^\dagger}^{M} \lambda_i(\mathbf{v}_i^*)^4, & \text{if } M \gtrsim T^{1/\max\{\beta,(\alpha+\beta)/2\}}, \\ 0, & \text{otherwise,} \end{cases} \tag{82}$$

where the estimation for $\left\langle \breve{\mathbf{H}}, \mathcal{II} \right\rangle$ under case $M \gtrsim T^{1/\max\{\beta,(\alpha+\beta)/2\}}$ is derived from the initialization $\breve{\mathbf{v}}_{D^\dagger:M}^0 < \frac{1}{4}\mathbf{v}_{D^\dagger:M}^*$ and $(1 + \eta\lambda_i(\mathbf{v}_i^*)^2)^{2T_2} \leq 2$ for any $i \in [D^\dagger : M]$.

Therefore, using equation 73-equation 82, we derive

$$\mathbb{E}\left[\mathcal{R}_M(\breve{\mathbf{v}}^{T_2}) - \mathcal{R}_M(\mathbf{v}^*)\right] \gtrsim \bar{\sigma}^2 \left\langle \breve{\mathbf{H}}, \frac{1}{K}\breve{\mathbf{H}}_{1:H_1}^{-1} + \eta_0\mathbf{I}_{H_1+1:H_2} + \eta_0^2 h\breve{\mathbf{H}}_{H_2+1:D} \right\rangle + \left\langle \breve{\mathbf{H}}, \mathcal{II} \right\rangle$$

$$= \bar{\sigma}^2 \left(\frac{H_1}{K} + \eta_0 \sum_{i=H_1+1}^{H_2} \lambda_i(\mathbf{v}_i^*)^2 + \eta_0^2 h \sum_{i=H_2+1}^{D} \lambda_i^2(\mathbf{v}_i^*)^4\right) + \left\langle \breve{\mathbf{H}}, \mathcal{II} \right\rangle.$$

According to Lemma B.3, we have $\mathbb{P}(t_s \leq T_2) \leq \delta$, which implies that

$$\mathbb{E}\left[\mathcal{R}_M(\breve{\mathbf{v}}^{T_2}) - \mathcal{R}_M(\mathbf{v}^*) \mid t_s > T_2\right]$$

$$\geq \mathbb{E}\left[\mathcal{R}_M(\breve{\mathbf{v}}^{T_2}) - \mathcal{R}_M(\mathbf{v}^*)\right] - \sum_{i=1}^{T_2} \mathbb{P}(t_s = i)\mathbb{E}\left[\mathcal{R}_M(\breve{\mathbf{v}}^{T_2}) - \mathcal{R}_M(\mathbf{v}^*) \mid t_s = i\right]$$

$$\overset{(b)}{\gtrsim} \bar{\sigma}^2 \left(\frac{H_1}{K} + \eta_0 \sum_{i=H_1+1}^{H_2} \lambda_i(\mathbf{v}_i^*)^2 + \eta_0^2 h \sum_{i=H_2+1}^{D} \lambda_i^2(\mathbf{v}_i^*)^4\right) + \left\langle \breve{\mathbf{H}}, \mathcal{II} \right\rangle. \tag{83}$$

Since $\delta$ is sufficiently small, (b) is drawn from two facts: 1) $\breve{\mathbf{v}}^{t_s}$ resides in a bounded neighborhood of $\widehat{\mathbf{b}}$ or $\overline{\mathbf{b}}$; 2) the risk upper bound for SGD established in [Theorem 4.1, Wu et al. (2022)]. The lower bound established in equation 83 is uniformly valid for all $\breve{\mathbf{v}}^0 \in [\overline{\mathbf{b}}, \widehat{\mathbf{b}}]$. Denote event

$$\mathcal{K}\left(\mathbf{v}^{T_1}\right) := \left\{\overline{\mathbf{b}} \leq \mathbf{v}^{T_1} \leq \widehat{\mathbf{b}} \bigwedge \left\{\mathbf{v}_{D^\dagger:M}^{T_1} \leq \frac{1}{4}\mathbf{v}_{D^\dagger:M}^*, \text{ if } M \gtrsim T^{1/\max\{\beta,(\alpha+\beta)/2\}}\right\}\right\}.$$

For $t_s > T_2$, the trajectory $\{\breve{\mathbf{v}}^t\}_{t=0}^{T_2}$ aligns with Algorithm 1's iterations over $[T_1 : T]$, given the initialization $\breve{\mathbf{v}}^0 = \mathbf{v}^{T_1}$ with $\mathcal{K}(\mathbf{v}^{T_1})$ occurs. Then, we have

$$\min_{\mathbf{v}^{T_1}} \mathbb{E}\left[\mathcal{R}_M(\mathbf{v}^T) - \mathcal{R}_M(\mathbf{v}^*) \mid \mathcal{K}(\mathbf{v}^{T_1})\right]$$

$$\geq (1-\delta) \min_{\mathbf{v}^{T_1}} \mathbb{E}\left[\mathcal{R}_M(\breve{\mathbf{v}}^{T_2}) - \mathcal{R}_M(\mathbf{v}^*) \mid t_s > T_2 \bigwedge \breve{\mathbf{v}}^0 = \mathbf{v}^{T_1} \bigwedge \mathcal{K}(\mathbf{v}^{T_1})\right]$$

$$\gtrsim \bar{\sigma}^2 \left(\frac{H_1}{K} + \eta_0 \sum_{i=H_1+1}^{H_2} \lambda_i(\mathbf{v}_i^*)^2 + \eta_0^2 h \sum_{i=H_2+1}^{D} \lambda_i^2(\mathbf{v}_i^*)^4\right) + \mathbb{1}_{M \gtrsim T^{1/\max\{\beta,(\alpha+\beta)/2\}}} \sum_{i=D^\dagger}^{M} \lambda_i(\mathbf{v}_i^*)^4. \tag{84}$$

Noticing that $\mathcal{K}(\mathbf{v}^{T_1})$ occurs with probability at least $1 - \delta$, and combining equation 83 with equation 84, we obtain

$$\mathbb{E}\left[\mathcal{R}_M(\mathbf{v}^T) - \mathcal{R}_M(\mathbf{v}^*)\right] \geq (1-\delta) \min_{\mathbf{v}^{T_1}} \mathbb{E}\left[\mathcal{R}_M(\mathbf{v}^T) - \mathcal{R}_M(\mathbf{v}^*) \mid \mathcal{K}(\mathbf{v}^{T_1})\right]$$

$$\gtrsim \bar{\sigma}^2 \left(\frac{H_1}{K} + \eta_0 \sum_{i=H_1+1}^{H_2} \lambda_i(\mathbf{v}_i^*)^2 + \eta_0^2 h \sum_{i=H_2+1}^{D} \lambda_i^2(\mathbf{v}_i^*)^4\right)$$

$$+ \mathbb{1}_{M \gtrsim T^{1/\max\{\beta,(\alpha+\beta)/2\}}} \sum_{i=D^\dagger}^{M} \lambda_i(\mathbf{v}_i^*)^4,$$

where $H_1 := \min\{D, \max\{i \mid \lambda_i(\mathbf{v}_i^*)^2 \geq \frac{1}{12\eta_0 K}\}\}$ and $H_2 := \min\{D, \max\{i \mid \lambda_i(\mathbf{v}_i^*)^2 \geq \frac{1}{12\eta_0 h}\}\}$. Furthermore, as analyzed in **Step I**, when $M \geq \widetilde{\mathcal{O}}(T^{1/\max\{\beta,(\alpha+\beta)/2\}})$, the last iterate risk can be bounded below by $D^{1-\beta}$ with high probability; whereas when $M \leq \widetilde{\mathcal{O}}(T^{1/\max\{\beta,(\alpha+\beta)/2\}})$, such a lower bound is governed by $M^{1-\beta}$ with high probability. Therefore, we complete the proof of the lower bound. $\qquad\square$

## D   PROOFS OF THEOREM 4.2

*Proof.* Without loss of generality, we suppose that the orthogonal matrix

$$\operatorname*{arg\,min}_{\mathbf{R} \in \mathbb{R}^{M \times M}, \mathbf{R}\mathbf{R}^\top = \mathbf{I}_M} \left\|\widetilde{\mathbf{U}}\mathbf{R} - \mathbf{U}_{\mathbf{A}}\right\|^2,$$

is $\mathbf{I}_M$. Moreover, let the SVD of $\widetilde{\mathbf{U}}\widetilde{\mathbf{U}}^\top$ be given by

$$\widetilde{\mathbf{U}}\widetilde{\mathbf{U}}^\top = \mathbf{Q}_{\widetilde{\mathbf{U}}}\widetilde{\boldsymbol{\Sigma}}\mathbf{Q}_{\widetilde{\mathbf{U}}}^\top.$$

The proof of Theorem 4.2 mirrors that of the upper bound established in Theorem 4.1. It is similarly divided into two parts: **Phase I** and **Phase II**. For simplicity, we denote $y - \langle \mathbf{Q}_{\mathbf{A}}^\top \mathbf{S}\mathbf{x}, \mathbf{v}^{*\odot 2}\rangle - \xi$ as $\widetilde{\xi}$,

**Phase I:** According to the update rule of $\mathbf{v}^t$ at $t+1$-th step, we have

$$
\begin{aligned}
\mathbf{v}_j^{t+1} =& \mathbf{v}_j^t - \eta_t \left( \left\langle \mathbf{Q}_{\widetilde{\mathbf{U}}}^\top \mathbf{S}\mathbf{x}^{t+1}, (\mathbf{v}^t)^{\odot 2} \right\rangle - \left\langle \mathbf{Q}_{\mathbf{A}}^\top \mathbf{S}\mathbf{x}^{t+1}, \mathbf{v}^{*\odot 2} \right\rangle - \xi^{t+1} - \widetilde{\xi}^{t+1} \right) \cdot \left( \mathbf{Q}_{\widetilde{\mathbf{U}}}^\top \mathbf{S}\mathbf{x}^{t+1} \right)_j \cdot \mathbf{v}_j^t \\
=& \mathbf{v}_j^t - \eta_t \left( \left\langle \widetilde{\boldsymbol{\Sigma}}^{1/2} \mathbf{z}^{t+1}, (\mathbf{v}^t)^{\odot 2} \right\rangle - \left\langle \widetilde{\boldsymbol{\Sigma}}^{1/2} \mathbf{z}^{t+1}, \mathbf{v}^{*\odot 2} \right\rangle - \xi^{t+1} - \widetilde{\xi}^{t+1} \right. \\
& \left. + \left\langle \mathbf{Q}_{\widetilde{\mathbf{U}}}^\top \left( \mathbf{U}_{\mathbf{A}} - \widetilde{\mathbf{U}} \right) \mathbf{z}^{t+1}, (\mathbf{v}^t)^{\odot 2} \right\rangle + \left\langle \left( \widetilde{\boldsymbol{\Sigma}}^{1/2} - \boldsymbol{\Sigma}^{1/2} \right) \mathbf{z}^{t+1}, \mathbf{v}^{*\odot 2} \right\rangle \right) \\
& \cdot \left[ \left( \widetilde{\boldsymbol{\Sigma}}^{1/2} \mathbf{z}^{t+1} \right)_j + \left( \mathbf{Q}_{\widetilde{\mathbf{U}}} \left( \mathbf{U}_{\mathbf{A}} - \widetilde{\mathbf{U}} \right) \mathbf{z}^{t+1} \right)_j \right] \cdot \mathbf{v}_j^t \\
=& \underbrace{\mathbf{v}_j^t - \eta_t \left( \left\langle \widetilde{\boldsymbol{\Sigma}}^{1/2} \mathbf{z}^{t+1}, (\mathbf{v}^t)^{\odot 2} \right\rangle - \left\langle \widetilde{\boldsymbol{\Sigma}}^{1/2} \mathbf{z}^{t+1}, \mathbf{v}^{*\odot 2} \right\rangle - \xi^{t+1} - \widetilde{\xi}^{t+1} \right) \cdot \left( \widetilde{\boldsymbol{\Sigma}}^{1/2} \mathbf{z}^{t+1} \right)_j \cdot \mathbf{v}_j^t}_{\mathcal{I}} \\
& - \eta_t \left( \left\langle \mathbf{Q}_{\widetilde{\mathbf{U}}}^\top \left( \mathbf{U}_{\mathbf{A}} - \widetilde{\mathbf{U}} \right) \mathbf{z}^{t+1}, (\mathbf{v}^t)^{\odot 2} \right\rangle + \left\langle \left( \widetilde{\boldsymbol{\Sigma}}^{1/2} - \boldsymbol{\Sigma}^{1/2} \right) \mathbf{z}^{t+1}, \mathbf{v}^{*\odot 2} \right\rangle \right) \\
& \cdot \left[ \left( \widetilde{\boldsymbol{\Sigma}}^{1/2} \mathbf{z}^{t+1} \right)_j + \left( \mathbf{Q}_{\widetilde{\mathbf{U}}} \left( \mathbf{U}_{\mathbf{A}} - \widetilde{\mathbf{U}} \right) \mathbf{z}^{t+1} \right)_j \right] \cdot \mathbf{v}_j^t \\
& - \eta_t \left( \left\langle \widetilde{\boldsymbol{\Sigma}}^{1/2} \mathbf{z}^{t+1}, (\mathbf{v}^t)^{\odot 2} \right\rangle - \left\langle \widetilde{\boldsymbol{\Sigma}}^{1/2} \mathbf{z}^{t+1}, \mathbf{v}^{*\odot 2} \right\rangle - \xi^{t+1} - \widetilde{\xi}^{t+1} \right) \\
& \cdot \left( \mathbf{Q}_{\widetilde{\mathbf{U}}} \left( \mathbf{U}_{\mathbf{A}} - \widetilde{\mathbf{U}} \right) \mathbf{z}^{t+1} \right)_j \cdot \mathbf{v}_j^t,
\end{aligned}
$$

for any $j \in [M]$, where $\mathbf{z} \sim \mathcal{N}(\mathbf{0}, \mathbf{I_M})$ is a standard $M$-dimensional Gaussian vector. Note that term $\mathcal{I}$ in the above expression is identical to the right-hand side of equation 12. Moreover, under Assumption 4.1, the influence of the remaining terms on the update of $\mathbf{v}_j^t$ at step $t+1$ is dominated by term $\mathcal{I}$. Therefore, using techniques similar to those employed in section B.2, we can derive a result analogous to Theorem B.1.

**Phase II:** Following the technique in section B.3, we can construct a truncated coupling $\{\widehat{\mathbf{v}}^t\}_{t=0}^{T_2}$ and a truncated sequence $\{\mathbf{w}^t\}_{t=0}^{T_2}$. Similarly, we can derive a result analogous to Theorem B.3, which shows that with high probability, the trajectory of $\mathbf{v}^t$ during **Phase II** ($t \in [T_1 : T]$) will remain within a neighborhood of $\mathbf{v}^*$. Then we estimate the risk between the last-step function value and the ground truth as:

$$
\begin{aligned}
& \mathbb{E}\left[ \mathcal{R}_M(\mathbf{w}^{T_2}) - \mathbb{E}_{(\mathbf{x},y)\sim\mathcal{D}} \left( \langle \mathbf{Q}_{\mathbf{A}}^\top \mathbf{S}\mathbf{x}, \mathbf{v}^{*\odot 2} \rangle - y \right)^2 \right] \\
& \overset{(a)}{=} \mathbb{E}\left[ \left| \left\langle \mathbf{Q}_{\widetilde{\mathbf{U}}}^\top \mathbf{S}\mathbf{x}, (\mathbf{w}^{T_2})^{\odot 2} \right\rangle - \left\langle \mathbf{Q}_{\mathbf{A}}^\top \mathbf{S}\mathbf{x}, \mathbf{v}^{*\odot 2} \right\rangle \right|^2 \right] \text{'} \\
& \overset{(b)}{\lesssim} \mathbb{E}\left[ \left\langle \widetilde{\boldsymbol{\Sigma}}^{1/2} \mathbf{z}, (\mathbf{w}^{T_2})^{\odot 2} - \mathbf{v}^{*\odot 2} \right\rangle^2 \right] + \mathbb{E}\left[ \left\langle \left( \boldsymbol{\Sigma}^{1/2} - \widetilde{\boldsymbol{\Sigma}}^{1/2} \right) \mathbf{z}, \mathbf{v}^{*\odot 2} \right\rangle^2 \right] \\
& \quad + \mathbb{E}\left[ \left\langle \mathbf{Q}_{\widetilde{\mathbf{U}}}^\top \left( \mathbf{U}_{\mathbf{A}} - \widetilde{\mathbf{U}} \right) \mathbf{z}, (\mathbf{w}^{T_2})^{\odot 2} \right\rangle \right] \\
& \overset{(c)}{\lesssim} \mathbb{E}\left[ \left\langle \widetilde{\boldsymbol{\Sigma}}^{1/2} \mathbf{z}, (\mathbf{w}^{T_2})^{\odot 2} - \mathbf{v}^{*\odot 2} \right\rangle^2 \right] + \left\| \widetilde{\mathbf{U}}\widetilde{\mathbf{U}}^\top - \mathbf{A} \right\| + \left\| \widetilde{\mathbf{U}} - \mathbf{U}_{\mathbf{A}} \right\|, & (85)
\end{aligned}
$$

where $\mathbf{z} \sim \mathcal{N}(\mathbf{0}, \mathbf{I_M})$ is a standard $M$-dimensional Gaussian vector. Here, (a) follows from condition **[A$_4$]** in Assumption 3.2, (b) is derived from the Cauchy–Schwarz inequality, and (c) relies on Assumption 4.1.

Therefore, according to Eq. equation 36 and the analysis in Part II (B.3.2) of section B.3, it suffices to use the update rule of $\mathbf{w}^t$ to determine the quantities of both the variance $\mathbf{V}^{T_2}$ and bias terms $\mathbf{B}^{T_2}$. We rewrite the update rule of $\mathbf{w}^t$ as follows:

$$
\begin{aligned}
\mathbf{w}^{t+1} =& \mathbf{w}^t - \eta_t \left( \mathbf{Q}_{\widetilde{\mathbf{U}}}^\top \mathbf{U}_{\mathbf{A}} \mathbf{z}^{t+1}, \left\langle (\mathbf{w}^t)^{\odot 2} \right\rangle - y^{t+1} \right) \cdot \left( \mathbf{w}^t \odot \mathbf{Q}_{\widetilde{\mathbf{U}}}^\top \mathbf{U}_{\mathbf{A}} \mathbf{z}^{t+1} \right) \\
=& \mathbf{w}^t - \eta_t \left( \left\langle \mathbf{Q}_{\widetilde{\mathbf{U}}}^\top \mathbf{U}_{\mathbf{A}} \mathbf{z}^{t+1}, (\mathbf{w}^t)^{\odot 2} \right\rangle - \left\langle \boldsymbol{\Sigma}^{1/2} \mathbf{z}^{t+1}, \mathbf{v}^{*\odot 2} \right\rangle - \xi - \widetilde{\xi} \right) \cdot \left( \mathbf{w}^t \odot \mathbf{Q}_{\widetilde{\mathbf{U}}}^\top \mathbf{U}_{\mathbf{A}} \mathbf{z}^{t+1} \right) \\
=& \mathbf{w}^t - \eta_t \left[ \left\langle \widetilde{\boldsymbol{\Sigma}}^{1/2} \mathbf{z}^{t+1}, (\mathbf{w}^t)^{\odot 2} \right\rangle - \left\langle \widetilde{\boldsymbol{\Sigma}}^{1/2} \mathbf{z}^{t+1}, \mathbf{v}^{*\odot 2} \right\rangle - \xi - \widetilde{\xi} \right.
\end{aligned}
$$

$$+ \left\langle \mathbf{Q}_{\widetilde{\mathbf{U}}}^\top \left(\mathbf{U_A} - \widetilde{\mathbf{U}}\right) \mathbf{z}^{t+1}, \left(\mathbf{w}^t\right)^{\odot 2} \right\rangle - \left\langle \left(\widetilde{\boldsymbol{\Sigma}}^{1/2} - \boldsymbol{\Sigma}^{1/2}\right) \mathbf{z}^{t+1}, \mathbf{v}^{*\odot 2} \right\rangle \Big]$$

$$\cdot \left[ \mathbf{w}^t \odot \widetilde{\boldsymbol{\Sigma}}^{1/2} \mathbf{z}^{t+1} + \mathbf{w}^t \odot \mathbf{Q}_{\widetilde{\mathbf{U}}}^\top \left(\mathbf{U_A} - \widetilde{\mathbf{U}}\right) \mathbf{z}^{t+1} \right]$$

$$= \mathbf{w}^t - \eta_t \underbrace{\left( \left\langle \widetilde{\boldsymbol{\Sigma}}^{1/2} \mathbf{z}^{t+1}, \left(\mathbf{w}^t\right)^{\odot 2} \right\rangle - \left\langle \widetilde{\boldsymbol{\Sigma}}^{1/2} \mathbf{z}^{t+1}, \mathbf{v}^{*\odot 2} \right\rangle - \xi - \widehat{\xi} \right) \cdot \left( \mathbf{w}^t \odot \widetilde{\boldsymbol{\Sigma}}^{1/2} \mathbf{z}^{t+1} \right)}_{\mathcal{I}}$$

$$- \eta_t \underbrace{\left( \left\langle \widetilde{\boldsymbol{\Sigma}}^{1/2} \mathbf{z}^{t+1}, \left(\mathbf{w}^t\right)^{\odot 2} \right\rangle - \left\langle \widetilde{\boldsymbol{\Sigma}}^{1/2} \mathbf{z}^{t+1}, \mathbf{v}^{*\odot 2} \right\rangle - \xi - \widehat{\xi} \right) \cdot \left[ \mathbf{w}^t \odot \mathbf{Q}_{\widetilde{\mathbf{U}}}^\top \left(\mathbf{U_A} - \widetilde{\mathbf{U}}\right) \mathbf{z}^{t+1} \right]}_{\mathcal{II}}$$

$$- \eta_t \underbrace{\left\langle \mathbf{Q}_{\widetilde{\mathbf{U}}}^\top \left(\mathbf{U_A} - \widetilde{\mathbf{U}}\right) \mathbf{z}^{t+1}, \left(\mathbf{w}^t\right)^{\odot 2} \right\rangle \cdot \left( \mathbf{w}^t \odot \widetilde{\boldsymbol{\Sigma}}^{1/2} \mathbf{z}^{t+1} \right)}_{\mathcal{III}}$$

$$+ \eta_t \underbrace{\left\langle \left(\widetilde{\boldsymbol{\Sigma}}^{1/2} - \boldsymbol{\Sigma}^{1/2}\right) \mathbf{z}^{t+1}, \mathbf{v}^{*\odot 2} \right\rangle \cdot \left( \mathbf{w}^t \odot \widetilde{\boldsymbol{\Sigma}}^{1/2} \mathbf{z}^{t+1} \right)}_{\mathcal{IV}}$$

$$- \eta_t \underbrace{\left\langle \mathbf{Q}_{\widetilde{\mathbf{U}}}^\top \left(\mathbf{U_A} - \widetilde{\mathbf{U}}\right) \mathbf{z}^{t+1}, \left(\mathbf{w}^t\right)^{\odot 2} \right\rangle \cdot \left( \mathbf{w}^t \odot \mathbf{Q}_{\widetilde{\mathbf{U}}}^\top \left(\mathbf{U_A} - \widetilde{\mathbf{U}}\right) \mathbf{z}^{t+1} \right)}_{\mathcal{V}}$$

$$+ \eta_t \underbrace{\left\langle \left(\widetilde{\boldsymbol{\Sigma}}^{1/2} - \boldsymbol{\Sigma}^{1/2}\right) \mathbf{z}^{t+1}, \mathbf{v}^{*\odot 2} \right\rangle \cdot \left( \mathbf{w}^t \odot \mathbf{Q}_{\widetilde{\mathbf{U}}}^\top \left(\mathbf{U_A} - \widetilde{\mathbf{U}}\right) \mathbf{z}^{t+1} \right)}_{\mathcal{VI}}.$$

Here, $\mathcal{I}$ corresponds to the term on the right-hand side of equation 25, while the remaining terms $\mathcal{II}, \mathcal{III}, \mathcal{IV}, \mathcal{V}$ and $\mathcal{VI}$ only affect $\mathbf{V}^{T_2}$. For simplicity, define matrix $\mathbf{H} := \mathrm{diag}\{\mathbf{v}^*\} \widetilde{\boldsymbol{\Sigma}} \, \mathrm{diag}\{\mathbf{v}^*\}$ and let $K = T_1$. Combining the Cauchy–Schwarz inequality with the proof technique of Lemmas B.4 and B.5, we derive the estimation for $\left\langle \mathbf{H}, \mathbf{V}^{T_2} \right\rangle$ in the following form:

$$\left\langle \mathbf{H}, \mathbf{V}^{T_2} \right\rangle \lesssim \sigma^2 \left( \frac{N_0'}{K} + \eta_0 \sum_{i=N_0'+1}^{N_0} \lambda_i (\mathbf{v}_i^*)^2 \right) + \sigma^2 \eta_0^2 (h + K) \sum_{i=N_0+1}^{M} \lambda_i^2 (\mathbf{v}_i^*)^4$$

$$+ \left\langle \mathbf{H}, \mathbf{V}_{\mathcal{II}}^{T_2} \right\rangle + \left\langle \mathbf{H}, \mathbf{V}_{\mathcal{III}}^{T_2} \right\rangle + \left\langle \mathbf{H}, \mathbf{V}_{\mathcal{IV}}^{T_2} \right\rangle + \left\langle \mathbf{H}, \mathbf{V}_{\mathcal{V}}^{T_2} \right\rangle + \left\langle \mathbf{H}, \mathbf{V}_{\mathcal{VI}}^{T_2} \right\rangle$$

$$\overset{(d)}{\lesssim} \sigma^2 \left( \frac{N_0'}{K} + \eta_0 \sum_{i=N_0'+1}^{N_0} \lambda_i (\mathbf{v}_i^*)^2 \right) + \sigma^2 \eta_0^2 (h + K) \sum_{i=N_0+1}^{M} \lambda_i^2 (\mathbf{v}_i^*)^4$$

$$+ \left\| \widetilde{\mathbf{U}} \widetilde{\mathbf{U}}^\top - \mathbf{A} \right\| + \left\| \widetilde{\mathbf{U}} - \mathbf{U_A} \right\|, \tag{86}$$

where the diagonal matrices $\mathbf{V}_{\mathcal{II}}^{T_2}, \mathbf{V}_{\mathcal{III}}^{T_2}, \mathbf{V}_{\mathcal{IV}}^{T_2}, \mathbf{V}_{\mathcal{V}}^{T_2}$ and $\mathbf{V}_{\mathcal{VI}}^{T_2}$ are defined as follows:

$$\left(\mathbf{V}_{\mathcal{II}}^{T_2}\right)_{i,i} = \begin{cases} \left\| \widetilde{\mathbf{U}} - \mathbf{U_A} \right\|^2 \cdot \frac{1}{\lambda_i^2 (\mathbf{v}_i^*)^2}, & \text{if } i \leq D. \\ \left\| \widetilde{\mathbf{U}} - \mathbf{U_A} \right\|^2 \cdot \frac{T \eta_0}{\lambda_i}, & \text{otherwise,} \end{cases}$$

$$\left(\mathbf{V}_{\mathcal{III}}^{T_2}\right)_{i,i} = \begin{cases} \left\| \widetilde{\mathbf{U}} - \mathbf{U_A} \right\|^2 \cdot \frac{\|\mathbf{v}^{*\odot 2}\|^2}{\lambda_i (\mathbf{v}_i^*)^2}, & \text{if } i \leq D, \\ \left\| \widetilde{\mathbf{U}} - \mathbf{U_A} \right\|^2 \cdot T \eta_0 \|\mathbf{v}^{*\odot 2}\|^2, & \text{otherwise,} \end{cases}$$

$$\left(\mathbf{V}_{\mathcal{IV}}^{T_2}\right)_{i,i} = \begin{cases} \left\| \widetilde{\mathbf{U}} \widetilde{\mathbf{U}}^\top - \mathbf{A} \right\|^2 \cdot \frac{\|\mathbf{v}^{*\odot 2}\|^2}{\lambda_M \lambda_i (\mathbf{v}_i^*)^2}, & \text{if } i \leq D, \\ \left\| \widetilde{\mathbf{U}} \widetilde{\mathbf{U}}^\top - \mathbf{A} \right\|^2 \cdot \frac{T \eta_0 \|\mathbf{v}^{*\odot 2}\|^2}{\lambda_M}, & \text{otherwise,} \end{cases}$$

$$\left(\mathbf{V}_{\mathcal{V}}^{T_2}\right)_{i,i} = \begin{cases} \left\| \widetilde{\mathbf{U}} - \mathbf{U_A} \right\|^4 \cdot \frac{\|\mathbf{v}^{*\odot 2}\|^2}{\lambda_i^2 (\mathbf{v}_i^*)^2}, & \text{if } i \leq D, \\ \left\| \widetilde{\mathbf{U}} - \mathbf{U_A} \right\|^4 \cdot \frac{T \eta_0 \|\mathbf{v}^{*\odot 2}\|^2}{\lambda_i}, & \text{otherwise,} \end{cases}$$

$$\left(\mathbf{V}_{\mathcal{VI}}^{T_2}\right)_{i,i} = \begin{cases} \left\|\widetilde{\mathbf{U}} - \mathbf{U}_{\mathbf{A}}\right\|^2 \left\|\widetilde{\mathbf{U}}\widetilde{\mathbf{U}}^\top - \mathbf{A}\right\|^2 \cdot \dfrac{\|\mathbf{v}^{*\odot 2}\|^2}{\lambda_M \lambda_i^2(\mathbf{v}_i^*)^2}, & \text{if } i \leq D, \\[3mm] \left\|\widetilde{\mathbf{U}} - \mathbf{U}_{\mathbf{A}}\right\|^2 \left\|\widetilde{\mathbf{U}}\widetilde{\mathbf{U}}^\top - \mathbf{A}\right\|^2 \cdot \dfrac{T\eta_0\|\mathbf{v}^{*\odot 2}\|^2}{\lambda_M \lambda_i}, & \text{otherwise.} \end{cases}$$

Inequality (d) is derived from combining Assumption 4.1 with above definitions. The estimation for $\left\langle \mathbf{H}, \mathbf{V}^{T_2} \right\rangle$ has been provided. It therefore remains only to bound $\left\langle \mathbf{H}, \mathbf{B}^{T_2} \right\rangle$, which can be done by an analysis analogous to that of Lemma B.6. This completes the proof. $\qquad\square$

# E  AUXILIARY LEMMA

**Definition E.1** (Sub-Gaussian Random Variable). A random variable $x$ with mean $\mathbb{E}x$ is sub-Gaussian if there is $\sigma \in \mathbb{R}_+$ such that

$$\mathbb{E}\left[e^{\lambda(x-\mathbb{E}x)}\right] \leq e^{\frac{\lambda^2\sigma^2}{2}}, \quad \forall \lambda \in \mathbb{R}.$$

**Proposition E.1.** *[(Wainwright, 2019)] For a random variable $x$ which satisfies the sub-Gaussian condition E.1 with parameter $\sigma$, we have*

$$\mathbb{P}\left(|x - \mathbb{E}x| > c\right) \leq 2e^{-\frac{c^2}{2\sigma^2}}, \quad \forall c > 0. \tag{87}$$

**Lemma E.1.** *Let $X_1, \cdots, X_n$ be independent and symmetric stochastic variables with zero mean. Denote $Y = \sum_{i=1}^n \mathbf{v}_i X_i \mathbb{1}_{|X_i| \leq R}$ for any unit vector $\mathbf{v} \in \mathbb{R}^n$ and positive scalar $R$. Then, we have $YX_1\mathbb{1}_{|X_1|\leq R}$ is sub-Gaussian with parameter at most $\sigma = \mathcal{O}\left(R^2\|\mathbf{v}\|_2\right)$.*

*Proof.* For simplicity, we denote $\hat{X}_i := X_i\mathbb{1}_{|X_i|\leq R}$ for any $i \in [1:n]$, and $Y_{-1} = \sum_{i=2}^n \mathbf{v}_i \hat{X}_i$. One can notice the following holds

$$\mathbb{E}\left[e^{\lambda\left(Y\hat{X}_1-\mathbb{E}[Y\hat{X}_1]\right)}\right] = \mathbb{E}\left[e^{\lambda\mathbf{v}_i\left(\hat{X}_1^2-\mathbb{E}[\hat{X}_1^2]\right)}\mathbb{E}\left[e^{\lambda\left(Y_{-1}\hat{X}_1-\mathbb{E}[Y_{-1}\hat{X}_1]\right)} \mid \hat{X}_1\right]\right], \tag{88}$$

for any $\lambda \in \mathbb{R}$. Letting $\hat{X}_i'$ be an independent copy of $\hat{X}_i$ for any $i \in [1:n]$, then we have

$$\mathbb{E}\left[e^{\lambda\left(Y_{-1}\hat{X}_1-\mathbb{E}[Y_{-1}\hat{X}_1]\right)} \mid \hat{X}_1\right] \overset{(a)}{\leq} \mathbb{E}\left\{\mathbb{E}\left[e^{\sum_{i=2}^n \lambda\mathbf{v}_i(\hat{X}_i\hat{X}_1-\mathbb{E}[\hat{X}_i]\hat{X}_1')} \mid \hat{X}_1, \hat{X}_1'\right]\right\}$$

$$\overset{(b)}{\leq} \mathbb{E}\left\{\mathbb{E}\left[e^{\sum_{i=2}^n \lambda\mathbf{v}_i(\hat{X}_i\hat{X}_1-\hat{X}_i'\hat{X}_1')} \mid \hat{X}_1, \hat{X}_1'\right]\right\}, \tag{89}$$

where (a) and (b) are derived from the convexity of the exponential, and Jensen's inequality. Letting $\xi$ be an independent Rademacher variable, since the distribution of $\hat{X}_i - \hat{X}_i'$ is the same as that of $\xi(\hat{X}_i - \hat{X}_i')$ for any $i \in [1:n]$, we obtain

$$\mathbb{E}\left[e^{\sum_{i=2}^n \lambda\mathbf{v}_i(\hat{X}_i\hat{X}_1-\hat{X}_i'\hat{X}_1')} \mid \hat{X}_1, \hat{X}_1'\right]$$

$$= \prod_{i=2}^n \mathbb{E}\left[e^{\lambda\mathbf{v}_i\left[\hat{X}_1(\hat{X}_i-\hat{X}_i')+\hat{X}_i'(\hat{X}_1-\hat{X}_1')\right]}\right]$$

$$\overset{(c)}{\leq} \prod_{i=2}^n \left(\mathbb{E}\left[e^{2\lambda^2\mathbf{v}_i^2\hat{X}_1^2(\hat{X}_i-\hat{X}_i')^2} \mid \hat{X}_1, \hat{X}_1'\right] \mathbb{E}\left[e^{2\lambda\mathbf{v}_i\hat{X}_i'(\hat{X}_1-\hat{X}_1')} \mid \hat{X}_1, \hat{X}_1'\right]\right)^{1/2}. \tag{90}$$

Noticing that $|\hat{X}_i - \hat{X}_i'| \leq 2R$ and $|\hat{X}_i| \leq R$, and applying the Hoeffding bound to $\hat{X}_i$ for any $i \in [1:n]$, we are guarantee that

$$\prod_{i=2}^n \left(\mathbb{E}\left[e^{2\lambda^2\mathbf{v}_i^2\hat{X}_1^2(\hat{X}_i-\hat{X}_i')^2} \mid \hat{X}_1, \hat{X}_1'\right] \mathbb{E}\left[e^{2\lambda\mathbf{v}_i\hat{X}_i'(\hat{X}_1-\hat{X}_1')} \mid \hat{X}_1, \hat{X}_1'\right]\right)^{1/2} \leq e^{\mathcal{O}\left(\lambda^2 R^4 \sum_{i=2}^n \mathbf{v}_i^2\right)}. \tag{91}$$

Combining Eq. equation 88-Eq. equation 91 and applying similar technique, we have

$$\mathbb{E}\left[e^{\lambda\left(Y\hat{X}_1-\mathbb{E}[Y\hat{X}_1]\right)}\right] \leq e^{\mathcal{O}\left(\lambda^2 R^4 \sum_{i=2}^n \mathbf{v}_i^2\right)}\mathbb{E}\left[e^{\lambda\mathbf{v}_i\left(\hat{X}_1^2-\mathbb{E}[\hat{X}_1^2]\right)}\right] \leq e^{\mathcal{O}\left(\lambda^2 R^4\|\mathbf{v}\|_2^2\right)}.$$

$$\square$$

**Lemma E.2.** *Consider a stochastic variable $X$ which is zero-mean and sub-Gaussian with parameter $\sigma$ for some $\sigma > 0$. Then, there exists $R > 0$ which depends on $\sigma$ such that*

$$\mathbb{E}\left[X^2 \mathbb{1}_{|X| \leq R}\right] \geq \frac{1}{2}\mathbb{E}\left[X^2\right]. \tag{92}$$

*Proof.* According to Eq. equation 87, we have $\mathbb{P}(|X| \geq r) \leq 2e^{-\frac{r^2}{2\sigma^2}}$ for any $r > 0$. Therefore, we obtain

$$
\begin{aligned}
\mathbb{E}\left[X^2 \mathbb{1}_{|X|>R}\right] &\stackrel{(a)}{=} 2\int_0^\infty r\mathbb{P}(|X|\mathbb{1}_{|X|>R} > r)\mathrm{d}r \\
&= 2\int_R^\infty r\mathbb{P}(|X| > r)\mathrm{d}r + R^2\mathbb{P}(|X| > R) \\
&\leq 4\int_R^\infty re^{-\frac{r^2}{2\sigma^2}}\mathrm{d}r + 2R^2 e^{-\frac{R^2}{2\sigma^2}} = 4\sigma^2 e^{-\frac{R^2}{\sigma^2}} + 2R^2 e^{-\frac{R^2}{2\sigma^2}},
\end{aligned} \tag{93}
$$

where (a) is derived from [Lemma 2.2.13, Wainwright (2019)]. $\qquad\square$

**Lemma E.3.** *Let $c > 0$, $\gamma < 1$ and $a_t > 0$ for any $t \in [0 : T-1]$. Consider a sequence of random variables $\{v^i\}_{i=0}^{T-1} \subset [0, c]$, which satisfies either $v^t = v^{t+1} = \cdots = v^T$, or $\mathbb{E}\left[v^{t+1} \mid \mathcal{F}^t\right] \leq (1-\eta_t)v^t$ with stepsize $\eta_t \geq 0$, given $\mathbb{E}[e^{\lambda(v^{t+1}-\mathbb{E}[v^{t+1}|\mathcal{F}^t])} \mid \mathcal{F}^t] \leq e^{\frac{\lambda^2 a_t^2}{2}}$ almost surely for any $\lambda \in \mathbb{R}$. Then, there is*

$$\mathbb{P}\left(v^T > c \bigwedge v^0 \leq \gamma c\right) \leq \max_{t\in[1:T]} \exp\left\{-\frac{(1-\gamma)^2 c^2}{2\sum_{j=0}^{t-1} a_j^2 \prod_{i=j+1}^{t-1}(1-\eta_i)^2}\right\}.$$

*Proof.* Similarly, we begin with constructing a sequence of couplings $\{\tilde{v}^i\}_{i=0}^T$ as follows: $\tilde{v}^0 = v^0$; if $v^t = v^{t+1} = \cdots = v^T$, let $\tilde{v}^{t+1} = (1-\eta_t)\tilde{v}^t$; otherwise, let $\tilde{v}^{t+1} = v^{t+1}$. Notice that $\prod_{i=0}^{t-1}(1-\eta_i)^{-1}\tilde{v}^t$ is a supermartingale. We define $D_{t+1} := \prod_{i=0}^{t}(1-\eta_i)^{-1}\tilde{v}^{t+1} - \prod_{i=0}^{t-1}(1-\eta_i)^{-1}\tilde{v}^t$ for any $t \in [0 : T-1]$. Therefore, applying iterated expectation yields

$$
\begin{aligned}
\mathbb{E}\left[e^{\lambda\left(\sum_{i=1}^{t} D_i\right)}\right] &= \mathbb{E}\left[e^{\lambda\left(\sum_{i=1}^{t-1} D_i\right)}\mathbb{E}\left[e^{\lambda D_t} \mid \mathcal{F}^{t-1}\right]\right] \\
&= \mathbb{E}\left[e^{\lambda\left(\sum_{i=1}^{t-1} D_i\right)}\mathbb{E}\left[e^{\frac{\lambda}{\prod_{i=0}^{t-1}(1-\eta_i)}\left(v^t-(1-\eta_{t-1})v^{t-1}\right)} \mid \mathcal{F}^{t-1}\right]\right] \\
&\stackrel{(a)}{\leq} \mathbb{E}\left[e^{\lambda\left(\sum_{i=1}^{t-1} D_i\right)}\mathbb{E}\left[e^{\frac{\lambda}{\prod_{i=0}^{t-1}(1-\eta_i)}\left(v^t-\mathbb{E}[v^t|\mathcal{F}^{t-1}]\right)} \mid \mathcal{F}^{t-1}\right]\right] \\
&\stackrel{(b)}{\leq} e^{\frac{\lambda^2 a_{t-1}^2}{2\prod_{i=0}^{t-1}(1-\eta_i)^2}}\mathbb{E}\left[e^{\lambda\left(\sum_{i=1}^{t-1} D_i\right)}\right] \\
&\leq e^{\frac{\lambda^2 \sum_{j=0}^{t-1} a_j^2 \prod_{i=0}^{j}(1-\eta_i)^{-2}}{2}},
\end{aligned} \tag{94}
$$

for any $\lambda \in \mathbb{R}^+$ and $t \in [1 : T]$, where (a) is derived from that $\lambda(\mathbb{E}[v^t \mid \mathcal{F}^{t-1}] - (1-\eta_{t-1})v^{t-1}) \leq 0$ and (b) follows from the condition that $\mathbb{E}[e^{\lambda(v^{t+1}-(1-\eta_t)v^t)} \mid \mathcal{F}^t] \leq e^{\frac{\lambda^2 a_t^2}{2}}$ almost surely for any $\lambda \in \mathbb{R}$. Then we obtain

$$
\begin{aligned}
\mathbb{P}\left(v^T > c \bigwedge v^0 \leq \gamma c\right) &\leq \max_{t\in[1:T]} \mathbb{P}\left(\prod_{i=0}^{t-1}(1-\eta_i)^{-1}\tilde{v}^t > \prod_{i=0}^{t-1}(1-\eta_i)^{-1}c \bigwedge \tilde{v}^0 \leq \gamma c\right) \\
&\leq \max_{t\in[1:T]} \min_{\lambda>0} \frac{\mathbb{E}\left[e^{\lambda\left(\sum_{i=1}^{t} D_i\right)}\right]}{e^{\lambda\left(\prod_{i=0}^{t-1}(1-\eta_i)^{-1}c-\gamma c\right)}} \\
&\stackrel{(b)}{\leq} \max_{t\in[1:T]} \exp\left\{-\frac{\left(\prod_{i=0}^{t-1}(1-\eta_i)^{-1}c - \gamma c\right)^2}{2\sum_{j=0}^{t-1} a_j^2 \prod_{i=0}^{j}(1-\eta_i)^{-2}}\right\}
\end{aligned}
$$

$$\leq \max_{t \in [1:T]} \exp \left\{ -\frac{(1-\gamma)^2 \left(\prod_{i=0}^{t-1}(1-\eta_i)^{-1}c\right)^2}{2\sum_{j=0}^{t-1} a_j^2 \prod_{i=0}^{j}(1-\eta_i)^{-2}} \right\}$$

$$= \max_{t \in [1:T]} \exp \left\{ -\frac{(1-\gamma)^2 c^2}{2\sum_{j=0}^{t-1} a_j^2 \prod_{i=j+1}^{t-1}(1-\eta_i)^2} \right\}, \tag{95}$$

where (b) is derived from Eq. equation 94. □

**Corollary E.1.** *Let $c > 0$, $\gamma < 1$ and $a_t > 0$ for any $t \in [0 : T-1]$. Consider a sequence of random variables $\{v^i\}_{i=0}^{T-1} \subset [0, c]$, which satisfies $\prod_{i=0}^{T-1}(1+\eta_t)^{-1}c - v^0 \geq \gamma c$ and $\mathbb{E}\left[v^{t+1} \mid \mathcal{F}^t\right] \leq (1+\eta_t)v^t$ with stepsize $\eta_t \geq 0$, given $\mathbb{E}[e^{\lambda(v^{t+1}-\mathbb{E}[v^{t+1}|\mathcal{F}^t])} \mid \mathcal{F}^t] \leq e^{\frac{\lambda^2 a_t^2}{2}}$ almost surely for any $\lambda \in \mathbb{R}$. Then, there is*

$$\mathbb{P}\left(v^T > c\right) \leq \max_{t \in [1:T]} \exp \left\{ -\frac{\gamma^2 c^2}{2\sum_{j=0}^{t-1} a_j^2 \prod_{i=0}^{j}(1+\eta_i)^{-2}} \right\}.$$

**Lemma E.4.** *For $L, K \in \mathbb{N}_+$, consider $T \in \mathbb{N}^+$ such that $LK \leq T < (L+1)K$. Then we have*

$$\sum_{t=0}^{T} \left(\prod_{i=t}^{T}(1-c\eta_t)\right) \eta_t^2 \leq \frac{2\eta_0}{c}, \tag{96}$$

*where $\eta_t = \frac{\eta_0}{2^l}$ if $lK \leq t \leq \min\{(l+1)K-1, T\}$ for any $l \in [0 : L]$ and $c > 0$ is a constant.*

*Proof.* For any $l \in [0 : L]$, we have

$$\sum_{t=lK}^{(l+1)K-1} \left(\prod_{i=t}^{T}(1-c\eta_t)\right) \eta_t^2 = \eta_{lK}^2 \left(\prod_{i=(l+1)K}^{T}(1-c\eta_t)\right) \sum_{t=lK}^{(l+1)K-1} (1-c\eta_{lK})^{(l+1)K-1-t}$$

$$\leq \frac{\eta_{lK}}{c} \left(\prod_{i=(l+1)K}^{T}(1-c\eta_t)\right). \tag{97}$$

Therefore, we obtain the following estimation

$$\sum_{t=0}^{T} \left(\prod_{i=t}^{T}(1-c\eta_t)\right) \eta_t^2 \leq \sum_{t=0}^{LK-1} \left(\prod_{i=t}^{T}(1-c\eta_t)\right) \eta_t^2 + \sum_{t=LK}^{T} (1-c\eta_{LK})^{T-t} \eta_{LK}^2$$

$$\overset{(a)}{\leq} \frac{\sum_{l=0}^{L} \eta_{lK}}{c} \leq \frac{2\eta_0}{c}. \tag{98}$$

□

**Lemma E.5.** *Under Assumption 3.3 and the setting of Theorem B.2, we have*

$$\eta(\mathbf{I} - \eta\widehat{\mathbf{H}})^{2t}\mathbf{H} \preceq \frac{25}{t+1}\mathbf{I},$$

*for any $t \in [0 : T-1]$.*

*Proof.* For index $i \in [1 : D]$, we have

$$\eta(\mathbf{I}_{1:D} - \eta\widehat{\mathbf{H}}_{1:D})^{2t}\mathbf{H}_{1:D} = 25\eta(\mathbf{I}_{1:D} - \eta\widehat{\mathbf{H}}_{1:D})^{2t}\widehat{\mathbf{H}}_{1:D} \preceq \frac{25}{t+1}\mathbf{I}_{1:D},$$

since $(1-x)^t \leq \frac{1}{(t+1)x}$ for any $x \in (0, 1)$. For index $i \in [D+1 : M]$, we obtain

$$\eta\mathbf{H}_{i,i} \leq \frac{1}{T} \leq \frac{1}{t+1}, \tag{99}$$

according to the parameter setting in Theorem B.2 for any $t \in [1 : T-1]$. □

**Lemma E.6.** *Suppose Assumption 3.1 hold and let $\mathbf{z} = \Pi_M \mathbf{x} \in \mathbb{R}^M$. Then there exists a constant $\gamma > 0$ such that*

$$\mathbb{E}\left[\mathbf{A}\mathbf{z}\|\mathbf{z}\|^2_{\mathbf{A}^\top \mathbf{B}\mathbf{A}}\mathbf{z}^\top \mathbf{A}^\top\right] \preceq \gamma \left\langle \mathbf{A}\mathbb{E}\left[\mathbf{z}\mathbf{z}^\top\right]\mathbf{A}^\top, \mathbf{B}\right\rangle \mathbf{A}\mathbb{E}\left[\mathbf{z}\mathbf{z}^\top\right]\mathbf{A}^\top, \tag{100}$$

*for any diagonal PSD matrix $\mathbf{A} \in \mathbb{R}^{M \times M}$ and PSD matrix $\mathbf{B} \in \mathbb{R}^{M \times M}$.*

*Proof.* We denote $\mathbf{D} := \mathbb{E}[\mathbf{A}\mathbf{z}\|\mathbf{z}\|^2_{\mathbf{A}^\top \mathbf{B}\mathbf{A}}\mathbf{z}^\top \mathbf{A}^\top]$. For any $i, j \in [1 : M]$ and $i \neq j$, we have $\mathbf{D}_{i,j} = 2\lambda_i\lambda_j\mathbf{A}_{i,i}\mathbf{A}_{j,j}(\mathbf{A}^\top \mathbf{B}\mathbf{A})_{i,j}$. In addition, we also have

$$\mathbf{D}_{i,i} = \mathbb{E}\left[\|\mathbf{z}\|^2_{\mathbf{A}^\top \mathbf{B}\mathbf{A}}\right]\mathbf{A}^2_{i,i}\lambda_i + (\mathbf{A}^\top \mathbf{B}\mathbf{A})_{i,i}\mathbf{A}^2_{i,i}\operatorname{Var}\left[\mathbf{z}^2_i\right] \leq (C+1)\mathbb{E}\left[\|\mathbf{z}\|^2_{\mathbf{A}^\top \mathbf{B}\mathbf{A}}\right]\mathbf{A}^2_{i,i}\lambda_i.$$

Therefore, we obtain that

$$\begin{aligned}
\mathbf{D} \preceq & (C+1)\mathbb{E}\left[\|\mathbf{z}\|^2_{\mathbf{A}^\top \mathbf{B}\mathbf{A}}\right]\mathbf{A}\mathbb{E}[\mathbf{z}\mathbf{z}^\top]\mathbf{A}^\top + 2\mathbf{A}\mathbb{E}[\mathbf{z}\mathbf{z}^\top]\mathbf{A}^\top \mathbf{B}\mathbf{A}\mathbb{E}[\mathbf{z}\mathbf{z}^\top]\mathbf{A}^\top \\
\preceq & (C+1)\mathbb{E}\left[\|\mathbf{z}\|^2_{\mathbf{A}^\top \mathbf{B}\mathbf{A}}\right]\mathbf{A}\mathbb{E}[\mathbf{z}\mathbf{z}^\top]\mathbf{A}^\top \\
& + 2\left\|\left(\mathbf{A}\mathbb{E}[\mathbf{z}\mathbf{z}^\top]\mathbf{A}^\top\right)^{1/2}\mathbf{B}\left(\mathbf{A}\mathbb{E}[\mathbf{z}\mathbf{z}^\top]\mathbf{A}^\top\right)^{1/2}\right\|^2_2 \mathbf{A}\mathbb{E}[\mathbf{z}\mathbf{z}^\top]\mathbf{A}^\top \\
\overset{(a)}{\preceq} & (C+2)\left\langle \mathbf{A}\mathbb{E}\left[\mathbf{z}\mathbf{z}^\top\right]\mathbf{A}^\top, \mathbf{B}\right\rangle \mathbf{A}\mathbb{E}[\mathbf{z}\mathbf{z}^\top]\mathbf{A}^\top,
\end{aligned}$$

where (a) is derived from that $\langle \mathbf{A}\mathbb{E}[\mathbf{z}\mathbf{z}^\top]\mathbf{A}^\top, \mathbf{B}\rangle = \mathbb{E}[\|\mathbf{z}\|^2_{\mathbf{A}^\top \mathbf{B}\mathbf{A}}]$ and $\|\mathbf{H}^{1/2}\mathbf{B}\mathbf{H}^{1/2}\|^2_2 \leq \langle \mathbf{H}, \mathbf{B}\rangle$ for any PSD matrix $\mathbf{B}, \mathbf{D} \in \mathbb{R}^{M \times M}$ since

$$\begin{aligned}
\mathbf{a}^\top \mathbf{H}^{1/2}\mathbf{B}\mathbf{H}^{1/2}\mathbf{a} &= \left\langle \mathbf{H}^{1/2}\mathbf{a}\mathbf{a}^\top \mathbf{H}^{1/2}, \mathbf{B}\right\rangle \\
&\leq \left\langle \mathbf{H}^{1/2}\mathbf{a}\mathbf{a}^\top \mathbf{H}^{1/2}, \mathbf{B}\right\rangle + \left\langle \mathbf{H}^{1/2}\mathbf{a}_\perp \mathbf{a}_\perp^\top \mathbf{H}^{1/2}, \mathbf{B}\right\rangle \\
&= \langle \mathbf{H}, \mathbf{B}\rangle.
\end{aligned}$$

Therefore, by choosing $\gamma = (C+2)$, we obtain Eq. equation 100. $\qquad\square$

**Lemma E.7.** *Under the setting of Theorem B.2, suppose following inequality holds*

$$\mathbf{B}^{t+1}_{\text{diag}} \preceq \left(\mathcal{I} - \eta\widehat{\mathcal{G}}\right)^{t+1} \circ \mathbf{B}^0_{\text{diag}} + \tau\eta \sum_{i=0}^{t} \frac{\langle \mathbf{H}, \mathbf{B}^i\rangle}{t+1-i} \cdot \mathbf{I},$$

*for any $t \in [0 : T-1]$ and some constant $\tau > 0$. Then, we have*

$$\sum_{i=0}^{t} \frac{\langle \mathbf{H}, \mathbf{B}^i\rangle}{t+1-i} \leq \left\langle \sum_{i=0}^{t} \frac{(\mathbf{I}-\eta\widehat{\mathbf{H}})^{2i}\mathbf{H}}{t+1-i}, \mathbf{B}^0\right\rangle + 2\tau\eta\log(t)\operatorname{tr}(\mathbf{H})\sum_{i=0}^{t} \frac{\langle \mathbf{H}, \mathbf{B}^i\rangle}{t+1-i},$$

*for any $t \in [1 : T]$.*

*Proof.* According to the condition of this lemma, we have

$$\langle \mathbf{H}, \mathbf{B}^t\rangle \leq \left\langle (\mathbf{I}-\eta\widehat{\mathbf{H}})^{2t}\mathbf{H}, \mathbf{B}^0\right\rangle + \tau\eta\operatorname{tr}(\mathbf{H})\sum_{i=0}^{t-1} \frac{\langle \mathbf{H}, \mathbf{B}^i\rangle}{t-i}. \tag{101}$$

Applying Eq. equation 101 to each $\langle \mathbf{H}, \mathbf{B}^t\rangle$, we obtain

$$\begin{aligned}
\sum_{i=0}^{t} \frac{\langle \mathbf{H}, \mathbf{B}^i\rangle}{t+1-i} \leq & \left\langle \sum_{i=0}^{t} \frac{(\mathbf{I}-\eta\widehat{\mathbf{H}})^{2i}\mathbf{H}}{t+1-i}, \mathbf{B}^0\right\rangle + \tau\eta\operatorname{tr}(\mathbf{H})\sum_{i=0}^{t}\sum_{k=0}^{i-1} \frac{\langle \mathbf{H}, \mathbf{B}^i\rangle}{(t+1-i)(i-k)} \\
\leq & \left\langle \sum_{i=0}^{t} \frac{(\mathbf{I}-\eta\widehat{\mathbf{H}})^{2i}\mathbf{H}}{t+1-i}, \mathbf{B}^0\right\rangle + \tau\eta\operatorname{tr}(\mathbf{H})\sum_{k=0}^{t-1} \frac{\langle \mathbf{H}, \mathbf{B}^k\rangle}{t+1-k}\sum_{i=k+1}^{t}\left(\frac{1}{t+1-i} + \frac{1}{i-k}\right) \\
\leq & \left\langle \sum_{i=0}^{t} \frac{(\mathbf{I}-\eta\widehat{\mathbf{H}})^{2i}\mathbf{H}}{t+1-i}, \mathbf{B}^0\right\rangle + 2\tau\eta\log(t)\operatorname{tr}(\mathbf{H})\sum_{k=0}^{t} \frac{\langle \mathbf{H}, \mathbf{B}^k\rangle}{t+1-k}.
\end{aligned}$$

$\qquad\square$

# F SIMULATIONS

In this paper, we present simulations in a finite but large dimension ($d = 10,000$). We artificially generate samples from the model $y = \left\langle \mathbf{x}, (\mathbf{v}^*)^{\odot 2} \right\rangle + \xi$, where $\mathbf{x} \sim \mathcal{N}(\mathbf{0}, \mathbf{H})$, $\mathbf{H} = \mathrm{diag}\{i^{-\alpha}\}$, $\mathbf{w}_i^* = i^{-\frac{\beta - \alpha}{4}}$, and $\xi \sim \mathcal{N}(0, 1)$ is independent of $\mathbf{x}$. In our simulations, given a total of $T$ iteration, we assume that Algorithm 1 can access $T$ independent samples $\{(\mathbf{x}_i, y_i)\}_{i=1}^{T}$ generated by the above model. $h$ in Algorithm 1 is set to $\frac{T}{\log_2(T)}$. We numerically approximate the expected error by averaging the results of 100 independent repetitions of the experiment. In the following, we detail the specific experimental settings and present the results obtained for each scenario.

- **Figure F (a):** We compare the curve of mean error of SGD against the number of iteration steps for both linear and quadratic models, under the setting $\alpha = 3$, $\beta = 2$ and $T = 500$. The results show that the quadratic model exhibits a phase of diminishing error, while the linear model demonstrates a continuous, steady decrease in error.

- **Figure F (b):** We compare the curve of mean error of SGD against the number of iteration steps for both linear and quadratic models, under the setting $\alpha = 2.5$, $\beta = 1.5$ and $T = 500$. The results show that the quadratic model exhibits a phase of diminishing error, while the linear model demonstrates a continuous, steady decrease in error.

- **Figure F (c):** We compare the curve of mean error of SGD against the number of sample size for both linear and quadratic models, under the setting $\alpha = 3$, $\beta = 2$ and $T$ ranging from 1000 to 5000. The results indicate that the quadratic model outperforms the linear model and exhibits convergence behavior that is closer to the theoretical algorithm rate.

- **Figure F (d):** We compare the curve of mean error of SGD against the number of sample size for both linear and quadratic models, under the setting $\alpha = 2.5$, $\beta = 1.5$ and $T$ ranging from 1000 to 5000. The results indicate that the quadratic model outperforms the linear model and exhibits convergence behavior that is closer to the theoretical algorithm rate.

- **Figure F (e):** We compare the curve of mean error of SGD against the number of sample size for quadratic models with model size $M = 10, 30, 50, 100, 200$, under the setting $\alpha = 3$, $\beta = 2$ and $T$ ranging from 1 to 10000. The results show that for a fixed $M$, when $T$ is small, the convergence rate approaches the rate observed as $M \to \infty$. As $T$ increases sufficiently, the convergence rate stabilizes. Increasing $M$ results in an increase in the value of at which this stabilization occurs, which is consistent with the scaling law.

- **Figure F (f):** We compare the curve of mean error of SGD against the number of sample size for quadratic models with model size $M = 10, 30, 50, 100, 200$, under the setting $\alpha = 2.5$, $\beta = 1.5$ and $T$ ranging from 1 to 10000. The results exhibit similar patterns to those observed in the previous figure.

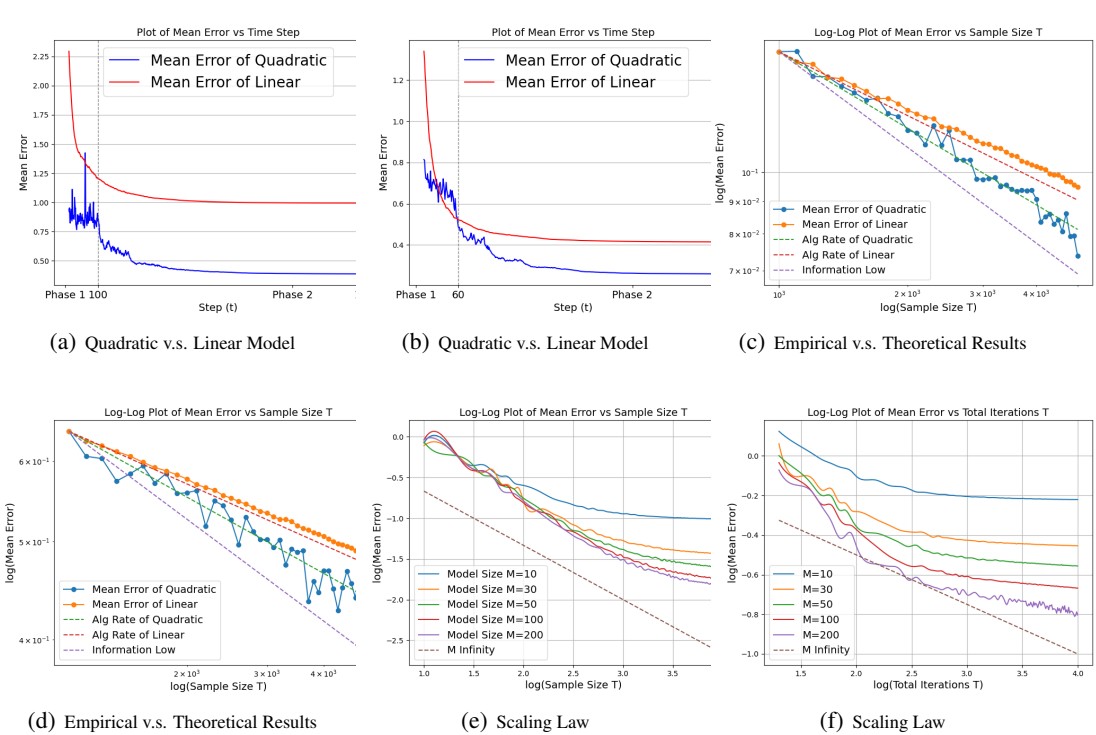

Figure 2: Numerical simulation results.

