# OpenReview forum: "Scaling Law for SGD in Quadratically Parameterized Linear Regression"
_ICLR.cc/2026/Conference — ICLR 2026 Conference Withdrawn Submission_

### Official Review · Reviewer_uaZA · 2025-10-21

**Soundness:** 3
**Presentation:** 2
**Contribution:** 2
**Rating:** 6
**Confidence:** 3

**Summary:**

This paper studies the scaling law for stochastic gradient descent (SGD) applied to quadratically parameterized linear regression under power-law decaying eigenvalues and ground truth. The authors model the problem as $f(x)=⟨Sx,v^{⊙2}⟩$ where S is a sketch matrix, analyzing how this parameterization enables feature learning compared to standard linear models. The analysis reveals two distinct regimes depending on the relationship between the covariance decay rate $\alpha$ and ground truth decay rate $\beta$: when $\alpha\le\beta$ Both linear and quadratic models achieve the information-theoretic lower bound, while when $\alpha>\beta$, the quadratic model outperforms the linear model, demonstrating the value of feature learning. The proof technique involves a two-phase analysis: an adaptation phase where SGD implicitly selects an effective dimension, and an estimation phase where convergence occurs in a reparameterized feature space.

**Strengths:**

1. The paper provide a novel theoretical contribution. It establishes a connection between quadratic parameterization and feature learning in scaling laws, addressing an important gap beyond the linear regime. They obtain the rate of decay of online SGD, they identify two regimes. They indicate that one of them is optimal.
	2. Dimension-free convergence: Achieves rates independent of ambient dimension despite estimating a 2d-dimensional function, demonstrating the power of low-dimensional structure.

**Weaknesses:**

1.	The proof is highly technical, with insufficient intuition in the main text. The coupling construction and auxiliary sequence techniques require clearer explanations before delving into details.
2.	The assumptions are restrictive, and not all are justified. However, I still think that even within this setting, the results of quadratic models are important.
3.	Presentation issues: Multiple typos, inconsistent notation, and grammatical errors throughout. There are some claims in the paper that are overstated.

**Questions:**

1.	Could you state the connection of this model to the phase retrieval problem, this is not even mentioned in the test. It would be nice to do some comparison to existing results in this context.
2.	Line 130, the sentence, is not clear. Are you missing “when $\alpha >\beta$ at the end of the sentence?
3.	Line 187: “Assumptions such as sparse or low-dimensional isotropic objective functions weaken the generality and fail to recover the polynomial decay of generalization error with respect to sample size and model parameters”. I agree that isotropic features fail to recover scaling behavior, but it is not clear that sparse features will fail as well, especially in a nonlinear model. In addition, there are work that study a multi-index model for general covariance structure (see for example Goldt et al. 2020 and Collins-Woodfin et al 2023)
4.	Could you explain why assumption A4 is less restrictive than A2? It seems to me that the assumption that $E[p(Sx)\zeta_M] =0$ or the  $E[p(Q_A^\top Sx)\zeta_M] =0$ is very strong.
5.	Line 318: Is it clear that a polynomial decaying schedule or other form of decay schedule design can’t achieve this “synergistic effects”? The choice provided is just one possible design, and it is not clear if it is the optimal one.
6.	In Theorem 4.1, is Eq. (2) an asymptotic relation or just an upper bound? The writing is confusing.
7.	Line 328, what do you mean by “sufficiently small initialization”? What is assumed about the initialization?
8.	Line 383: Could you provide more detail on the information theoretic lower bound and how your result achieves this bound?
9.	In Line 340 and 343, should that be Eq. 2?
10.	The norm in $\|\tilde{U}\tilde{U}^\top-A\|$ is not clear to me. If $\tilde{U}$ is an estimator, it should depend on the data. Does it include expectations over the data?
11.	Line 417: Where do you present this estimator? Is there an analytical result or numerical simulation that shows that?
12.	In all theorems, the author should clarify with respect to what is the in probability statement. The definition of the main text indicates that for a given initialization $\mathcal{R}_M(v^T)- E[\xi^2]$ is deterministic.
13. Could you provide a comparison to the work by Arous et al. 2025 who also study scaling law for quadratic models?


Additional feedback:
The manuscript contains numerous typos and overly strong statements. Please correct and avoid them. In addition, I would remove general assumption 3.2 and 4,1 together with Theorem 4.2, as it doesn't add more to the result. Unless the author shows some more indication on the validity of the assumption and the estimation of the eigenvalues, even with numerical simulation. In general, I would add more explanation and clarify the proof steps of Theorem 4.1.


1. Arous, Gérard Ben, et al. "Learning quadratic neural networks in high dimensions: SGD dynamics and scaling laws." arXiv preprint arXiv:2508.03688 (2025).
2. Goldt, S., Mézard, M., Krzakala, F., & Zdeborová, L. (2020). Modeling the influence of data structure on learning in neural networks: The hidden manifold model. Physical Review X, 10(4), 041044.
3. Collins-Woodfin, E., Paquette, C., Paquette, E., & Seroussi, I. (2024). Hitting the high-dimensional notes: An ode for sgd learning dynamics on glms and multi-index models. Information and Inference: A Journal of the IMA, 13(4), iaae028.

---

### Official Review · Reviewer_BiCJ · 2025-10-31

**Soundness:** 3
**Presentation:** 2
**Contribution:** 2
**Rating:** 4
**Confidence:** 4

**Summary:**

This paper obtains the scaling laws for quadratically-parameterized linear regression models, optimized using SGD.

**Strengths:**

1. This work successfully shows a separation between the linear model and the quadratically-parameterized model.
2. The theoretical results are supported by experiments.

**Weaknesses:**

1. My major concern is about how quadratically parameterized model resembles feature learning. I do not fully understand the argument in Lines 217-219 (especially the "discriminative features"), which is not supported by any references either. Why not study the single-index model (i.e., linear model with ReLU activation) that is probably closer to practical neural networks?
2. The remainder term $\xi$ and $\zeta_M$ requires further explanations, since they do not appear in Lin et al. (2024).
3. A phase plane like Figure 4(a) in Paquette et al. (2024) could make the results more interpretable.
4. Further illustration around the optimality of the learning rate $\eta$ in Theorem 4.1 is expected (along with a comparison against the linear model).

Lin et al., Scaling laws in linear regression: Compute, parameters, and data. 2024.

Paquette et al., 4+3 phases of compute-optimal neural scaling laws. 2024.

**Questions:**

1. In Lin et al. 2024, the high-probability event is with respect to the randomness of $\mathbf{S}$. However, Assumptions A1 and A2 already discussed the distribution of $\mathbf{Sx}$ (which is exactly the high-probability event that Lin et al. (2024) tried to characterize), so according to my understanding, it is more natural to bound the excess risk **in expectation** in Theorem 4.1, instead of the current version.
2. Lin et al. failed to obtain the results for $\beta\ge\alpha+1$, while this paper does not seem to have such a constraint. What are the specific techniques, or the specific properties of quadratically-parameterized models, that enable this?
3. Can the authors further explain (an intuition would be great) the separation between the linear model and the quadratically-parameterized model in the regime of $\alpha>\beta$?
4. It seems that in Figure 1, the risk curve of quadratically-parameterized model is far less stable than the linear model. Combined with Weakness 4, is it possible that the quadratically-parameterized model requires a smaller learning rate, while the learning rate in the experiments is too large?

---

### Official Review · Reviewer_Pant · 2025-10-31

**Soundness:** 3
**Presentation:** 2
**Contribution:** 2
**Rating:** 4
**Confidence:** 3

**Summary:**

This paper studies scaling laws for SGD applied to linear regression with a quadratically parameterized model of the form $f(\mathbf{x}) = \langle \mathbf{S} \mathbf{x}, \mathbf{v}^{\odot 2} \rangle$, where $\mathbf{S}$ is the sketch matrix, $\mathbf{x}$ is the input data, and $\mathbf{v}$ are the model parameters. While the model can be seen as a diagonal linear neural network, this parametrization allows the authors to analyze scaling laws in the feature learning regime. In comparison to the prior literature, this work considers an anisotropic data covariance and studies the feature learning regime, while providing explicit bounds (and scaling laws) on the excess risk.

**Strengths:**

1. Good theory paper with clearly stated assumptions, theorems, and their proofs. Additionally, the authors explicitly state the theoretical challenge they faced (due to their setting) in comparison to the prior work.

2. Clear motivation: analyzing the excess risk for SGD in a feature learning regime while the data distribution exhibits anisotropic covariance with power-law decay rates.

3. The paper is mostly well-written and easy to follow.

**Weaknesses:**

1. Part of the gap the paper aims to address has already been covered by two NeurIPS 2025 papers:
   * Arous et al., "Learning quadratic neural networks in high dimensions: SGD dynamics and scaling laws." NeurIPS 2025.
   * Ren et al., "Emergence and scaling laws in SGD learning of shallow neural networks." NeurIPS 2025.

These papers also address the study of scaling laws for SGD in the feature learning regime. However, I would like to note that the mentioned works assume isotropic Gaussian inputs, whereas the current paper considers anisotropic Gaussian data, in addition to other differences in the settings. Still, the authors should discuss their contribution relative to these works.

2. The paper does not provide any intuition or take-away message that is solicited from the theoretical results, limiting the impact factor of the paper.

3. The empirical/experimental side of the paper is too weak. Extensions to real-world data settings should be considered.

4. Similarly, the considered setting seems pretty artificial. It is unclear to me how the results in this paper can enhance our theoretical understanding of scaling laws for SGD or how they can be applied to generate practical insights.

**Questions:**

1. Could the authors clarify the additional contribution/novelty of this work compared to the papers I mentioned above?

2. Could the authors provide experimental results in a real-world data scenario?

3. Could the authors explicitly state the intuitions or take-away messages (based on their results) that improve our understanding of scaling laws for SGD or apply to practical settings?

---

### Official Review · Reviewer_Gs5K · 2025-10-31

**Soundness:** 3
**Presentation:** 2
**Contribution:** 1
**Rating:** 4
**Confidence:** 3

**Summary:**

The paper studies scaling laws for quadratically parameterized linear models

**Strengths:**

The results appear to be rigorously formulated and proven in detail

**Weaknesses:**

I am not sure I understand significance and novelty of the problem. Denoting $w = v^{\odot}$, we know that the model is now linear in $w$.  Thus, the new implicit bias of the corresponding problem  will be $\|w\| = \|v\|_4^2$ with an additional constraint $w \ge 0$, according to "The Implicit Bias of Gradient Descent on Separable Data" by Soudry et al. Since the data is assumed to be Gaussian, the corresponding generalization error and other properties of the learned solutions could be analyzed precisely using methods like Convex Gaussian Min-Max theorem or Approximate Message Passing.  Thus, I would like to ask the authors to shed more light on the significance of the results and methods used in this work.

**Questions:**

1) Please address the novelty question from the weaknesses part

2) In line 216, what do $v^{\odot 2}_+$ and $v_{\odot 2}-$ mean? I am confused because  $v^{\odot 2}$ is already entry-wise positive.

3)  I see Figure 1 with a presentation of some empirical results, but I couldn't find a section describing the details of the simulations within the main body. Could you please describe the details?

---

### Official Review · Reviewer_pSEo · 2025-11-02

**Soundness:** 2
**Presentation:** 2
**Contribution:** 2
**Rating:** 2
**Confidence:** 5

**Summary:**

This paper studies scaling laws in an infinite dimensional linear regression problem under source and capacity conditions. They consider a two-layer diagonal linear network trained by online SGD. Under suitable assumptions on the alignment between the network parameterization and the unknown data covariance matrix, they obtain sharp excess risk bounds on the learnt model, exhibiting a power-law type scaling law on their dependence on sample size and model size. Their bound suggests that in a certain regime, the two-layer diagonal linear network parameterization enables SGD to achieve a better rate than a typical linear parameterization. This paper is of theory nature; while simulation is provided, they are not critical to the key contribution of this work.

**Strengths:**

See below.

**Weaknesses:**

See below.

**Questions:**

I have served as a reviewer of this paper earlier this year — it is sad to see many of the issues still exist in the current version. Specifically, I (still) have the following two concerns.

1. Assumption 3.1[A1] is prohibitively restrictive. This gives the two-layer diagonal linear network an advantage of knowing the coordinate system of the data covariance matrix, which is typically considered an unknown information in literature (and in practice). Compared to the last version that I reviewed, this version makes efforts to fix this issue by introducing Assumptions 3.2[A3] and 4.1. However, these two assumptions basically suggest the coordinate system is known up to a sufficiently small error — this is not really a fix, but simply a lazy way of cheating.

2. The conditions on the sketching matrix, buried in Assumptions 3.1, 3.2, and 3.3, are prohibitively restrictive. The earlier version that I reviewed assumes the sketching matrix selects precisely the top $M$ features — which is unrealistic. This version, however, assumes the same, but hides it into the assumptions statements, instead of saying it explicitly. In comparison, the prior works [Lin et al 2024, Paquette et al 2024] all considered a uninformative isotropic Gaussian sketch — one immediately sees why the sketching matrix considered in this paper is problematic. Additionally, this paper writes some incorrect statements saying that their sketching assumptions follow from prior work, which is rather unscholarly.

I think this paper has made some very interesting and legit findings on how the two-layer diagonal linear network parameterization improves the sample complexity of SGD over the linear parameterization. Unfortunately, this version (as well as the earlier version) chooses to wrap their real contribution around the buzzy word “scaling laws”. In my opinion, authors should have just written a paper on how a two-layer diagonal linear network improves linear parameterization, removing all the sketching stuff for the sake of making up scaling laws — their sketching is fake anyways. In this way, their paper might receive much less criticism.

I have a technical question:

3. Why are all the theorems stated to hold with a constant probability? What prevents obtaining bounds in high probability or expectation?


Minor issues:

4. In Theorem 1, if equation (2) provides matching upper and lower bounds, then line 332 “The error of output can be bounded from above by” should be revised.
5. The discussions in lines 40-49 originate from [Lin et al 2024], which should be cited properly in this piece of text to give them proper credit.
6. In line 119, $R_M$ and $\xi$ are used without being given proper definitions.
7. Lines 129-130, “and the excess risk as…” should be removed.
8. Line 198, “a universal constants” -> ”a universal constant”.
9. Line 310. This is not considered as “warm up” typically, as the initial stepsize is already large.
10. Line 383. I believe the exponent of $1/T$ should be $1-1/\beta$ instead of $1/\beta$.

Some of these minor issues were flagged in the review of their earlier version. It is rather discouraging to see their existence in the newer version -- feels the review efforts were wasted  :(

---

### Note · Authors · 2025-11-24

I have read and agree with the venue's withdrawal policy on behalf of myself and my co-authors.